# Addressing soil data needs and data-gaps in catchment scale environmental modelling: the European perspective

Brigitta Szabó[1,2], Piroska Kassai[1,2], Svajunas Plunge[3], Attila Nemes[4], Péter Braun[1,2,5], Michael Strauch[6], Felix Witing[6], János Mészáros[1,2], Natalja Čerkasova[5,7]

[1]Institute for Soil Sciences, HUN-REN Centre for Agricultural Research, Budapest, 1022, Hungary
[2]National Laboratory for Water Science and Water Security, Budapest, 1022, Hungary
[3]Department of Hydrology, Meteorology, and Water Management, Institute of Environmental Engineering, Warsaw University of Life Sciences, Warsaw, 0-653, Poland
[4]Norwegian Institute of Bioeconomy Research, Ås, 1431, Norway
[5]Marine Research Institute, Klaipeda University, Klaipeda, 92294, Lithuania
[6]Helmholtz Centre for Environmental Research GmbH - UFZ, Department of Computational Landscape Ecology, Leipzig, 04318, Germany
[7]Texas A&M AgriLife, Blackland Research and Extension Center, Temple, TX 76502, USA

*Correspondence to*: Piroska Kassai (kassai.piroska@atk.hun-ren.hu)

**Abstract.** To effectively guide agricultural management planning strategies and policy, it is important to simulate water quantity and quality patterns and to quantify the impact of land use and climate change on soil functions, soil health, hydrological, and other underlying processes. Environmental models that depict alterations in surface and groundwater quality and quantity at a catchment scale require substantial input, particularly concerning movement and retention in the unsaturated zone. Over the past few decades, numerous soil information sources, containing structured data on diverse basic and advanced soil parameters, alongside innovative solutions to estimate missing soil data, have become increasingly available. This study aims to: i) catalogue open-source soil datasets and pedotransfer functions (PTFs) applicable in simulation studies across European catchments, ii) evaluate the performance of selected PTFs and iii) present compiled R scripts proposing estimation solutions to address soil physical, hydraulic, and chemical soil data needs and gaps in catchment-scale environmental modelling in Europe. Our focus encompassed basic soil properties, bulk density, porosity, albedo, soil erodibility factor, field capacity, wilting point, available water capacity, saturated hydraulic conductivity, and phosphorus content. We aim to recommend widely supported data sources and pioneering prediction methods that maintain physical consistency, and present them through streamlined workflows.

## 1 Introduction

The availability of raw and derived soil datasets, specifically soil hydraulic data, has increased significantly in Europe over the last 10 years as a results of the European Green Deal through initiatives and strategies aimed at promoting sustainable land use, soil health, and environmental protection (Montanarella and Panagos, 2021). Both the collection and harmonisation of soil datasets and the preparation of soil maps have intensified. Further to these improvements, the derivation of prediction

algorithms, which can compute specific soil properties from easily available soil or other environmental variables (the pedotransfer functions (PTFs)) has continued to be refined since the 1980s. The growing amount of spatiotemporal

environmental data opens up possibilities for different prediction approaches, which is reflected in the terminology of the transfer functions, e.g. i) the classical PTFs mostly use only soil properties as input (Bouma, 1989), ii) those PTFs that consider not only soil properties but other environmental variables as well, are called covariate-based geo transfer functions (Gupta et al., 2021a), iii) spectral transfer functions predict non easily available soil properties from spectral data (Babaeian et al., 2015), while machine-learning-based (ML-based) soil mapping fuses prediction algorithms with geostatistical methods (Romano et

al., 2023). All these improvements resulted in the emergent availability of soil maps at global, regional, and local scales.

The basic soil properties, i.e., soil organic carbon content, particle size distribution, in most cases are locally available at high resolution (< 100 m), but information on bulk density, albedo, soil erodibility factor, soil hydraulic properties, and soil nutrient content is often lacking. There are many PTFs available in the literature that can be used to calculate soil physical (Abbaspour et al., 2019) and hydrological (Bouma and van Lanen, 1987; Van Looy et al., 2017) parameters from basic soil properties, but

determining the most suitable one might not be obvious. Parameter estimations derive the parameters of a model that describes either water retention, hydraulic conductivity, or both across the entire matric potential range. These estimations aim to ensure a cohesive physical relationship between the computed soil hydraulic properties.

Information on soil nutrient properties often essential for environmental modelling, such as plant-available soil phosphorus or soil nitrate content, is seldom accessible at a catchment or regional scale. In the absence of measured data on nutrient content,

estimating highly mobile nutrients like nitrate poses a challenge due to seasonal fluctuations influenced by factors such as fertilizer application, rainfall, harvest cycle, plant nutrient uptake, and microbial activity. Regarding plant-available phosphorus, its levels typically exhibit minimal variation throughout a year. Therefore, approximating its quantity could rely on land use type and area-specific phosphorus fertilization loads (Ballabio et al., 2019). Nevertheless, multiple methods are employed across Europe to measure plant-available soil phosphorus content, potentially requiring conversions between these

methods for broader-scale applications. A comprehensive review on conversion equations is available specifically for European studies in Steinfurth et al.(2021).

Often those soil properties are required as model input data as well, which are rarely available. One example is the data on soil cracking, which is rarely readily available. Cracking intensity and number of cracks are determined by i) soil mineralogy, specifically the amount and type of clay minerals, ii) type of strength that forms soil structure (Lal and Shukla, 2004) and iii)

human activity, e.g. tillage, plant spacing. The aperture and closure of cracks can be dynamically related to soil water content (Xing et al., 2023). The data that could describe the variability of cracking is also not easily available, therefore prediction of this parameter is limited at catchment scale.

Dai et al. (2019b) provides an extensive review on global soil property maps applicable for Earth system models. Abbaspour et al. (2019) collected both soil datasets and pedotransfer functions for global Soil and Water Assessment Tool (SWAT)

applications. From these global comprehensive review studies and a variety of soil datasets available among others from the

**Table 1.** Open access soil data that could be applied for environmental modelling in Europe.

| Name of dataset | Data type | Horizontal resolution | Vertical resolution (cm) | Available soil properties* | Coverage | Availability | Notes | References |
|---|---|---|---|---|---|---|---|---|
| **Soil basic data** | | | | | | | | |
| LUCAS Topsoil dataset | topsoil point data | - | 0-20 | coarse, Sa, Si, Cl, pH_H2O, pH_CaCl2, OC, CaCO3, P, N, K_ext, CEC, MSP, HM | European Union and some other countries | https://esdac.jrc.ec.europa.eu/projects/lucas | from 2018 BD, soil biodiversity indicators, visual assessment of soil erosion and depth of organic soil are available, too (Fernandez-Ugalde et al., 2022) | Orgiazzi et al. (2018), Tóth et al. (2013) |
| SPADE 2 | soil profile point data | - | actual soil depth | ST, coarse, Sa, Si, Cl, BD, OC, pH_H2O, pH_CaCl2, pH_KCl | European Union | https://esdac.jrc.ec.europa.eu/content/soil-profile-analytical-database-2 | includes information on water regime class and water management type | Hannam et al. (2009) |
| SoilGrids | map | 250 m | 0-5, 5-15, 15-30, 30-60, 60-100, 100-200 | ST, coarse, Sa, Si, Cl, BD, OC, pH_H2O, CEC, N, with information on uncertainty. | global | https://soilgrids.org/ | - | Poggio et al. (2021) |
| OpenLand-Map | map | 250 m | 0, 10, 30, 60, 100, 200 | ST, Sa, Si, Cl, OC, BD, pH_H2O, other | global | https://openlandmap.org | - | https://openlandmap.org |
| Topsoil chemical properties for Europe | map | 500 m | 0-20 | pH_H2O, pH_CaCl2, CEC, CaCO3, C:N ratio, P, N, K_ext | European Union | https://esdac.jrc.ec.europa.eu/content/chemical-properties-european-scale-based-lucas-topsoil-data | can be used with topsoil physical properties for Europe dataset (Ballabio et al., 2016) | Ballabio et al. (2019) |

Table 1. cont.

| Name of dataset | Data type | Horizontal resolution | Vertical resolution (cm) | Available soil properties* | Coverage | Availability | Notes | References |
|---|---|---|---|---|---|---|---|---|
| **Soil hydraulic or physical data** | | | | | | | | |
| EU-SoilHydro-Grids | map | 250 m | 0, 5, 15, 30, 60, 100, 200 | THS, FC, WP, KS, VG, MVG | Europe | https://elkh-taki.hu/en/eu_soilhydrogrids_3d , https://esdac.jrc.ec.europa.eu/content/3d-soil-hydraulic-database-europe-1-km-and-250-m-resolution | can be used with the SoilGrids 2017 (Hengl et al., 2017) dataset | Tóth et al. (2017) |
| Montzka et al. (2017) | map | 0.25° | 0, 5, 15, 30, 60, 100, 200 | VG, KS and scaling parameters based on 1 km SoilGrids 2017 | global | https://doi.org/10.5194/essd-9-529-2017 | can be used with the SoilGrids 2017 (Hengl et al., 2017) dataset | Montzka et al. (2017) |
| Zhang and Schaap (2018) | map | 1 km | 0, 5, 15, 30, 60, 100, 200 | KO, FC, AWC with standard deviations | global | https://dataverse.harvard.edu/dataset.xhtml?persistentId=doi:10.7910/DVN/UI5LCE | can be used with the SoilGrids 2017 (Hengl et al., 2017) dataset | Zhang and Schaap (2018) |
| Zhang et al. (2020) | map | 10 km | 0, 5, 15, 30, 60, 100, 200 | THS, FC, WP with coefficient of variation | global | https://dataverse.harvard.edu/dataset.xhtml?persistentId=doi:10.7910/DVN/VPIN2B | can be used with the soil basic data of the OpenLandMap dataset | Zhang et al. (2020) |
| Gupta et al. (2022) | map | 1 km | 0, 30, 60, 100 | VG | global | https://doi.org/10.5281/zenodo.6348799 | - | Gupta et al. (2022) |
| Gupta et al. (2021b) | map | 1 km | 0, 30, 60, 100 | KS | global | https://doi.org/10.5281/zenodo.3935359 | - | Gupta et al. (2021b) |
| Topsoil physical properties for Europe | map | 500 m | 0-20 | coarse, Sa, Si, Cl, BD, TEX, AWC | European Union | https://esdac.jrc.ec.europa.eu/content/topsoil-physical-properties-europe-based-lucas-topsoil-data | can be used with topsoil chemical properties for Europe dataset (Ballabio et al., 2019) | Ballabio et al. (2016) |

**Table 1.** cont.

| Name of dataset | Data type | Horizontal resolution | Vertical resolution (cm) | Available soil properties* | Coverage | Availability | Notes | References |
|---|---|---|---|---|---|---|---|---|
| Soil bulk density in Europe | map | 100 m | 0-20, 0-10, 10-20 | BD | European Union | https://esdac.jrc.ec.europa.eu/content/soil-bulk-density-europe | - | Panagos et al. (2024) |
| Panagos et al. (2014) | map | 500 m | - | K_fact | Europe | https://esdac.jrc.ec.europa.eu/content/soil-erodibility-k-factor-high-resolution-dataset-europe | can be used with the LUCAS Topsoil dataset | Panagos et al. (2014) |
| Global soil erodibility | map | 1 km | - | K_fact | global | https://esdac.jrc.ec.europa.eu/content/global-soil-erodibility | can be used with the SoilGrids 2021 (Poggio et al., 2021) dataset | Gupta et al. (2024) |
| Surface albedo, MCD43A3 | map | 500 m | - | ALB | global | https://doi.org/10.5067/MODIS/MCD43A3.061 | 16-daily data | https://doi.org/10.5067/MODIS/MCD43A3.061 |
| Surface albedo (Copernicus Climate Change | map | 300 m, | - | ALB | global | https://doi.org/10.24381/cds.ea87ed30 | 10-daily data | Copernicus Climate Change Service (2018) |
| HYSOGs250m | map | 250 m | 0, 5, 15, 30, 60, 100 | HSG | global | https://doi.org/10.3334/ORNLDAAC/1566 | can be used with the SoilGrids 2017 (Hengl et al., 2017) dataset | Ross et al. (2018) |
| **Soil basic and hydraulic data** | | | | | | | | |
| WOSIS | soil profile point data | - | actual soil depth | coarse, Sa, Si, Cl, BD, OC, pH_H2O, CaCO3, CEC, TOC, P, FC, WP, other | global | https://soilgrids.org/ | - | Batjes et al. (2020) |
| HWSD v 2.0 | map | 1 km | 0-20, 20-40, 40-60, 60-80, 80-100, 100-150, 150-200 | ST, coarse, Sa, Si, Cl, BD, OC, pH_H2O, CaCO3, CEC, EC, D_C, AWC_C, IL, RSD, SWR, other | global | https://gaez.fao.org/pages/hwsd | - | FAO & IIASA (2023) |

**Table 1.** cont.

| Name of dataset | Data type | Horizontal resolution | Vertical resolution (cm) | Available soil properties* | Coverage | Availability | Notes | References |
|---|---|---|---|---|---|---|---|---|
| DSOLMap | map | 250 m | 0-5, 5-10, 10-30, 30-60, 60-100, 100-200 | coarse, Sa, Si, Cl, BD, OC, AWC, KS, K_fact, ALB_m | global | https://www.wateritech.com/data | data is based on OpenLandMap, and is in format of the SWAT model | López-Ballesteros et al. (2023) |
| GSDE (2014) | map | 30″ (~1km) | 0-4.5, 4.5-9.1, 9.1-16.6, 16.6-28.9, 28.9-49.3, 49.3-82.9, 82.9-138.3, 138.3-229.6 | ST, coarse, Sa, Si, Cl, BD, OC, TOC, pH_H2O, CaCO3, CEC, EC, N, P, D_C, AWC_C, IL, RSD, SWR, other | global | http://globalchange.bnu.edu.cn/research/soilw | dataset is based on the Soil Map of the World and various regional and national soil databases | Shangguan et al. (2014) |
| Dai et al. (2019) | map | 30″ (~1km) | 0-5, 5-15, 15-30, 30-60, 60-100, 100-200 and specific layering | coarse, OM, Sa, Si, Cl, Qa, CB, MVG, SHC, STC | global | http://globalchange.bnu.edu.cn/research/soil5.jsp | can be used with GSDE and SoilGrids 2017 | Dai et al. (2019a) |

*ST: soil type; coarse: coarse fragments; Sa, Si, Cl: sand, silt and clay content; Qa: quartz content of the mineral soil; BD: bulk density; OM: organic matter content; OC: organic carbon content; TOC: total organic carbon content; pH_H2O: pH in water; pH_CaCl2: pH in calcium chloride; pH_KCl: pH in potassium chloride; CaCO3: calcium carbonate content; CEC: cation exchange capacity; EC: electrical conductivity; N: total nitrogen content; K_ext: extractable potassium content; P: phosphorus content extracted by Olsen method; MSP: multispectral properties; HM: heavy metals; K_fact: soil erodibility (K-factor) based on the Revised Universal Soil Loss Equation; ALB: surface albedo; ALB_m: moist soil albedo; HSG: hydrological soil groups based on the National Engineering Handbook(U.S. Department of Agriculture Natural Resources Conservation Service, 2009) of the U.S. Department of Agriculture-Natural Resources Conservation Service; D_C: drainage class, AWC_C: available water capacity class in the rootable depth; IL: depth class to impermeable layer; RSD: rootable soil depth class; SWR: soil water regime class; THS: saturated water content; FC: water content at field capacity; WP: water content at wilting point; KS: saturated hydraulic conductivity; VG: parameters of the van Genuchten model ($\theta$s, $\theta$r , $\alpha$, n, m) to describe the soil water retention curve; MVG: parameters of the Mualem-van Genuchten model to describe the soil water retention and hydraulic conductivity curve ($\theta$s, $\theta$r, $\alpha$, n, m, K0, L); CB: parameters of the Campbell model ($\theta$s, $\psi$, $\lambda$, KS) to describe the soil water retention and hydraulic conductivity curve ($\theta$s, $\theta$r, hm, $\sigma$, KS); KO parameters of the Kosugi model to describe the soil water retention and hydraulic conductivity curve ($\theta$s, $\theta$r, hm, $\sigma$, KS); SHC: heat capacity of soil solids; STC: soil thermal conductivity of unfrozen saturated soil, frozen saturated soil and dry soil; other: other soil properties, which are less frequently required by environmental models, AWC: available water capacity; TEX: USDA soil texture class.


European Soil Data Centre (Panagos et al., 2022) (https://esdac.jrc.ec.europa.eu/) or ISRIC – World Soil Information (https://www.isric.org/), it is not straightforward which data and/or pedotransfer functions could be used for the environmental modelling in European case studies. Therefore, in this study we support soil data retrieval for environmental modelling across Europe by i) systemizing information on open access datasets and PTFs applicable for Europe, ii) demonstrating and quantifying the difference between some PTFs and prediction approaches to cover missing soil properties based on the point data of EU-HYDI, and iii) providing a comprehensive workflow and accompanying open-source R script and library for the derivation of missing soil data. For the selection of the prediction approaches, three requirements had to be fulfilled: 1) the prediction algorithm had to be trained on temperate soils and should not be specific to a particular soil reference group, 2) the required predictors had to be available in most of the open access soil datasets, and 3) its ease of application. Hence, despite certain published prediction methods specifying soil depth, texture, and organic matter as requirements, those reliant on, for instance, artificial neural networks, lacking a user-friendly interface, or integration into accessible tools like R packages or Python modules, were excluded from testing due to their challenging application. For ease of reference, all the equations needed to calculate the most often required soil input parameters are given below.

## 2 Materials and methods

We distinguish and list soil physical and chemical parameters similarly to the terminology used by the Soil and Water Assessment Tool model documentation (Neitsch et al., 2009). We include the prediction of soil porosity since this parameter is frequently used in environmental models, e.g. MIKE SHE (DHI, 2023), HEC RAS (US Army Crops of Engineers, 2023), PIHM (Li and Duffy, 2011). Noteworthy that some models and accompanying model setup tools have an internal built-in PTF to compute porosity, e.g. SWAT+. The codes to compute the soil parameters were built based on the structure and terminology used by the SWAT+ *usersoil* table (Arnold et al., 2012). Soil properties most frequently required by the environmental models – e.g. (Abbaspour et al., 2019; Dam et al., 2008; Dang et al., 2022; DHI, 2023; Hansen et al., 2012; Šimůnek et al., 2012; Yu et al., 2020) – are:

- soil layering,
- maximum rooting depth,
- effective bulk density,
- field capacity,
- wilting point,
- available water capacity,
- porosity,
- saturated hydraulic conductivity,
- organic carbon content,
- sand, silt, and clay content,

- rock fragment content,
- moist soil albedo,
- Universal Soil Loss Equation (USLE) soil erodibility factor,
- hydrologic soil group, and
- nutrient content of the surface soil layer.

We summarised information about potential open access sources for soil information applicable in Europe in Table 1, covering most of the above listed soil properties. The availability of datasets is continuously improving. The following data sites include most of the updates:

- European Soil Data Centre, which includes soil datasets from Europe and information on EU Soil Observatory (https://esdac.jrc.ec.europa.eu/),
- ISRIC Soil Data Hub, which hosts soil data from around the world (https://data.isric.org/geonetwork/srv/eng/catalog.search#/home),
- soil related layers of the GAEZ Data Portal developed by the Food and Agriculture Organization of the United Nations (FAO) and the International Institute for Applied Systems Analysis (IIASA) (https://data.apps.fao.org),
- soil related layers of the OpenLandMap, which shares open geographical and geoscientific data (https://openlandmap.org).

Nevertheless, these sources do not include products from specific institutes, such as http://globalchange.bnu.edu.cn/research. The datasets included in Table 1 might be appropriate for regional and continental modelling. However, for catchment scale and national studies, local and national spatially explicit modelled datasets provide more accurate input information. When a certain local dataset is selected to be used as basic soil information, it is more consistent to compute the missing soil properties from this local data source rather than using other data sources. This allows to maintain consistency between the different soil properties. For example, it is not recommended to combine local soil property maps at 100 m resolution with soil hydraulic properties retrieved from EU-SoilHydroGrids at 250 m resolution. Where local soil maps with soil layering, organic carbon content, clay, silt, and sand content are available, it is suggested that missing soil properties, such as bulk density, soil hydraulic properties, and albedo are estimated from the locally available basic soil properties to ensure consistency. The predictions are subject to uncertainty, which depends on the similarity between the training data used for the selected prediction method and the target area in terms of soil physical and chemical characteristics (Román Dobarco et al., 2019; Tranter et al., 2009).

## 2.1 Evaluation of methods

For soil physical and hydrological properties, the performance of the prediction algorithms was assessed using the European Hydropedological Data Inventory (EU-HYDI), specifically focusing on soil parameters with available measured values in the dataset. The EU-HYDI is one of the most comprehensive European soil hydraulic datasets, which has soil data of 18,682

samples from 6,014 profiles (Weynants et al., 2013). The number of measured values varies by soil properties. EU-HYDI dataset was used to derive hydraulic PTFs, called euptfs. When comparing the performance of euptf with other methods found in the literature, only the test sets from the EU-HYDI dataset, which were not utilized in euptf's training, were included. This approach aimed to facilitate a more accurate and fair comparison among different PTFs, but decreased the number of samples used for the analysis. The analysis of bulk density prediction was performed on both the EU-HYDI and the LUCAS Topsoil dataset (Orgiazzi et al., 2018; Tóth et al., 2013) of 2018. The LUCAS Topsoil dataset of 2009 was used for the computation of nutrient content of the surface soil layer. For the assessment of the topsoil phosphorus maps, we used locally measured data obtained from an agricultural company. This dataset includes soil phosphorus content measured at a depth of 30 cm using the acid ammonium acetate lactate extraction (AL-P) method (Egnér et al., 1960) for 34 agricultural parcels in the year 2009. As the phosphorus content was required according to the Olsen method (Olsen-P) (Olsen et al., 1954), we applied the equation of Sárdi et al. (2009) for converting AL-P into Olsen-P. Table 2 shows the descriptive statistics of this database.

**Table 2.** Descriptive statistics of the locally measured phosphorus content, converted to Olsen-P, from 34 agricultural parcels.

| Min | Max | Range | Mean | Median | Standard deviation |
|---|---|---|---|---|---|
| 8.39 | 65.02 | 56.63 | 27.54 | 25.73 | 13.47 |

We compared the algorithms using the mean error (ME), mean absolute error (MAE), root mean squared error (RMSE), Nash-Sutcliffe efficiency (NSE), and coefficient of determination ($R^2$). The non-parametric Kruskal-Wallis test of the R package agricolae (De Mendiburu, 2017) at the 5% significance level was applied on the squared error values to asses if there were significant difference in performance. For soil parameters without measured data in the EU-HYDI dataset, descriptive statistics and histograms of the computed parameters were compared with studies from peer-reviewed literature focused on European applications. The statistical analysis was performed using R statistics library (R Core Team, 2022).

## 2.2 Analysed soil properties

We analysed soil physical, hydraulic, and chemical parameters. Under soil physical parameters, we addressed bulk density, porosity, albedo, and soil erodibility factor. For soil hydraulic parameters, we examined water retention, saturated hydraulic conductivity and hydrological soil groups. Regarding soil nutrient content, we focused on topsoil phosphorus content and described the challenges of retrieving soil nitrate content. Hereinafter information about the analysis by soil properties is provided.

### 2.2.1 Soil physical parameters

**Bulk density**

Table 3 lists the PTFs that were tested on point data in EU-HYDI and 2018 LUCAS Topsoil dataset. We selected the bulk density PTFs – derived on soils of the temperate region – based on previous works (Casanova et al., 2016; Hossain et al., 2015; Palladino et al., 2022; Xiangsheng et al., 2016) that tested the prediction performance of several methods.

**Table 3.** List of pedotransfer functions tested on point data in EU-HYDI for the prediction of bulk density.

| Name of the PTF | Equation | Reference | Eq. |
|---|---|---|---|
| BD_Rawls | $BD = \dfrac{100}{\left(\left(\dfrac{OM}{0.224}\right) + \dfrac{100 - OM}{1.27}\right)}$ | (Rawls, 1983) | (1) |
| BD_Alexander_A | $BD = 1.72 - 0.294 \cdot OC^{0.5}$ | (Alexander, 1980) | (2) |
| BD_Alexander_B | $BD = 1.66 - 0.308 \cdot OC^{0.5}$ | (Alexander, 1980) | (3) |
| BD_MAn_J_A | $BD = 1.510 - 0.113 \cdot OC$ | (Manrique and Jones, 1991) | (4) |
| BD_MAn_J_B | $BD = 1.66 - 0.318 \cdot OC^{0.5}$ | (Manrique and Jones, 1991) | (5) |
| BD_Hollis | -for cultivated topsoils: $BD = 0.80806 + \left(0.823844 \cdot (\exp(-0.27993 \cdot OC))\right) + 0.0014065 \cdot sand - 0.0010299 \cdot clay$ <br> - for mineral subsoils: $BD = 0.69794 + \left(0.750636 \cdot (\exp(-0.230355 \cdot OC))\right) + 0.0008687 \cdot sand - 0.0005164 \cdot clay$ <br><br> - for organic horizons*: $BD = 1.4903 + 0.33293 \cdot \log(OC)$ | (Hollis et al., 2012) | (6) |
| BD_Bernoux | $BD = 1.398 - 0.042 \cdot OC - 0.0047 \cdot clay$ | (Bernoux et al., 1998) | (7) |
| BD_Hossain ** | $BD = 0.074 + 2.632 \cdot \exp(-0.076 \cdot OC)$ | (Hossain et al., 2015) | (8) |

* For histic and follic horizons, which have organic carbon content equal to or greater than 20 % (IUSS Working Group WRB, 2022). **Applied only for organic soils with organic carbon content equal to or greater than 12 %. OM: organic matter content (mass %); OC: organic carbon content (mass %); sand: sand content (0.05-2 mm fraction) (mass %); clay: clay content (<0.002 mm fraction) (mass %).

**Porosity**

Porosity can be computed based on the bulk density and particle density with the following equation:

$$POR = \left(1 - \left(\frac{BD}{PD}\right)\right) \cdot 100 \tag{9}$$

where POR is porosity (volume %), BD is dry bulk density (g cm$^{-3}$), PD is particle density (g cm$^{-3}$).

As seen in literature and in SWAT+ model default assumptions (Neitsch et al., 2009), the particle density is usually set as equal to 2.65 g cm$^{-3}$ (Lal and Shukla, 2004). However, there are PTFs that calculate the porosity based on the particle size

distribution (sand, silt, clay content) and organic matter content. We selected the PTFs (Table 4) based on the findings of
180 Ruehlmann (2020) and analysed their prediction performance on the EU-HYDI dataset.

**Table 4.** List of pedotransfer functions tested on point data in EU-HYDI for the prediction of porosity.

| Name of the PTF | Equation | Reference | Eq. |
|---|---|---|---|
| POR_Schjonning_etal | $PD_{OM} = 1.241 + 0.173 \cdot \left(\frac{OM}{100}\right)$ $PD_{SMS} = 2.663 + 0.107 \cdot \left(\frac{clay}{100}\right)$ $PD = \left(\frac{\left(1 - \frac{OM}{100}\right)}{PD_{SMS}} + \frac{\frac{OM}{100}}{PD_{OM}}\right)^{-1}$ $POR = \left(1 - \left(\frac{BD}{PD}\right)\right) \cdot 100$ | (Schjønning et al., 2017) | (10) |
| POR_Schjonning_etal_recal | $PD = 2.654 + 0.216 \cdot \frac{clay}{100} - 2.237 \cdot \frac{OM}{100}$ $POR = \left(1 - \left(\frac{BD}{PD}\right)\right) \cdot 100$ | (Ruehlmann, 2020) | (11) |
| POR_2_65 | $POR = \left(1 - \left(\frac{BD}{2.65}\right)\right) \cdot 100$ | (Lal and Shukla, 2004) | (12) |

$PD_{OM}$: particle density of the soil mineral substance; $PD_{MS}$: particle density of the soil organic matter; OM: organic matter
content (mass %); sand: sand content (0.05-2 mm fraction) (mass %); clay: clay content (<0.002 mm fraction) (mass %).

**Albedo**

Bare soil albedo mostly depends on soil moisture variations, surface roughness, soil texture and organic matter content (Carrer
et al., 2014). Time series of soil surface albedo could be retrieved e.g. from the MCD43A3 database or Copernicus Climate
Change Service (2018) (Table 1). If a single characteristic value of soil surface albedo is required for the entire modelling
period, such as e.g. in the case of the SWAT+ model, the study of Abbaspour et al. (2019) provides several formulas for its
computation and suggests to substitute the actual volumetric water content with field capacity. For European applications the
equation of Gascoin et al. (2009) could be used:

$ALB = 0.31 \cdot \exp(-12.7 \cdot \theta) + 0.15$ (13)

where ALB is soil albedo and $\theta$ is volumetric water content (cm$^3$ cm$^{-3}$), which could be set to the value of field capacity.
We computed the soil albedo with Eq. (13) for the EU-HYDI topsoil samples with setting the water content to saturation, field
capacity and wilting point. The EU-HYDI dataset does not include the measured soil albedo values at a certain moisture
content, therefore we extracted the median surface albedo for year 2022 from the MCD43A3 database
(https://doi.org/10.5067/MODIS/MCD43A3.061) for two cases: i) any surfaces in the entire year and ii) only dry bare soils.
We compared the descriptive statistics of computed and mapped values. We considered the visible broadband black-sky albedo
for the analysis. Dry bare soil pixels were selected using MOD09GA.061 dataset based on Normalized Difference Vegetation

Index (NDVI) and Normalized Burn Ratio 2 (NBR2) indices (Safanelli et al., 2020) in Google Earth Engine platform (Gorelick et al., 2017) when NDVI values fell in the range of -0.05 and 0.30, and NBR2 values between -0.15 and 0.15. Pixels for dry bare soils were selected to mask and compare the remote sensed soil albedo values to the albedo computed at different moisture states.

## Soil erodibility factor

The soil erodibility factor (K-factor) required for modelling soil erosion can be computed with several methods described e.g. in Kinnell (2010) or Panagos et al. (2014). The most widely used equation that can be readily applied to the most frequently available soil properties was published by Sharpley and Williams (1990) (Eq. 14) and Renard et al. (1997) (Eq. 15). The advantage of these methods is that they require only the sand, silt, clay, and organic carbon content of the soil.

$$K_{USLE} = \left( 0.2 + 0.3 \cdot \exp\left( 0.0256 \cdot sand \cdot \left( 1 - \frac{silt}{100} \right) \right) \right) \cdot \left( \left( \frac{silt}{clay + silt} \right)^{0.3} \right) \cdot$$

$$\left( 1 - \left( \frac{0.25 \cdot OC}{(OC + \exp(3.72 - 2.95 \cdot OC))} \right) \right) \cdot \left( 1 - \left( \frac{0.7 \cdot \left( 1 - \frac{sand}{100} \right)}{\left( \left( 1 - \frac{sand}{100} \right) + \exp\left( -5.51 + 22.9 \cdot \left( 1 - \frac{sand}{100} \right) \right) \right)} \right) \right) \tag{14}$$

$$K_{RUSLE} = 7.594 \left( 0.0034 + 0.0405 \cdot exp\left( -0.5 \cdot \left( \frac{log(D_g) + 1659}{0.7101} \right)^2 \right) \right) \quad \text{with } D_g = \exp(0.01 \cdot \sum f_i \cdot \ln m_i) \tag{15}$$

where $K_{USLE}$ is the Universal Soil Loss Equation (USLE), $K_{RUSLE}$ is the Revised Universal Soil Loss Equation (RUSLE) soil erodibility factor $\left( \frac{t \cdot arce \cdot h}{hundreds\ of\ acre \cdot foot - tonf \cdot inch} \right)$, silt is silt content (mass%, 0.002-0.05 mm), sand is sand content (mass %, 0.05-2 mm), OC is organic carbon content (mass %), $D_g$ is the geometric mean particle diameter (mm), $f_i$ is the particle size fraction (mass%), $m_i$ is the arithmetic mean of the particle size limits of the $f_i$ particle size fraction (mm) . If the unit is required in $\left( \frac{t \cdot ha \cdot h}{ha \cdot MJ \cdot mm} \right)$, the value of the soil erodibility factor computed with Eq. (14) or Eq. (15) has to be multiplied with 0.1317 (Foster et al., 1981).

We computed the soil erodibility factor for the EU-HYDI dataset. Similarly to the above mentioned albedo, there is no measured soil erodibility value in the EU-HYDI dataset, thus we compared the values computed for the topsoils of EU-HYDI with the values extracted from the European map of Panagos et al. (2014).

### 2.2.2 Soil hydraulic parameters

### Water retention and saturated hydraulic conductivity

Soil water retention and hydraulic conductivity can be computed from the parameters of the widely used van Genuchten model (VG) (van Genuchten, 1980):

$$\theta(\psi) = \theta_r + \frac{\theta_s - \theta_r}{[1 + (\alpha\psi^n)]^m} \quad \text{with} \quad m = 1 - 1/n \tag{16}$$

where $\theta_r$ (cm$^3$ cm$^{-3}$) and $\theta_s$ (cm$^3$ cm$^{-3}$) are the residual and saturated soil water contents, respectively, $\alpha$ (cm$^{-1}$) is a scale parameter, $m$ (-) and $n$ (-) are shape parameters.

Alternative models, like the Kosugi model (Kosugi, 1996) exist for characterizing the water retention curve. However, the availability of predictive tools for their parameters and equations to derive specific soil hydraulic properties (such as saturated hydraulic conductivity and field capacity based on internal drainage dynamics) from these parameters is either limited or non-existent (Zhang et al., 2018). Utilizing the VG model to compute all necessary soil hydraulic properties ensures self-consistency of parameters and relies on a dynamic criterion based on soil internal drainage dynamics (Assouline and Or, 2014; Nasta et al., 2021).

Usually, the FC is considered as water content at a static soil matric potential. The -330 cm matric potential is widely used for this approximation (Kutílek and Nielsen, 1994). Assouline and Or (2014) derived a physically-based analytical equation with self-consistent static and dynamic criteria for the prediction of FC. It requires the parameters of the van Genuchten model:

$$FC = \theta_r + (\theta_s - \theta_r)\left\{1 + \left[\frac{n-1}{n}\right]^{(1-2n)}\right\}^{\left(\frac{1-n}{n}\right)} \tag{17}$$

where FC (cm$^3$ cm$^{-3}$) is water content at field capacity, $\theta_r$ (cm$^3$ cm$^{-3}$) and $\theta_s$ (cm$^3$ cm$^{-3}$) are the residual and saturated soil water contents, respectively, $\alpha$ (cm$^{-1}$) is a scale parameter, and $n$ (-) is the shape parameter of the van Genuchten model (van Genuchten, 1980).

Computation of WP could be performed based on the VG parameters, using Eq. (18):

$$WP = \theta_r + \frac{\theta_s - \theta_r}{[1+(\alpha \cdot 15000^n)]^{1-1/n}} \tag{18}$$

AWC is defined by FC and WP with the following equation:

$$AWC = FC - WP \tag{19}$$

Physically, it is impossible to have AWC < 0, therefore its minimum value has to be set to 0.001 cm$^3$ cm$^{-3}$.

Computation of KS from parameters of the van Genuchten model can be performed by e.g. the equation of Guarracino (2007):

$$KS = 4.65 \cdot 10^4 \theta_s \alpha^2 \tag{20}$$

where KS is expressed in units of cm d$^{-1}$. If a unit in mm h$^{-1}$ is required, the Eq. (20) has to be multiplied by 0.41667.

The most frequently used pedotransfer functions, which can be used to predict soil water content and hydraulic conductivity from easily available soil information, were tested by Nasta et al. (2021) on European datasets: GRIZZLY, HYPRES and EU-HYDI. Based on their results we selected the approaches that performed well on the European datasets. Using the selected approaches, we then computed the field capacity (FC), wilting point (WP), plant available water capacity (AWC), and saturated hydraulic conductivity (KS) for the EU-HYDI dataset. The selected approaches are listed under Table 5. We considered only those approaches, which required the mean soil depth, sand, silt, clay content, organic carbon content, and bulk density as input. When FC, WP, AWC and KS is computed from the measured or predicted parameters of the VG model, it secures that all required soil hydraulic properties are derived from a physically based model, resulting in a physically plausible soil hydraulic property combination.

**Table 5.** Approaches tested in the EU-HYDI for the prediction of field capacity (FC), wilting point (WP), available water capacity (AWC) and saturated hydraulic conductivity (KS)

| Soil hydraulic property | Type of the prediction | Description | Abbreviation of the prediction | Reference |
|---|---|---|---|---|
| FC | direct | FC at -100 cm matric potential with PTF03 of euptfv2 | pred_FC_100 | (Szabó et al., 2021) |
| | direct | FC at -330 cm matric potential with PTF02 of euptfv2 | pred_FC_330 | (Szabó et al., 2021) |
| | from VG parameters | VG parameters predicted with PTF07 of euptfv2 for mineral soils and PTF18 of euptfv1 for organic soils, matric potential set to -100 cm | pred_FC_VG_100 | (van Genuchten, 1980; Szabó et al., 2021; Tóth et al., 2015) |
| | from VG parameters | VG parameters predicted with PTF07 of euptfv2 for mineral soils and PTF18 of euptfv1 for organic soils, matric potential set to -330 cm | pred_FC_VG_330 | (van Genuchten, 1980; Szabó et al., 2021; Tóth et al., 2015) |
| | from VG parameters | VG parameters predicted with PTF07 of euptfv2 for mineral soils and PTF18 of euptfv1 for organic soils + equation of Assouline and Or (2014) based on $\theta_s$, $\theta_r$ and $\alpha$ | pred_FC_VG_AO | (Assouline and Or, 2014; Szabó et al., 2021; Tóth et al., 2015) |
| WP | direct | WP at -1500 kPa with PTF02 of euptfv2 | pred_WP | (Szabó et al., 2021) |
| | direct | SWAT approach | pred_WP_SWAT | (Neitsch et al., 2009) |
| | from VG parameters | VG parameters predicted with PTF07 of euptfv2 for mineral soils and PTF18 of euptfv1 for organic soils + van Genuchten function | pred_WP_VG | (van Genuchten, 1980; Szabó et al., 2021; Tóth et al., 2015) |
| AWC | from VG parameters | AWC from pred_FC_VG_100 and pred_WP_VG | pred_AWC_VG_100 | (van Genuchten, 1980; Szabó et al., 2021) |
| | from VG parameters | AWC from pred_FC_VG_330 and pred_WP_VG | pred_AWC_VG_330 | (van Genuchten, 1980; Szabó et al., 2021) |
| | from VG parameters | AWC from pred_FC_VG_AO and pred_WP_VG | pred_AWC_VG_AO | (Assouline and Or, 2014; van Genuchten, 1980; Szabó et al., 2021) |
| KS | from VG parameters | VG parameters predicted with PTF07 of euptfv2 + equation of Guarracino (2007) based on $\theta_s$ and $\alpha$ | pred_KS_VG | (Guarracino, 2007; Szabó et al., 2021) |

### 2.2.3 Soil chemical parameters

For mapping soil phosphorus (P) content of the topsoil we present a simple approach based on mean statistics, which is suitable for areas where data is scarce. Land use has the strongest influence on soil P content, with most agricultural areas exhibiting higher P levels compared to regions with natural land cover (Ballabio et al., 2019). The available P level in agricultural soils is influenced by the P inputs – fertilizers, manure, atmospheric deposition, chemical weathering – and outputs – plant uptake and erosion. The agricultural management practices (Tóth et al., 2014) are determined by factors such as the country's

economy, climate, tillage practices, and crop production characteristics. Based on the relationships mentioned above, the geometric mean of soil P is computed by land use categories and assigned to the local land use map using the mean statistics-
270 based method. This approach comprises three main steps:

1) Selection of LUCAS Topsoil samples (EUROSTAT, 2015; Orgiazzi et al., 2018) from the adequate year and an agroclimatic zone (Ceglar et al., 2019) similar to the target area, preferably in the same country (NUTS region).Additional criteria for the data selection could be comparable soil types and fertilization systems. If this information is not known, the NUTS2 phosphorus map of the European cropland areas might be useful in the data
selection (Tóth et al., 2014).

2) Computation of the geometric mean of soil P for each land use category.

3) Assigning the mean values to the local land use map.

Further details about the mapping can be found in Szabó and Kassai (2022) .

We prepared a soil P content map by applying the proposed method for a case study called Felső-Válicka, located in Hungary
(Figure 1). The resulting map was then compared to i) the European topsoil phosphorus content map (Ballabio et al., 2019) and ii) a locally measured independent dataset provided by an agricultural company. Limited availability of soil nutrient data hampered the wider scale of comparison.

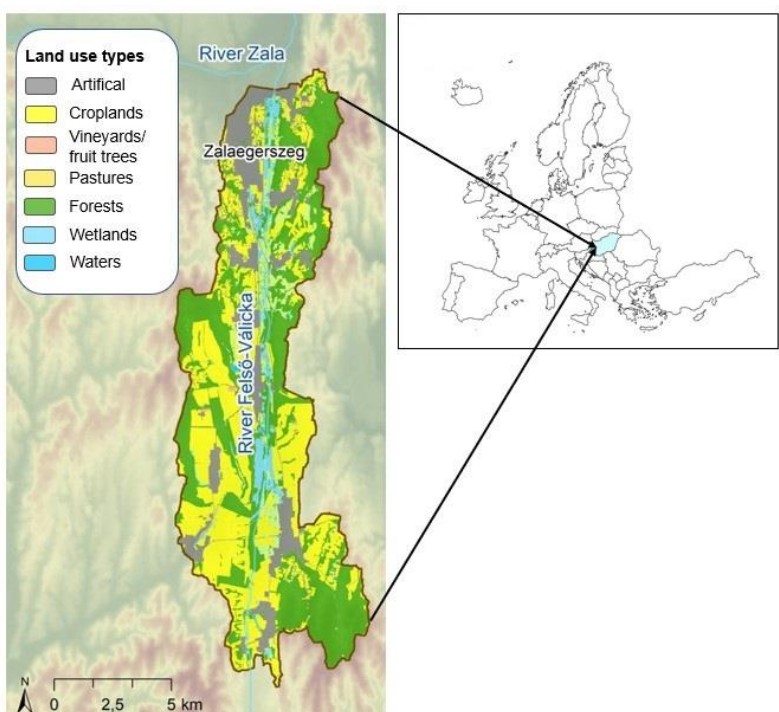

**Figure 1.** Local land use map of the Felső-Válicka case study in Hungary.

Organic nitrogen can be estimated from soil organic carbon content (Amorim et al., 2022; Liu et al., 2016; Pu et al., 2012; Zhai et al., 2019) if measured data are not available. The concentration of inorganic nitrogen in soil is highly variable in space and time and the dynamic of its amount is significantly influenced by leaching, denitrification, volatilization, and nitrogen fertilization (Zhu et al., 2021). Therefore, no general method is available for its prediction so far. However, when simulating nitrogen uptake and losses on catchment-scale, information on the amount and timing of nitrogen fertilization is often more crucial, than knowledge of the initial nitrate content of the soil (Krevh et al., 2023). The mineral and relatively dynamic N pools are often considered to be initialized during the warmup period of the models (Yuan and Chiang, 2015). It is especially important to have a proper parameterization of the agricultural management (e.g. fertilization, residue management) setup in the model application with an appropriate length of the warmup period, where we recommend it to be no less than 4 years. Furthermore, it is beneficial to initialise the SOM levels accurately to define the large and rather slow pool of organic nitrogen (Liang et al., 2023).

## 3 Results and discussion

### 3.1 Bulk density

Table 6 shows the prediction performance of the selected PTFs. The performance varies depending on the texture classes, e.g., it is lower for clayey soils, sandy clay loams, and loams in the EU-HYDI dataset (Figure S1 a). For the LUCAS topsoil samples, the performance of all PTFs is lower compared to their performance on EU-HYDI in terms of RMSE. Additionally, all analysed methods tend to overpredict bulk density. The BD_Alexander_A PTF (E.q. 3) ranks highest based on the sample-number-weighted average results of the Kruskal-Wallis test, analysed on both the EU-HYDI and LUCAS dataset (Table 6, weighted rank). The BD_Alexander_A_Hossain PTF shows the performance of the combined use of the BD_Alexander_A (for soils with organic carbon content less than 12 %) and BD_Hossain (for soils with organic carbon content equal to or higher than 12 %) PTFs. This combined PTF performs similarly to the simple BD_Alexander_A method but helps to properly derive bulk density for soils with high organic matter content. Figure 2 shows the scatterplot of measured versus predicted bulk density values of the best performing PTF, where the predefined bulk density is capped at 1.72 g cm$^{-3}$ as product of the models constraints.

**Table 6.** Prediction performance of bulk density (g cm$^{-3}$) computed by available pedotransfer functions on the point data of EU-HYDI (N = 11,273) and LUCAS (N = 5821). ME: mean error, MAE: mean absolute error, RMSE: root mean squared error, NSE: Nash-Sutcliffe efficiency, $R^2$: coefficient of determination.

| PTF | EU-HYDI (N=11273) | | | | | | | LUCAS (N = 5821) | | | | | | | Weighted rank*** |
|---|---|---|---|---|---|---|---|---|---|---|---|---|---|---|---|
| | ME | MAE | RMSE | NSE | $R^2$ | Sign. diff.* | Rank** | ME | MAE | RMSE | NSE | $R^2$ | Sign. diff.* | Rank | |
| BD_Alexander_A | 0.01 | 0.15 | 0.19 | 0.22 | 0.27 | g | 1 | -0.22 | 0.26 | 0.32 | -0.01 | 0.49 | b | 6 | 2.70 |
| BD_Alexander_A_Hossain | 0.01 | 0.15 | 0.19 | 0.22 | 0.27 | g | 1 | -0.24 | 0.27 | 0.33 | -0.06 | 0.49 | b | 6 | 2.70 |
| BD_Alexander_B | 0.08 | 0.16 | 0.21 | 0.05 | 0.27 | e | 4 | -0.14 | 0.21 | 0.27 | 0.28 | 0.49 | e | 3 | 3.66 |
| BD_MAn_J_A | 0.07 | 0.16 | 0.21 | -0.04 | 0.23 | f | 3 | -0.10 | 0.27 | 0.44 | -0.90 | 0.39 | c | 5 | 3.68 |
| BD_MAn_J_B | 0.09 | 0.17 | 0.21 | -0.01 | 0.27 | d | 5 | -0.12 | 0.20 | 0.26 | 0.32 | 0.49 | f | 2 | 3.98 |
| BD_Rawls | 0.27 | 0.29 | 0.33 | -1.40 | 0.27 | a | 8 | -0.03 | 0.18 | 0.23 | 0.47 | 0.51 | g | 1 | 5.62 |
| BD_Bernoux | 0.20 | 0.23 | 0.28 | -0.72 | 0.22 | b | 7 | -0.15 | 0.24 | 0.30 | 0.13 | 0.35 | d | 4 | 5.98 |
| BD_Hollis | 0.04 | 0.20 | 0.25 | -0.45 | 0.10 | c | 6 | -0.26 | 0.28 | 0.34 | -0.17 | 0.47 | a | 8 | 6.68 |

\*Different letters indicate significant differences at the 0.05 level between the accuracy of the methods based on the squared error; for example, performance indicated with the letter c is significantly better than the one noted with letters b and a. \*\*Rank based on the Kruskal-Wallis test, 1 denotes the best performing method.\*\*\* Sample-number-weighted average results of the Kruskal-Wallis test.

If only the soil's organic carbon content is known, the prediction accuracy is restricted. The RMSE value of BD_Alexander_A_Hossain PTF on the EU-HYDI is comparable with the accuracy of an ML-based PTF built on a French dataset (Chen et al., 2018), when computed on independent validation sets, which reported RMSE between 0.17 and 0.22 g cm$^{-3}$. This performance is better than the results of a model transferability test of a PTF derived on soils from Campania, Italy, analysed on the EU-HYDI (Palladino et al., 2022), which had RMSE = 0.210 g cm$^{-3}$. Yi et al. (Xiangsheng et al., 2016) and De Souza et al.(Souza et al., 2016) found RMSE values higher than 0.185 g cm$^{-3}$ when they applied PTFs trained on temperate soils, available from the literature, on a Chinese permafrost region and Brazilian catchment, respectively. This outcome underscores the significance of refraining from using a PTF that was trained on soils formed under different conditions – i.e. with different soil forming factors – , making it inapplicable to the specific target area (Chen et al., 2018; Tranter et al., 2009).

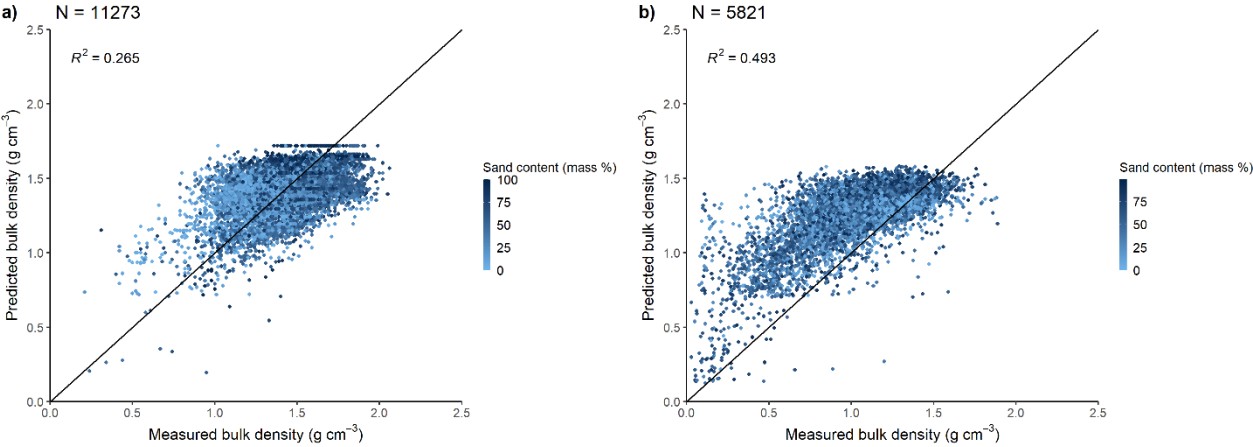

**Figure 2.** Scatterplot of measured versus predicted bulk density values of the best performing PTF (BD_Alexander_A_Hossain) analysed on the point data of EU-HYDI (a) and LUCAS (b) dataset.

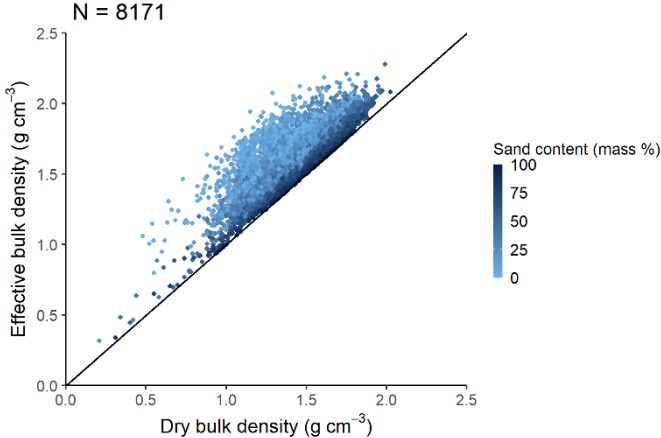

**Figure 3.** Scatterplot of dry versus effective bulk density analysed based on the point data of EU-HYDI.

Effective bulk density is always higher than dry bulk density. Effective bulk density value computed for the EU-HYDI dataset with Eq. (22) and (23) was between 0.32 and 2.17 g cm$^{-3}$. Figure 3 shows the scatterplot of dry bulk density versus computed
effective bulk density based on the EU-HYDI dataset.

Based on the performance analysis on EU-HYDI (N = 11,273) the prediction of dry bulk density could be performed with i) Eq. (2) (BD_Alexander_A) for soils with OC < 12% and ii) Eq. (8) (BD_Hossain) for soils with OC >= 12%.

**3.2 Porosity**

The porosity values computed based on the particle density predicted by Schjønning et al. PTF (POR_Schjonning_etal)
implemented in Eq. (10) were significantly more accurate on those EU-HYDI samples, which considered measured particle
density value for the computation of porosity (Table 7). If solely samples with low organic matter content, specifically less
than 1%, were considered for analysis, no notable differences between the methods were observed. In the case of soils with
organic matter content higher than 1 % the prediction of porosity significantly improved if particle density was computed
based on distinction between organic matter and mineral substrates. Figure 4 displays the scatterplot of measured versus Eq.
(10) (POR_Schjonning_etal) predicted porosity values.

**Table 7.** Prediction performance of porosity (vol %) computed by available pedotransfer functions on the point data of EU-
HYDI results are structured by organic matter content. OM: organic matter content (mass %), N: number of samples, ME:
mean error, MAE: mean absolute error, RMSE: root mean squared error, NSE: Nash-Sutcliffe efficiency, $R^2$: coefficient of
355 determination.

| Name of PTF | OM (mass %) | N | ME | MAE | RMSE | NSE | $R^2$ | Sign. diff.* |
|---|---|---|---|---|---|---|---|---|
| POR_Schjonning_etal | any | 2290 | 0.19 | 1.38 | 2.53 | 0.882 | 0.889 | c |
| POR_Schjonning_etal_recal | | 2290 | 1.05 | 1.81 | 2.84 | 0.852 | 0.878 | a |
| POR_2_65 | | 2290 | 0.23 | 1.67 | 2.71 | 0.866 | 0.883 | b |
| POR_Schjonning_etal | 0 =< OM < 10 | 2246 | 0.20 | 1.38 | 2.55 | 0.860 | 0.869 | c |
| POR_Schjonning_etal_recal | | 2246 | 1.06 | 1.81 | 2.86 | 0.824 | 0.855 | a |
| POR_2_65 | | 2246 | 0.29 | 1.64 | 2.70 | 0.843 | 0.861 | b |
| POR_Schjonning_etal | 0 =< OM < 5 | 1943 | 0.23 | 1.34 | 2.48 | 0.841 | 0.849 | c |
| POR_Schjonning_etal_recal | | 1943 | 1.01 | 1.76 | 2.78 | 0.801 | 0.834 | a |
| POR_2_65 | | 1943 | 0.52 | 1.57 | 2.61 | 0.824 | 0.840 | b |
| POR_Schjonning_etal | 0 =< OM < 1 | 492 | -0.22 | 1.32 | 1.84 | 0.879 | 0.881 | a |
| POR_Schjonning_etal_recal | | 492 | -0.01 | 1.25 | 1.69 | 0.898 | 0.898 | a |
| POR_2_65 | | 492 | 0.23 | 1.23 | 1.63 | 0.905 | 0.907 | a |
| POR_Schjonning_etal | 10 =< OM | 44 | -0.24 | 1.41 | 1.94 | 0.968 | 0.969 | b |
| POR_Schjonning_etal_recal | | 44 | 0.92 | 1.49 | 1.91 | 0.969 | 0.980 | b |
| POR_2_65 | | 44 | -2.85 | 2.86 | 3.29 | 0.909 | 0.977 | a |

*Different letters indicate significant differences at the 0.05 level between the accuracy of the methods based on the squared
error; for example, performance indicated with the letter c is significantly better than the one noted with letters b and a.

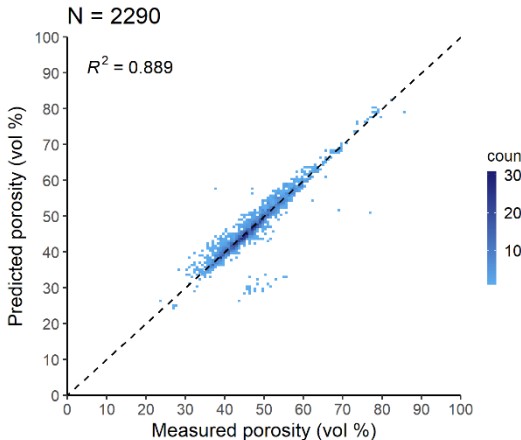

**Figure 4.** Scatterplot of measured versus predicted porosity values of the best performing PTF, POR_Schjonning_etal (Eq. 10) analysed based on the EU-HYDI subset with measured particle density values. Count: the number of cases in each quadrangle.

When data on porosity is missing, some studies use the saturated water content as its approximation, although based on the literature the saturated water content is usually equal or less than the total porosity (Lal and Shukla, 2004). Figure 5 shows the relationship between porosity and saturated water content for 391 EU-HYDI samples with measured values of both parameters. Among these samples, 56.5% have a total porosity larger or equal to the saturated water content. For the samples where the saturated water content is higher than the total porosity, the reason may be the uncertainties in the measurement of both parameters. It is possible that free water could have pounded on top of the sample when its saturated weight was measured, and errors in the measurement of particle density used to compute porosity may have also contributed (Kutílek and Nielsen, 1994; Nimmo, 2004), resulting in a lower porosity.

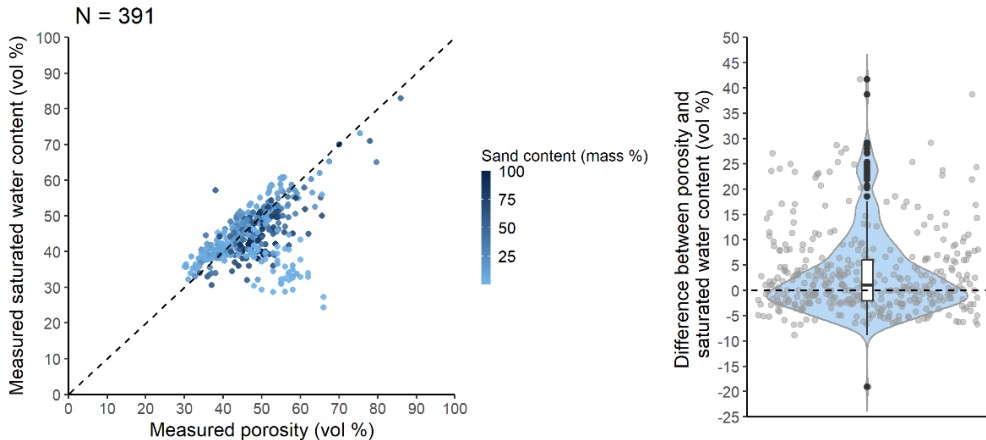

**Figure 5.** Scatterplot of measured porosity values versus measured saturated water content and boxplot of the difference between the two values tested on point data in EU-HYDI dataset.

Based on the study performed in EU_HYDI, prediction of porosity could be performed with the Schjønning et al. PTF Eq. (10) instead of defining particle density as 2.65 g cm$^{-3}$ in Eq. (12).

### 3.2 Albedo

The range of soil albedo computed with Eq. (13) for the topsoil layers of the EU-HYDI dataset with different moisture states (Table 8) is within the range of the values available from the literature, which is 0.10-0.43 in the case of ECOCLIMAP-U dataset (Carrer et al., 2014). The median dry, bare soil albedo and surface albedo values of year 2022 extracted from the MCD43A3 database to the EU-HYDI topsoil layers are significantly lower than the computed values (Figure 6). The histogram of the monthly surface albedo and dry, bare soil albedo values extracted to the EU-HYDI topsoil samples are show on Figure S2a and b. It's crucial to specify the moisture condition for which the albedo value is needed in the modelling process.

**Table 8.** Descriptive statistics of soil albedo values computed with the simplified Gascoin et al. (2009) equation on the topsoil samples of EU-HYDI dataset (N = 7,537) at different moisture states: based on saturation (ALB_comp_THS), field capacity (ALB_comp_FC), wilting point (ALB_comp_WP).

| Albedo at different moisture state | Minimum | Maximum | Range | Mean | Median | Standard deviation |
|---|---|---|---|---|---|---|
| ALB_comp_THS | 0.15 | 0.17 | 0.02 | 0.15 | 0.15 | 0.00 |
| ALB_comp_FC | 0.15 | 0.31 | 0.16 | 0.17 | 0.16 | 0.02 |
| ALB_comp_WP | 0.15 | 0.46 | 0.31 | 0.22 | 0.19 | 0.08 |

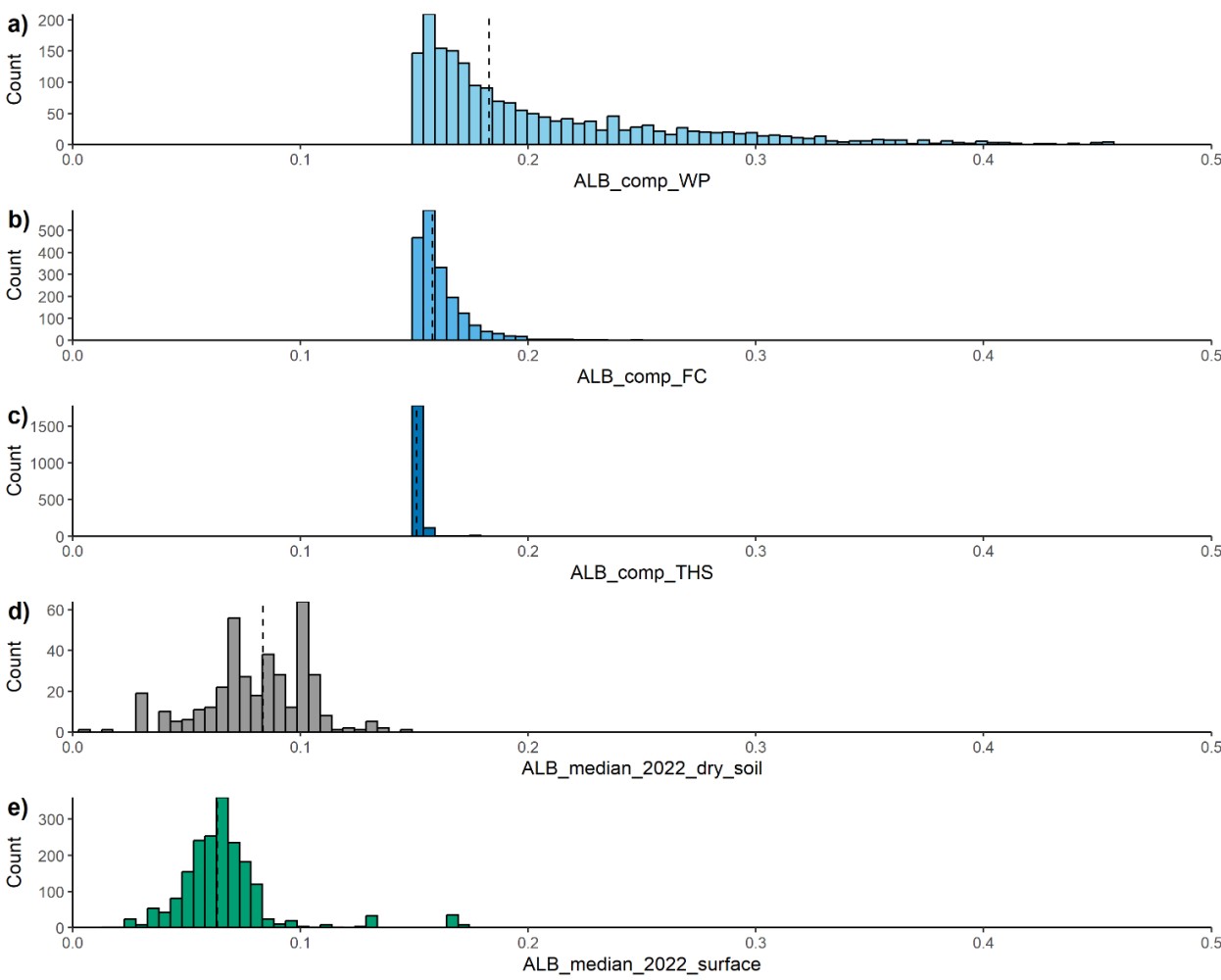

**Figure 6.** Histograms of the soil albedo computed with the Gascoin et al. (2009) equation for the topsoil layers of the EU-HYDI dataset in the case of three moisture states: at saturation (ALB_comp_THS) (a), internal drainage dynamics-based field capacity (ALB_comp_FC) (b) and wilting point (ALB_comp_WP) (c) (N = 2408), and median surface (d) and dry, bare soil albedo (e) of year 2022 (ALB_median_2022_dry_soil, ALB_median_2022_surface) extracted from the MCD43A3 global database for the EU-HYDI topsoil layers. Vertical dashed lines indicate the median values.

### 3.3 Soil erodibility factor

The soil erodibility factor (K-factor) computed on the topsoil samples of the EU-HYDI dataset with Eq. (14) are comparable with the values of the European 500 m resolution soil erodibility map published by Panagos et al. (Panagos et al., 2014) in terms of range, mean and density of the values (Table 9 and Figure 7), although the relationship between the computed and mapped values was weak (Figure 8). For the computation of the European map soil organic matter content, soil texture, coarse fragments content, soil structure and stoniness were considered. The Renard et al. (Eq. 15) equation resulted in a higher median

value but lower possible maximum value because the computed soil erodibility factor is capped at 0.044 $\left(\frac{t \cdot ha \cdot h}{ha \cdot MJ \cdot mm}\right)$ due to the constraints of the model. The relationship between the soil erodibility factors derived by different methods is strongest between the values computed using the Sharpley and Williams (1990) method and the Renard et al. (1997) method. This is logical because both methods consider the particle size distribution of the soil as input information.

Both approaches, whether directly applying the equations (Eq. 14 or 15) or extracting values, generate predicted soil erodibility values. While both can be used for environmental modelling, i) European soil erodibility map could be linked with LUCAS topsoil dataset and maps, ii) employing Eq. (14) or (15) might offer greater consistency with the other local basic and physical soil data, aligning more seamlessly with the modelling process. Given the scarcity of measured K-factor values, our suggestion is to initially utilize these predicted values as preliminary approximations. However, we recommend fine-tuning this factor during the model calibration process.

**Table 9.** Descriptive statistics of soil erodibility factor values computed with the Sharpley and Williams (1990) and Renard et al. (1997) equations on the topsoil samples of the EU-HYDI dataset (N = 11,287) provided in U.S. Customary Unit $\left(\frac{t \cdot arce \cdot h}{hundreds\ of\ acre \cdot foot - tonf \cdot inch}\right)$ and SI Unit $\left(\frac{t \cdot ha \cdot h}{ha \cdot MJ \cdot mm}\right)$.

| Method | USLE K factor in different units | | | | | | |
|---|---|---|---|---|---|---|---|
| | Unit | Min | Max | Range | Mean | Median | Standard deviation |
| Sharpley and Williams (1990) | $\left(\frac{t \cdot arce \cdot h}{hundreds\ of\ acre \cdot foot - tonf \cdot inch}\right)$ | 0.00 | 0.48 | 0.48 | 0.27 | 0.27 | 0.09 |
| | $\left(\frac{t \cdot ha \cdot h}{ha \cdot MJ \cdot mm}\right)$ | 0.000 | 0.063 | 0.063 | 0.036 | 0.035 | 0.012 |
| Renard et al. (1997) | $\left(\frac{t \cdot arce \cdot h}{hundreds\ of\ acre \cdot foot - tonf \cdot inch}\right)$ | 0.05 | 0.33 | 0.29 | 0.24 | 0.27 | 0.09 |
| | $\left(\frac{t \cdot ha \cdot h}{ha \cdot MJ \cdot mm}\right)$ | 0.006 | 0.044 | 0.038 | 0.032 | 0.035 | 0.012 |

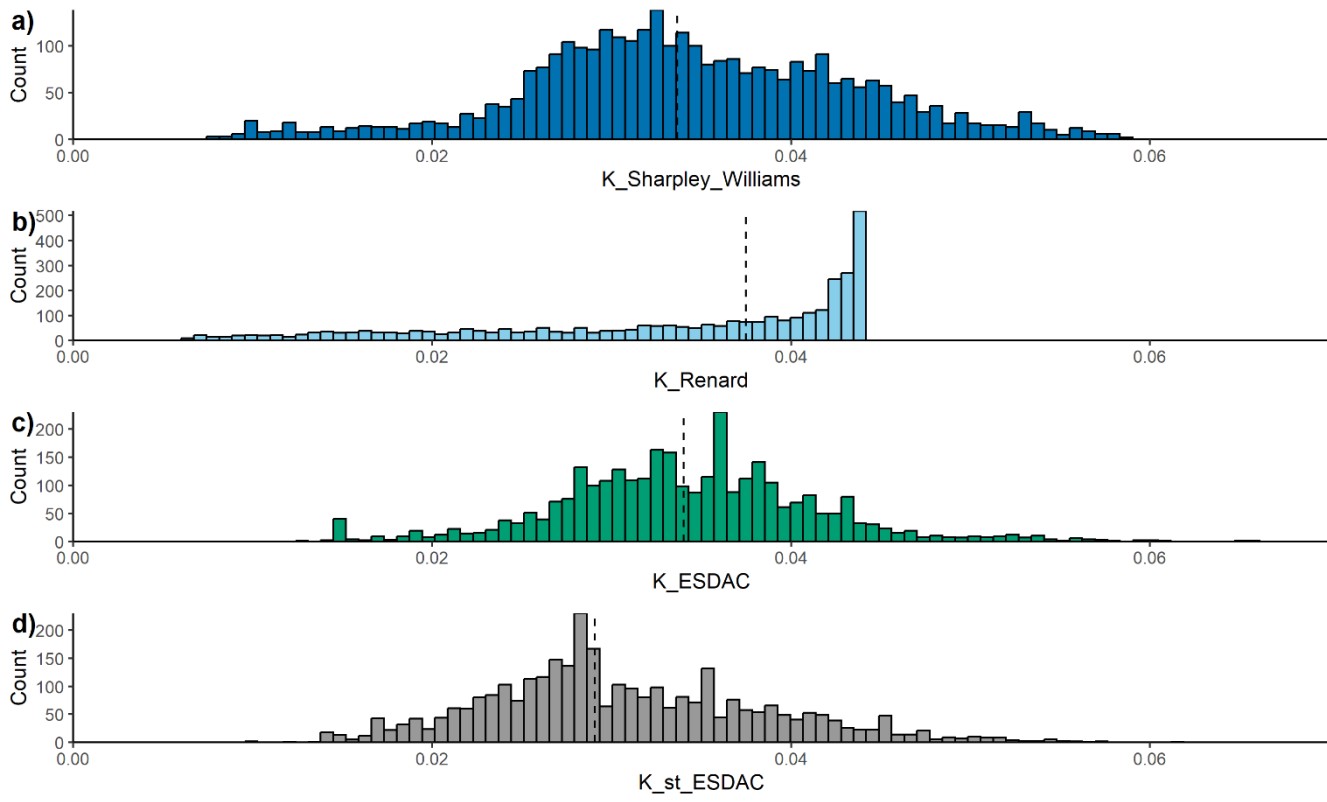

**Figure 7.** Histogram of the soil erodibility factor $\left(\frac{t \cdot ha \cdot h}{ha \cdot MJ \cdot mm}\right)$ computed with the Sharpley and Williams (1990) (K_Sharpley_Williams, N = 3276) (a) and Renard et al. (1997) (K_Renard, N = 3276) (b) equations on the topsoil samples of the EU-HYDI dataset, and extracted from the soil erodibility map of Europe for the EU-HYDI topsoil layers without (K_ESDAC, N = 3100) (c) and considering stoniness (K_st_ESDAC, N = 3190) (d). Vertical dashed lines indicate the median

values.

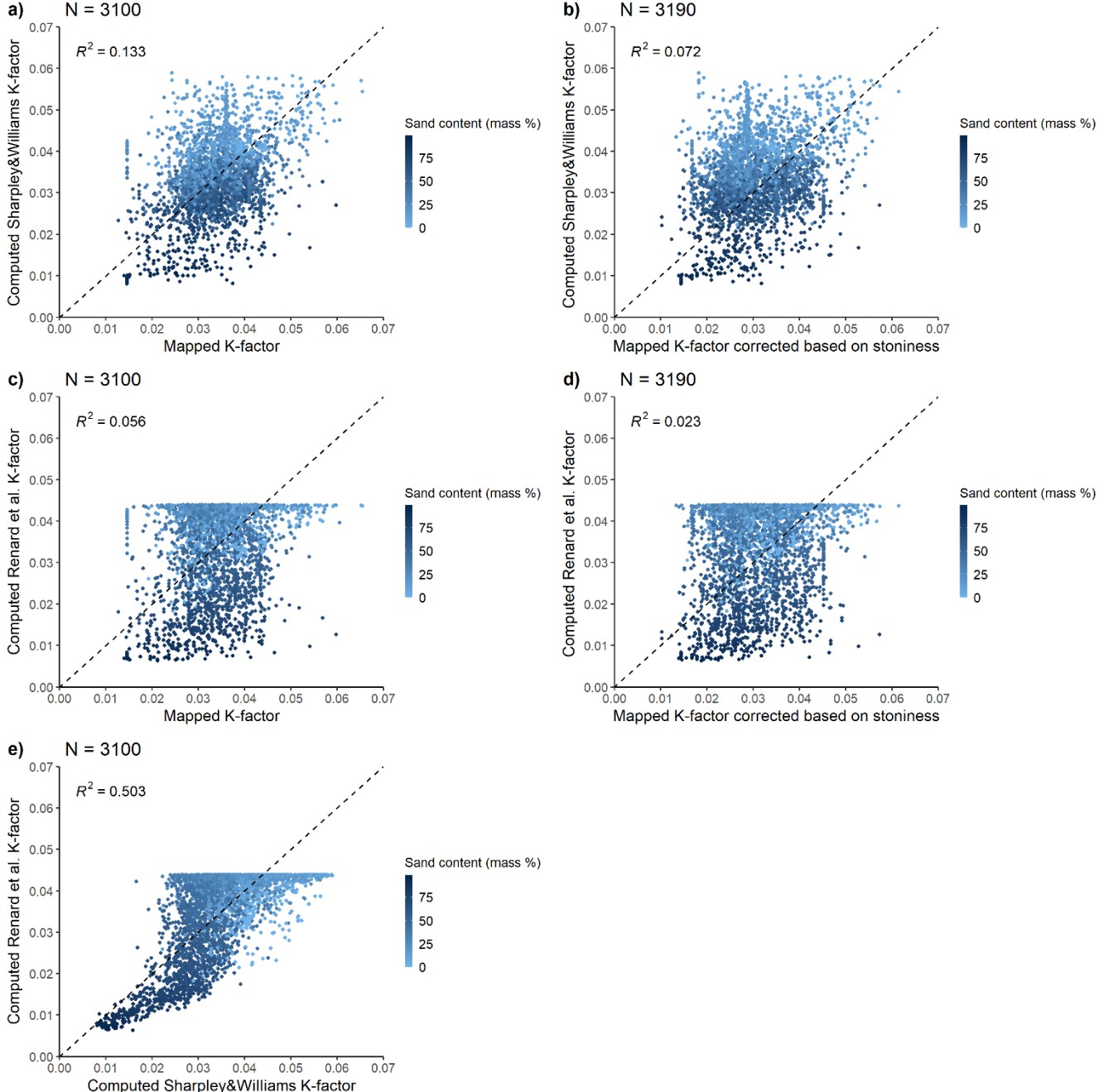

**Figure 8.** Scatterplot of computed soil erodibility factors versus extracted from the European soil erodibility factor map without (a, c) and with considering stoniness (b, d) based on the topsoil samples of the EU-HYDI dataset $\left(\frac{t \cdot ha \cdot h}{ha \cdot MJ \cdot mm}\right)$. Plot (e) shows the relationship between the values computed by the Sharpley and Williams (1990) and Renard et al. (1997) methods.

**3.4 Field capacity**

The FC defined (see abbreviations in Table 5) based on soil internal drainage dynamics (FC_VG_AO) differed from the field capacity measured at -100 cm, or -330 cm matric potential (FC_100 and FC_330 respectively) or computed from VG parameters at -100 cm, or -330 cm matric potential (FC_VG_100 and FC__VG_330 respectively) (Figure 9), as was expected. The scale of difference depends on i) the predefined soil matric potential value, which we consider using as measured field capacity, and ii) characteristics soil properties that influence soil hydraulic behaviour, such as soil texture, organic matter content, bulk density, clay mineralogy, structure, etc. Figures S3 and S4 show that for soils with low sand content (< 25 %) and high silt content (> 50 %) or low bulk density (< 0.7 g cm$^{-3}$) the FC_VG_AO is lower than water content measured at -100 cm or -330 cm matric potential (FC_VG_AO vs. FC_100 and FC_VG_AO vs. FC_330).

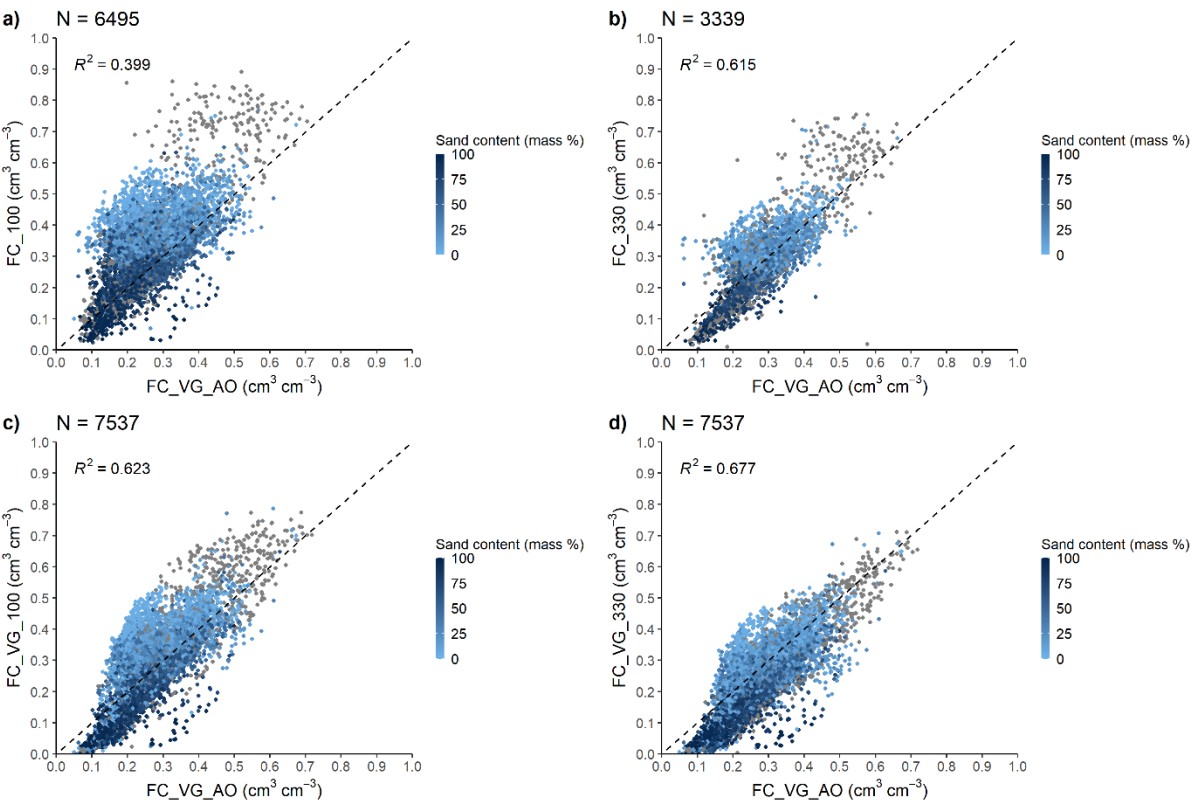

**Figure 9.** Scatterplot of internal drainage dynamics-based field capacity (FC_VG_AO) versus field capacity at -100 cm matric potential (a), at -330 cm matric potential (b), computed based on VG model with parameter h (head) set at -100 cm matric potential (c) and -330 cm matric potential (d).

If FC at a single matric potential value is computed from the fitted VG parameters (FC_VG_100, FC_VG_330) their Pearson correlation with the FC_VG_AO is higher than in the case of FC measured at -100 or -330 cm matric potential (Figure S5). This is logical because in the case of FC_VG_100 and FC_VG_300 the same VG parameters are used for the computation as

for FC_VG_AO. In the case of EU-HYDI the FC_VG_330 is the closest to the FC_VG_AO. The only exception are sands where FC measured at -330 cm matric potential has the highest correspondence with FC_VG_AO (Figure S6).

**Table 10.** Prediction performance of internal drainage dynamics-based field capacity (cm$^3$ cm$^{-3}$) computed by pedotransfer functions on the FC and VG test sets of the EU-HYDI dataset. N: number of samples, ME: mean error, MAE: mean absolute

error, RMSE: root mean squared error, NSE: Nash-Sutcliffe efficiency, R$^2$: coefficient of determination.

| Approach to predict FC* | N | ME | MAE | RMSE | NSE | R$^2$ |
|---|---|---|---|---|---|---|
| pred_FC_VG_AO | 1591 | 0.005 | 0.043 | 0.058 | 0.514 | 0.519 |
| pred_FC_100 | 1413 | -0.071 | 0.083 | 0.106 | -0.779 | 0.297 |
| pred_FC_330 | 782 | -0.010 | 0.047 | 0.061 | 0.210 | 0.395 |
| pred_FC_VG_100 | 1591 | -0.015 | 0.070 | 0.090 | -0.184 | 0.320 |
| pred_FC_VG_330 | 1591 | 0.045 | 0.073 | 0.091 | -0.198 | 0.339 |

*pred_FC_VG_AO: predicted internal drainage dynamics-based field capacity based on VG parameters predicted from basic soil properties; pred_FC_100, pred_FC_330: field capacity at -100 and -330 cm matric potential directly predicted from basic soil properties; pred_FC_VG_100, pred_FC_VG_330: field capacity at -100 and -330 cm matric potential based on VG parameters predicted from basic soil properties.


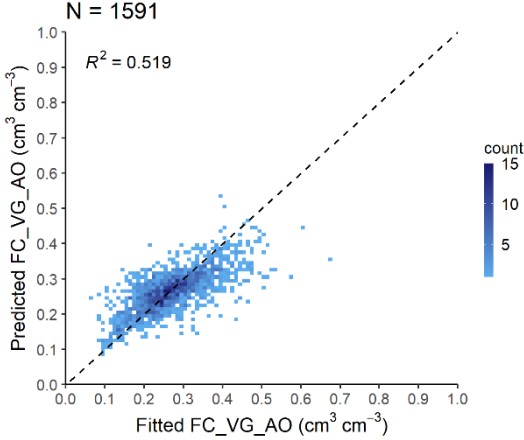

**Figure 10.** Scatterplot of internal drainage dynamics-based FC (FC_VG_AO) computed from fitted and predicted VG parameters analysed on the VG test set of the EU-HYDI dataset. Count: the number of cases in each quadrangle.

Table 10 illustrates the prediction performance of the FC_VG_AO for various approaches. If the FC_VG_AO was computed based on VG parameters predicted by the PTF07 of euptfv2, the RMSE value was 0.058 cm$^3$ cm$^{-3}$, which is comparable with

the literature values (Román Dobarco et al., 2019; Zhang and Schaap, 2017). Its correlation with the FC computed based on predicted VG parameters at -100 or -330 cm matric potential is weaker (with RMSE 0.090 and 0.091 $cm^3$ $cm^{-3}$), aligning with the results drawn from the FC computed from fitted VG parameters (Figure 9 c) and d)).

Figure 10 shows the scatterplot of FC_VG_AO computed from fitted and predicted VG parameters analysed only on those samples of the EU-HYDI which were not used for training of the VG PTF07. Performance of VG PTF07 was published in Szabó et al. (2021) with 0.054 $cm^3$ $cm^{-3}$ RMSE on the test set.

Thus FC_VG_AO could be used as FC and computed with Eq. (17) based on VG parameters predicted with i) euptfv2 (Szabó et al., 2021) for mineral soils and ii) euptfv1 (Tóth et al., 2015) class PTF (PTF18) for organic soils.

**3.5 Wilting point**

Calculating WP (see abbreviations in Table 5) from predicted VG parameters yields greater accuracy compared to using the equation provided by SWAT+ model (Figure 11, Table 11). Predicting WP directly from soil properties instead of deriving it from predicted VG parameters tends to yield greater accuracy (Børgesen and Schaap, 2005; Szabó et al., 2021; Tomasella et al., 2003) (Table 12). When multiple soil hydraulic parameters are needed, deriving all from a model encompassing the entire

matric potential range secures the physical relationship between them (Weber et al., 2023).

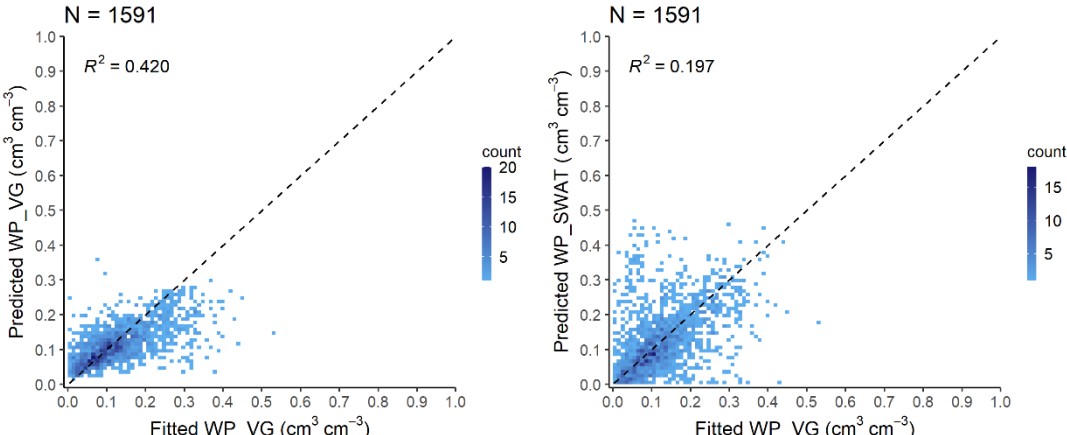

**Figure 11.** Scatterplot of wilting point computed from fitted VG parameters (Fitted WP_VG) versus a) wilting point computed from VG parameters predicted with euptfv2 (Predicted WP_VG) and b) wilting point predicted with the SWAT+ approach

(Predicted WP_SWAT), analysed on the VG test set of the EU-HYDI dataset. Count: the number of cases in each quadrangle.

**Table 11.** Prediction performance of wilting point (cm³ cm⁻³) derived with the VG model, computed by pedotransfer functions on the VG test set of the EU-HYDI dataset. Observed variable is the WP value computed based on the fitted parameters of the VG model. N: number of samples, ME: mean error, MAE: mean absolute error, RMSE: root mean squared error, NSE: Nash-Sutcliffe efficiency, $R^2$: coefficient of determination.

| Approach to predict WP* | N | ME | MAE | RMSE | NSE | $R^2$ |
|---|---|---|---|---|---|---|
| pred_WP_VG | 1591 | 0.016 | 0.045 | 0.065 | 0.382 | 0.420 |
| pred_WP_SWAT | 1591 | -0.001 | 0.062 | 0.093 | -0.239 | 0.197 |

*pred_WP_VG: wilting point computed based on VG parameters predicted from basic soil properties; pred_WP_SWAT: wilting point predicted with the equation built in the SWAT model.

**Table 12.** Prediction performance of wilting point (cm³ cm⁻³) computed by pedotransfer functions on the WP test set of the EU-HYDI dataset. Observed variable is the measured WP value. N: number of samples, ME: mean error, MAE: mean absolute error, RMSE: root mean squared error, NSE: Nash-Sutcliffe efficiency, $R^2$: coefficient of determination.

| Approach to predict WP* | N | ME | MAE | RMSE | NSE | $R^2$ |
|---|---|---|---|---|---|---|
| pred_WP_VG | 2088 | 0.052 | 0.060 | 0.087 | 0.105 | 0.431 |
| pred_WP_SWAT | 2088 | 0.028 | 0.046 | 0.066 | 0.490 | 0.630 |
| pred_WP | 2088 | 0.000 | 0.033 | 0.046 | 0.755 | 0.755 |

*pred_WP_VG: wilting point computed based on VG parameters predicted from basic soil properties; pred_WP_SWAT: wilting point predicted with the equation built in the SWAT model; pred_WP: wilting point directly predicted from basic soil properties.

WP could be computed with Eq. (18) based on VG parameters predicted with i) euptfv2 (Szabó et al., 2021) for mineral soils and ii) euptfv1 (Tóth et al., 2015) class PTF (PTF18) for organic soils.

### 3.6 Available water capacity

If only AWC (see abbreviations in Table 5) is required as input for a model, i.e., without FC and WP, a feasible option could involve direct prediction using a PTF like euptfv2. However, its estimation is more accurate if the internal drainage dynamics-based FC is considered for its computation (Gupta et al., 2023). Figure 12 and S9 show that coefficient of determination is low between the internal drainage dynamics-based AWC (AWC_VG_AO) and AWC based on FC at fixed matric potential (AWC_100, AWC_300, AWC_VG_100, AWC_VG_330). Which approach is the closest to the AWC_VG_AO varies based on texture classes (Figure S10).

The available water capacity based on field capacity measured at -100 cm head (AWC_100) is higher than the AWC_VG_AO, especially in the case of low sand content (< 25 %) and high silt content (> 50 %) (Figure 12c and S7). The available water capacity based on field capacity measured at -330 cm head (AWC_330) is higher than AWC_AO_VG when sand content is low (< 25 %) and silt content is high (> 50 %) and lower than AWC_AO_VG when sand content is higher than 25 % and silt content is less than 50 % (Figure 12d and S8).

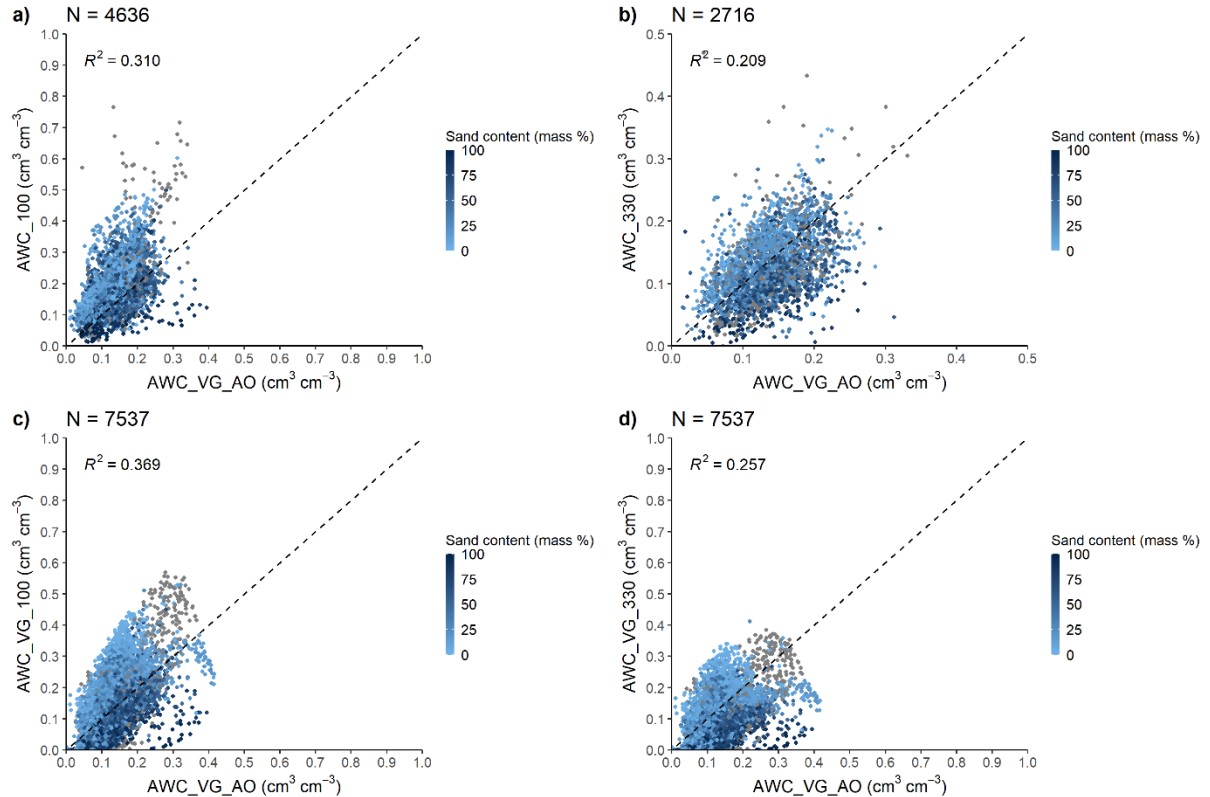

**Figure 12.** Scatterplot of available water capacity computed from internal drainage dynamics-based field capacity and wilting point derived based on VG parameters predicted from basic soil properties (AWC_VG_AO) versus (a, b) available water capacity computed from measured field capacity at -100 and -330 cm matric potential and wilting point, (c, d) available water capacity computed from field capacity at -100 and -330 cm matric potential and wilting point based on VG parameters predicted from basic soil properties.

**Table 13.** Prediction performance of available water capacity ($cm^3$ $cm^{-3}$) computed by pedotransfer functions on the VG test set of the EU-HYDI dataset. N: number of samples, ME: mean error, MAE: mean absolute error, RMSE: root mean squared error, NSE: Nash-Sutcliffe efficiency, $R^2$: coefficient of determination.

| Approach to predict AWC* | N | ME | MAE | RMSE | NSE | $R^2$ |
|---|---|---|---|---|---|---|
| pred_AWC_VG_AO | 1591 | -0.011 | 0.034 | 0.048 | 0.339 | 0.372 |
| pred_AWC_VG_100 | 1591 | -0.031 | 0.071 | 0.090 | -1.325 | 0.072 |
| pred_AWC_VG_330 | 1591 | 0.029 | 0.061 | 0.078 | -0.725 | 0.044 |

*pred_AWC_VG_AO: available water capacity computed from internal drainage dynamics-based field capacity and wilting point derived based on VG parameters predicted from basic soil properties; pred_AWC_VG_100, pred_AWC_VG_330: available water capacity computed from field capacity at -100 and -330 cm matric potential and wilting point based on VG parameters predicted from basic soil properties.

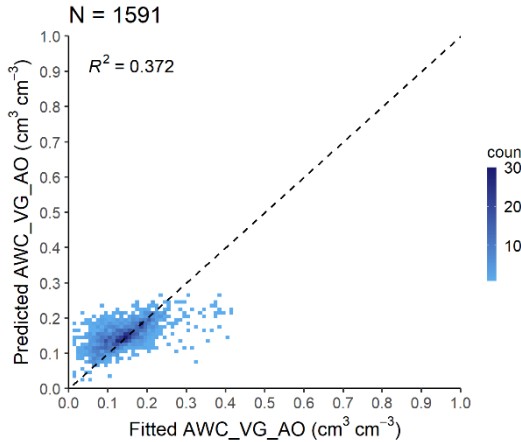

**Figure 13.** Scatterplot of internal drainage dynamics-based AWC (AWC_VG_AO) computed from fitted and predicted VG parameters analysed on the VG test set of the EU-HYDI dataset. Count: the number of cases in each quadrangle.

Table 13 shows the prediction performance of internal drainage dynamics-based AWC (AWC_VG_AO). As expected, the predicted internal drainage dynamics-based AWC had the lowest RMSE and highest $R^2$ value. The AWC computed based on the FC at 100 cm matric head derived with the predicted VG parameters (pred_AWC_VG_100) had the lowest performance. This approach yielded over-prediction of the AWC_VG_AO values when AWC_VG_AO is lower than 0.10 cm$^3$ cm$^{-3}$ and under-prediction when AWC_VG_AO is higher than 0.25 cm$^3$ cm$^{-3}$ (Figure 13).

Based on the findings, we recommend to compute the AWC based on the internal drainage dynamics-based FC (FC_VG_AO) and VG parameters-based WP (WP_VG) in Eq. (19).

## 3.7 Saturated hydraulic conductivity

Figure 14 shows the relationship between measured KS and computed with Eq. (20) based on the fitted VG parameters (KS_VG) (see abbreviation in Table 5). The coefficient of determination between the measured and computed values is low, however fitted (not predicted) VG parameters were used for the computation. Prediction performance of KS_VG is comparable with the published widely used PTFs (Nasta et al., 2021) (Figure 15, Table 14).

Prediction of saturated hydraulic conductivity (KS) has the highest uncertainty among the soil hydraulic properties. This uncertainty originates from the differences in the measurement methods applied to measure KS, in terms of sampling volume, sample dimensions, difference between in-situ and laboratory methods (Ghanbarian et al., 2017). Due to the uncertainty of the measurements, uncertainty of the prediction is minimum one order of magnitude during the application of a PTF (Nasta et al., 2021). Estimation of KS by traditional PTFs that use basic soil properties as input is rather limited, because KS of a sample is largely determined by its structural properties and pore network characteristics, of which we lack quantitative descriptors and data (Lilly et al., 2008). There is also at least one order of magnitude difference between replicated measurements on samples

coming from the same soil horizon due to the extreme spatial variability of this particular soil property. Hence, it's important to note that while we might improve individual sample predictions for KS, the representativeness of these samples within their specific fields remains constrained. We suggest initializing this soil property using the VG parameters with Eq. (20), but keeping in mind that it should be adjusted during model calibration as a variable.

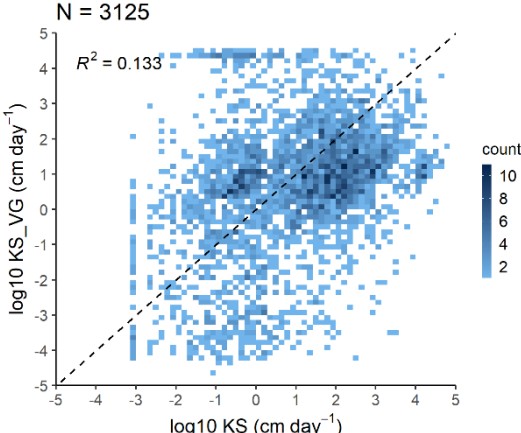

**Figure 14.** Scatterplot of measured saturated hydraulic conductivity (KS) versus saturated hydraulic conductivity computed from fitted VG parameters (KS_VG).


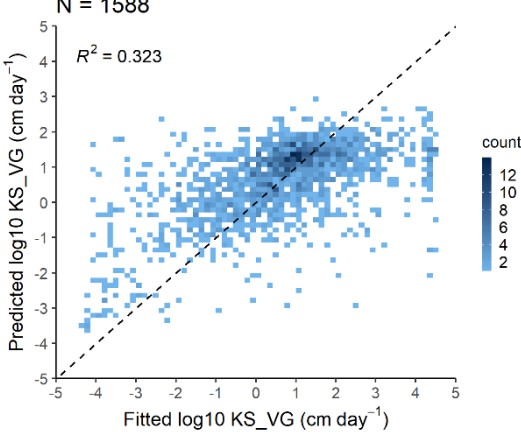

**Figure 15.** Scatterplot of saturated hydraulic conductivity computed from fitted and predicted VG parameters (KS_VG) analysed on the VG test set of the EU-HYDI dataset. Count: the number of cases in each quadrangle.

**Table 14.** Prediction performance of saturated hydraulic conductivity (cm day$^{-1}$) computed by pedotransfer function on the VG test set of the EU-HYDI dataset. N: number of samples, ME: mean error, MAE: mean absolute error, RMSE: root mean squared error, NSE: Nash-Sutcliffe efficiency, R$^2$: coefficient of determination.

| Approach to predict KS* | N | ME | MAE | RMSE | NSE | R$^2$ |
|---|---|---|---|---|---|---|
| log10pred_KS_VG | 1591 | -0.06 | 1.07 | 1.48 | 0.303 | 0.307 |

*log10pred_KS_VG: logarithmic 10 based saturated hydraulic conductivity computed based on VG parameters predicted from basic soil properties.

**3.8 Phosphorus content of the topsoil**

Figure 16 shows the European P map (Ballabio et al., 2019) clipped for the area of the Felső-Válicka study site (A) and the P map created with the mean statistics-based method using the local land use map (B) and the map of the hydrological response units (HRU) defined in the SWAT+ model (C). The spatial pattern of the two phosphorus maps is similar, but the map created with our proposed method has a higher resolution and follows the polygons of the HRU map.


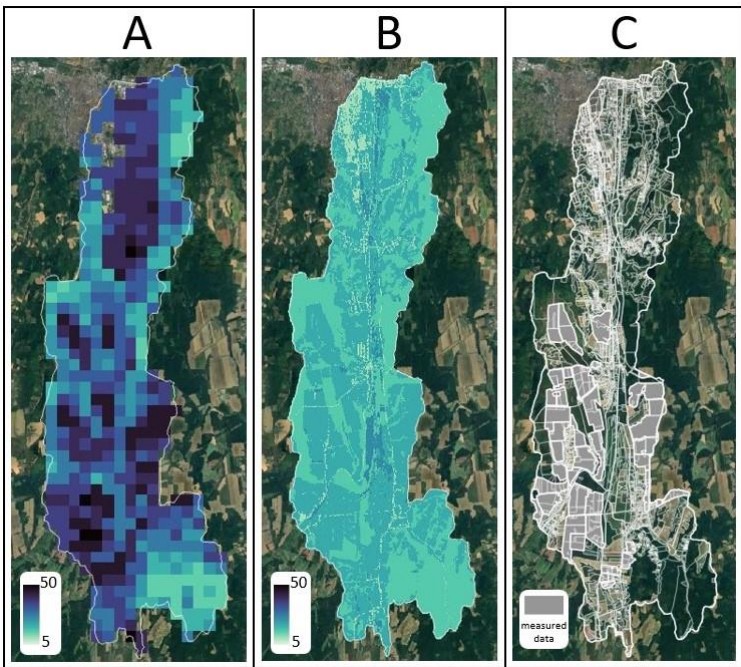

**Figure 16.** European topsoil P content map (Ballabio et al., 2019) (A), region-specific mean statistics-based P content map (B), hydrological response units with indication of agricultural parcels with measured P values (C) in the Felső-Válicka case study.


Figure 17 shows the geometric mean P values of the HRUs by land use categories of the European soil P map and the region-specific mean statistics-based P map in the area of Felső-Válicka. Comparing the results of the geometric mean P values, we can see that the European topsoil P map on average has a higher P concentration, with no significant differences observed between the land use categories. Based on the region specific LUCAS Topsoil dataset, artificial land use areas (urban fabric and industrial, commercial and transport units), forests and pastures are expected to have lower P concentration values. The mean statistics-based P map is more suitable at identifying differences resulting from local land use variation in the analysed case study. The P monitoring data measured on the 34 agricultural parcels, classified as arable land shows that the geometric mean of Olsen P in the area is 24 mg kg$^{-1}$, which is slightly higher than predicted by the mean statistics-based method (19.78 mg kg$^{-1}$).

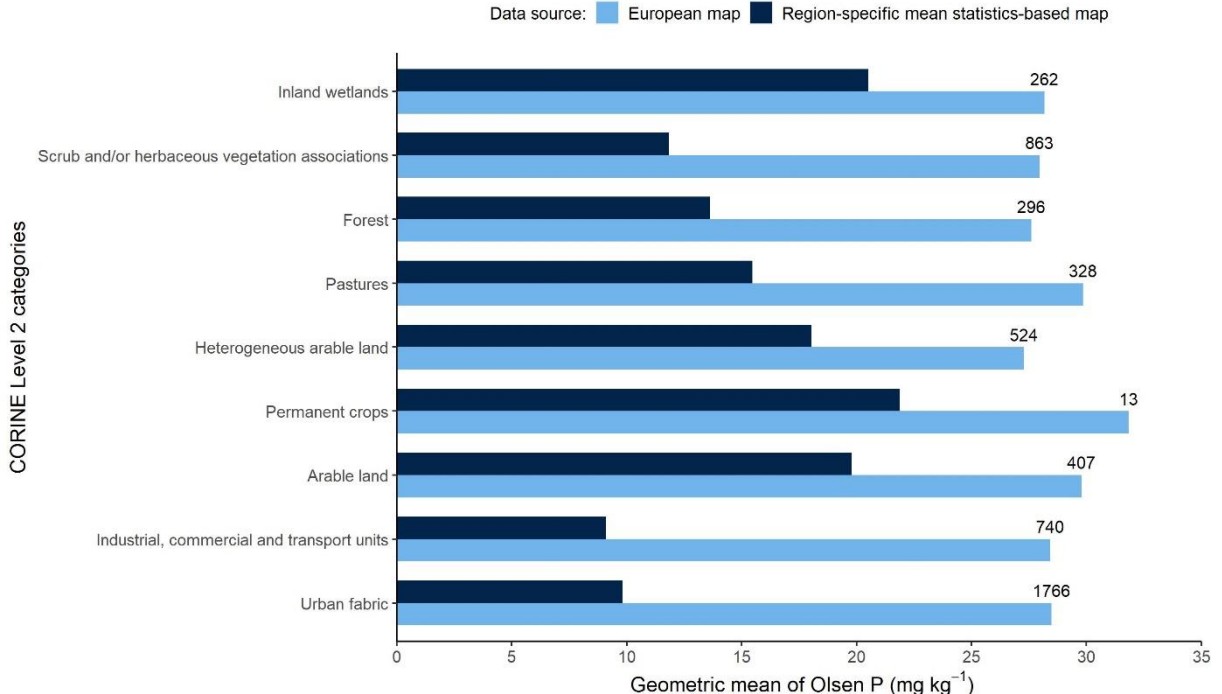

**Figure 17.** Geometric mean values of Olsen P across CORINE Level 2 land cover categories in the Felső-Válicka case study for both the European topsoil P content map and the region-specific mean statistics-based P content map with number of samples by categories indicated.

Ballabio et al. (2019) found that land use was the most important predictor for computing the topsoil phosphorus content map for Europe. This underscores that a soil P content map derived based on a local, fine-resolution, field-boundary-based land use map can provide more accurate results than one based on continental land use maps.

For regional or local studies, it is more plausible to use a local land use map and compute the geometric mean soil P values by
land use categories based on the LUCAS Topsoil dataset, which is relevant for the target area from a fertilization point of view. Where available, it is recommended to use measured data to overwrite the geometric mean values, creating a multi-data source solution that reflects the spatial pattern of nutrient content within arable land areas. For continental-scale studies, the European topsoil P map (Ballabio et al., 2019) could be used.

### 3.9 Suggested workflow to derive soil input parameters

Based on the above results, we describe the most efficient workflow to retrieve the soil input parameters for European environmental modelling.

Initially, the data source of the most relevant soil basic properties, such as soil layering, rooting depth, organic carbon content, clay, silt, and sand content, must be selected. Local data can describe the spatial variability of soil properties the best. Even if only soil basic properties are available locally, this data source could have priority against the more inclusive continental or
global datasets, i.e. containing information on both soil physical, chemical, and hydraulic properties, because local datasets aim to capture the area-specific variability of soil properties as accurately as possible. If no local or national soil basic data is available with the resolution required to study a target environmental process, possible input source for soil profile data or 3D soil dataset can be found in Table 1.

Different countries and institutions measure sand, silt, and clay content using different ISO protocols and methods by
recognizing different cutoff limits and classification standards. It is important to check which particle size limits are required by the environmental model. As an example, in the widely used SWAT/SWAT+ model, the sand, silt and clay content are assumed to be classified according to the USDA system, which defines particle size limit < 0.002 mm for clay, 0.002-0.05 mm for silt and 0.05-2 mm for sand fraction. When conversions between different classifications are required to bring the local dataset to the appropriate format, it is advised to apply the k-nearest neighbour interpolation (formerly called: 'similarity
technique'), which results in less uncertainty, smaller bias and shrinkage of resulting texture range compared to the simpler loglinear interpolation (Nemes et al., 1999).

In other cases, such as soil organic material, it is important to distinguish if soil organic carbon or soil organic matter is required by the model, and which of the two is available from the data source. The following most frequently used equation describes the relationship between those:

$OM = OC \cdot 1.724$ (21)

where OM is the organic matter content (mass %) and OC is organic carbon content (mass %). The 1.724 conversion factor was defined by Van Bemmelen (1890), but can vary between 1.4 and 2.5 depending on the method used to measure organic carbon content, composition of organic matter, degree of decomposition and clay content (Minasny et al., 2020; Pribyl, 2010). Pribyl (2010) recommends using the value 2 as a general conversion factor if no specific value is available.


When specifying bulk density, it is important to clarify whether the dry or effective value is required. If a measured value of neither is available, the dry bulk density can be computed from organic carbon content and particle size distribution. Further predictors, such as taxonomical information, soil structure, soil management parameters, environmental covariates are important as well (Hollis et al., 2012; Ramcharan et al., 2017) and can significantly improve the prediction performance.

However, PTFs including these variables are not always possible to apply to a data scarce region.

If effective bulk density is required, it can be derived from the dry bulk density with the method of Wessolek et al. (2009):

– for soils with organic carbon content higher than 0.58 mass %:

$$BD_{eff} = BD_{dry} + 0.009 \cdot clay \tag{22}$$

– for soils with organic carbon content less than or equal to 0.58 mass %:

$$BD_{eff} = BD_{dry} + 0.005 \cdot clay + 0.001 \cdot silt \tag{23}$$

where $BD_{eff}$ (g cm$^{-3}$) is effective bulk density, $BD_{dry}$ (g cm$^{-3}$) is the dry bulk density, clay is clay content (< 0.002 mm, mass %), silt is silt content (0.002-0.063 mm, mass %). It is important to note that Eq. (23) requires the silt content with 0.002-0.063 mm limit. It can be predicted from the clay (< 0.002 mm), silt (0.002 – 0.05 mm) and sand (0.05 – 2 mm) content with the TT.text.trans function of the soiltexture R package (Moeys, 2018). This method meets the accuracy required for computing

$BD_{eff}$, however, for other applications the transformation methods discussed by Nemes et. al. (1999) should be considered.

The hydrologic soil groups (HSG) are based on the infiltration characteristic of the soil and include four groups having similar runoff potential. The groups are defined based on the saturated hydraulic conductivity, depth to high water table and depth to water impermeable layer (Table 15). More details can be found in U.S. Department of Agriculture Natural Resources Conservation Service (2009).

For modelling purposes, it is important if tile drainage is present in the modelled area, because tile drainage systems influence the soil infiltration rate and runoff potential. Derivation of HSG requires local input data. If local datasets are not available, and SoilGrids 2017 (Hengl et al., 2017) was chosen as the source for the basic soil data, HSG can be retrieved from the global HYSOGs250m (Ross et al., 2018) database.

**Table 15.** Definition of soil hydrologic groups based on U.S. Department of Agriculture Natural Resources Conservation Service (2009). KS: saturated hydraulic conductivity ($\mu m\ s^{-1}$).

| Depth to water impermeable layer* | Depth to high water table** | KS of least transmissive layer in depth range ($\mu m\ s^{-1}$) | KS depth range | HSG*** |
|---|---|---|---|---|
| <50 cm | — | — | — | D |
| 50 to 100 cm | <60 cm | >40.0 | 0 to 60 cm | A/D |
| | | >10.0 to ≤40.0 | 0 to 60 cm | B/D |
| | | >1.0 to ≤10.0 | 0 to 60 cm | C/D |
| | | ≤1.0 | 0 to 60 cm | D |
| | ≥60 cm | >40.0 | 0 to 50 cm | A |
| | | >10.0 to ≤40.0 | 0 to 50 cm | B |
| | | >1.0 to ≤10.0 | 0 to 50 cm | C |
| | | ≤1.0 | 0 to 50 cm | D |
| >100 cm | <60 cm | >10.0 | 0 to 100 cm | A/D |
| | | >4.0 to ≤10.0 | 0 to 100 cm | B/D |
| | | >0.40 to ≤4.0 | 0 to 100 cm | C/D |
| | | ≤0.40 | 0 to 100 cm | D |
| | 60 to 100 cm | >40.0 | 0 to 50 cm | A |
| | | >10.0 to ≤40.0 | 0 to 50 cm | B |
| | | >1.0 to ≤10.0 | 0 to 50 cm | C |
| | | ≤1.0 | 0 to 50 cm | D |
| | >100 cm | >10.0 | 0 to 100 cm | A |
| | | >4.0 to ≤ 10.0 | 0 to 100 cm | B |
| | | >0.40 to ≤4.0 | 0 to 100 cm | C |
| | | ≤0.40 | 0 to 100 cm | D |

*An impermeable layer has a KS less than 0.01 $\mu m\ s^{-1}$ [0.0014 in $h^{-1}$] or a component restriction of fragipan; duripan; petrocalcic; orstein; petrogypsic; cemented horizon; densic material; placic; bedrock, paralithic; bedrock, lithic; bedrock, densic; or permafrost. **High water table during any month during the year. ***Dual HSG classes are applied only for wet

soils (water table less than 60 cm [24 in]). If these soils can be drained, a less restrictive HSG can be assigned, depending on the KS.

Figures 18-22 summarize the workflows to derive soil physical, hydraulic, and chemical parameters covered in this study. The workflows highlight the target soil property, necessary input, computation approach with suggested order of the computations.

Indirect initialization of soil mineral N is recommended via proper management data and model warm-up period. It is important to highlight that prediction approaches trained on local data are expected to be more accurate; therefore, those could replace the indicated methods where possible.

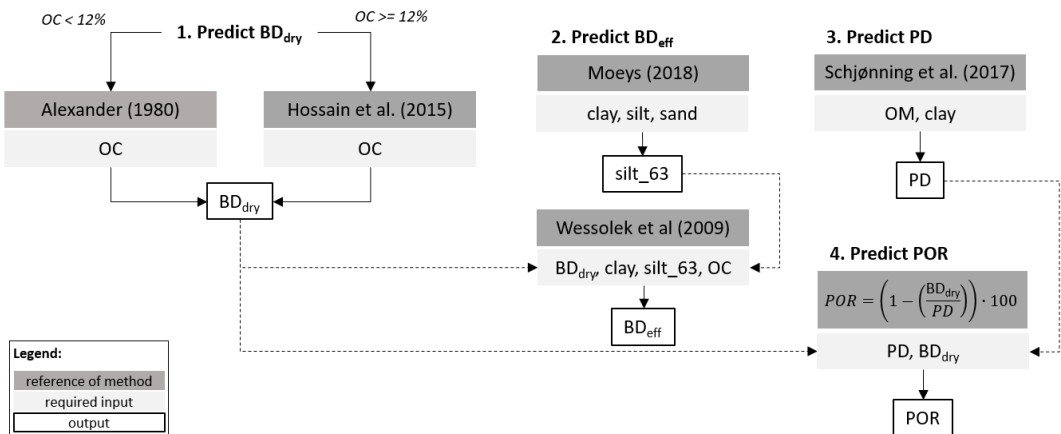

**Figure 18.** Prediction of soil physical properties. $BD_{dry}$: dry bulk density; clay: clay content (0-0.002 mm); silt: silt content (0.002-0.05 mm); sand: sand content (0.05-2 mm); silt_63: silt content (0.002-0.063 mm) OC: organic carbon content; $BD_{eff}$: effective bulk density; PD: particle density; POR: porosity.

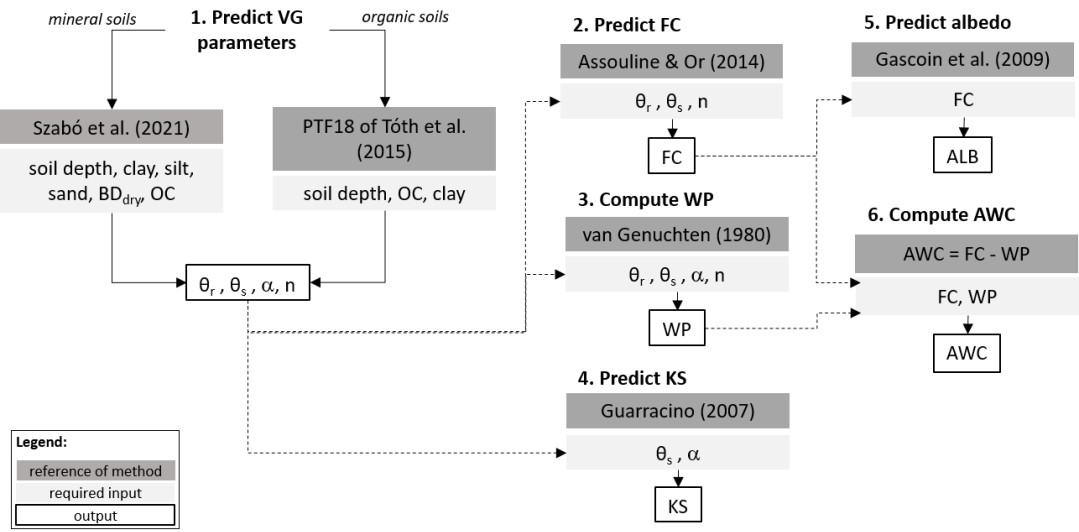

**Figure 19.** Prediction of soil hydraulic properties and moist soil albedo. Soil depth: mean soil depth of the soil sample; clay: clay content (0-0.002 mm); silt: silt content (0.002-0.05 mm); sand: sand content (0.05-2 mm); $BD_{dry}$: dry bulk density; OC: organic carbon content; $\theta_r$ : residual water content; $\theta_s$: saturated soil water content; $\alpha$ : scale parameter; n: shape parameter; FC: water content at field capacity; WP: water content at wilting point; KS: saturated hydraulic conductivity; AWC: available water capacity; ALB: soil albedo.


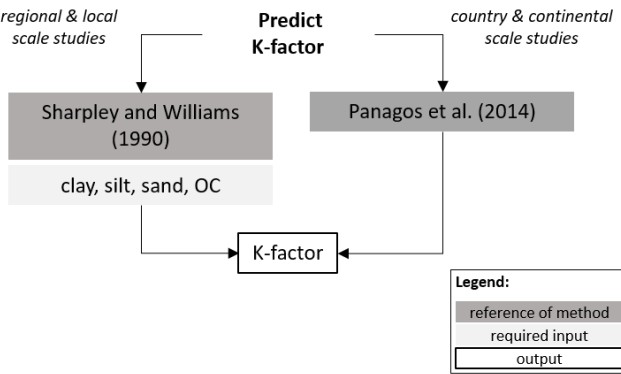

**Figure 20.** Prediction of soil erodibility factor (K-factor). clay content (0-0.002 mm); silt: silt content (0.002-0.05 mm); sand: sand content (0.05-2 mm); OC: organic carbon content.


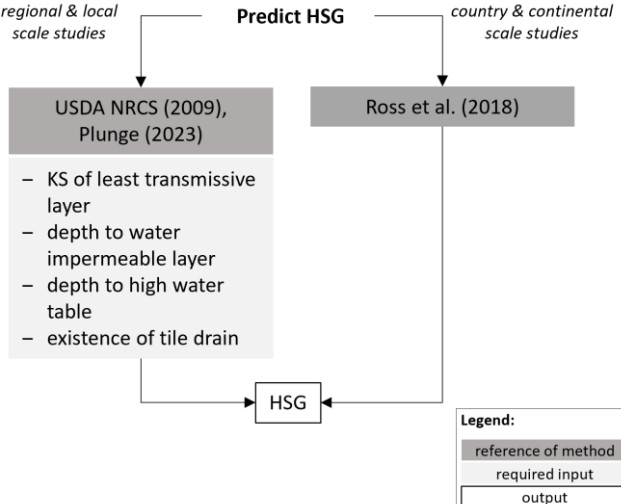

**Figure 21.** Prediction of hydraulic soil groups (HSG). KS: saturated hydraulic conductivity.

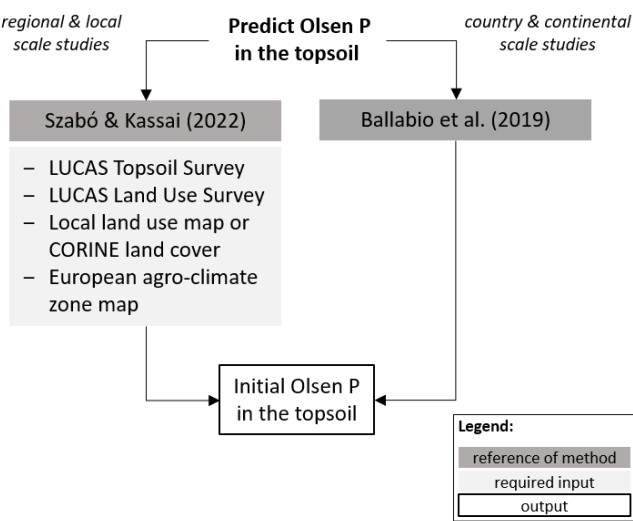


**Figure 22.** Prediction of Olsen phosphorus (P) content of the topsoil.

**4 Conclusions**

This study presents particular techniques and resources for extracting region-specific soil characteristics from national and global databases. While these databases might contain segments of soil information, they often lack comprehensive data
required by various environmental models, such as the SWAT+ model. Through evaluation and recommendation of selected PTFs, as well as the provision of compiled R scripts for estimation solutions addressing soil data gaps, the study aims to streamline input data preparation procedures for soil physical, hydraulic, and chemical properties in environmental modelling. Local data tend to retain finer soil details, hence it is recommended that users prioritise the utilisation of local (national) soil databases when it is deemed representative and reliable. Even if these databases only offer basic soil properties, they should
take precedence over broader continental or global datasets. The study demonstrated that missing soil properties could be estimated effectively from a basic set of soil parameters using appropriate PTFs developed for specific pedoclimatic regions, ensuring consistency in computed properties.

We prepared a set of workflows to derive soil input parameters for usage in various modelling studies. In cases where this approach is not viable, we offer comprehensive guidance on alternative soil databases, outlining strategies to derive the absent
soil properties effectively.

When using any available soil dataset, it is important to check the detailed description (metadata) of the dataset to avoid misinterpretation and errors in the models. Considerations such as consistent size limits for clay, silt, and sand content classification as per the model's requirements, distinction between organic carbon and organic matter, the need for dry or moist bulk density, and similar details are vital. Understanding whether the model derives soil properties that already exist in the
dataset is essential, aiding in selecting the most precise parameters for the model's application.

When retrieving or deriving missing soil input data, it is crucial to consider: i) which dataset and prediction approaches can offer physically plausible soil input data, and ii) the uncertainty associated with the derived soil input data for their appropriate use in the environmental model. For computing physically consistent soil hydraulic property values, namely FC, WP, AWC, and KS, it is plausible to use parameters of a model that describes soil water retention across the entire matric potential range. The parameters of the VG model have been widely employed to derive water retention at specific matric potential values or KS, hence can be used to derive physically plausible soil hydraulic properties. The static definition of FC could be replaced with a dynamic one that considers a soil-specific matric potential at which the continuity of soil water is reduced or disrupted. For the computation of the drainage dynamics-based AWC, the use of the VG model parameters is required for deriving both FC and WP. When computing FC, WP, AWC, and KS using the predicted VG parameters, we maintain the physical relationships among them, which is highly relevant in process-based modelling applications. Misuse of these parameters could lead to flawed model outcomes, impacting policy-making and agricultural management decisions.

It is important to note that soil parameter uncertainty encompasses not only the uncertainty of the PTF but also that stemming from the fitting of the VG model. The prediction uncertainty of soil properties varies significantly. It is essential to tailor its treatment based on the specificities of the target environmental model, particularly when it is utilized as an initial static value, in model calibration, or as a fixed input parameter.

The research emphasized the challenge of selecting suitable datasets and PTFs due to their abundance, providing quantitative performance metrics to aid potential environmental modellers. The workflows and findings presented in the study offer practical guidance for model setup and data preprocessing in various modelling endeavours across Europe, such as hydrological simulations, assessment of soil health, land evaluation, crop modelling, and analysis of soil erosion risk among others. The study's methodology can be applied for soil databases not only in Europe but also in other regions or global datasets, highlighting its potential for broader applicability in multiple modelling contexts worldwide. We encourage the wider scientific and modelling community to use and adopt our recommended workflows to derive soil input parameters, bridging gaps in data for broader utilisation in diverse modelling studies. The presented workflows could be further improved by using a multi-model approach and applying geostatistical methods. The open-source library is available (see Code availability section) for use and adoption to meet the user-specific need.

**Code availability**

The `get_usersoil_table()` function in the R package SWATprepR (Plunge, 2023) was developed for this study. It facilitates the calculation of multiple soil parameters using PTF methods presented in the article. The functionality requires information on soil depth, sand, silt, clay, and organic matter content. The functions use the input information and calculates other soil parameters required for the SWAT+ model. The derivation of HSG is optional based on the KS of the least transmissive layer, depth to water impermeable layer, depth to high water table and information on the existence of any tile drains. The entire

package, source code, documentation, and installation instructions are openly accessible on the GitHub repository: https://github.com/biopsichas/SWATprepR/ .

**Data availability**

6,583 samples of 1999 soil profiles, summing up to 35 % of the EU-HYDI dataset, are available upon request from the European Soil Data Centre (ESDAC) at the European Commission Joint Research Centre. The entire dataset cannot be made publicly available due to its legal restrictions. LUCAS TOPSOIL data can be accessed through ESDAC (European Commission Joint Research Centre, 2024; Panagos et al., 2012, 2022). Local measured topsoil phosphorus data is private, only results of analysis and derived information can be published.

**Supplement link**

Supplement is attached, the link will be added by Copernicus.

**Author contributions**

Conceptualization: BS, AN, NC, FW, MS. Data curation: BS, KP, JM**,** SP. Formal analysis: BS, KP, JM, MS, AN. Funding acquisition: FW, MS. Methodology: BS, KP, NC, AN, FW, SP, MS, BP, JM. Project administration: FW. Software: BS, SP,
KP, JM. Supervision: BS. Validation: BS, KP, AN, NC, JM, SP, MS, FW. Visualization: BS, KP. BS prepared the manuscript with contributions from all co-authors. All authors reviewed the manuscript.

**Competing interests**

The authors declare that they have no known competing financial interests or personal relationships that could have appeared to influence the work reported in this paper.

**Acknowledgements**

This work received funding from the European Union's Horizon 2020 research and innovation programme under grant agreement No. 862756, project OPTAIN (OPtimal strategies to retAIN and re-use water and nutrients in small agricultural catchments across different soil-climatic regions in Europe) and the Sustainable Development and Technologies National Programme of the Hungarian Academy of Sciences (FFT NP FTA).

## Financial support

This research has been supported by the Horizon 2020 Framework Programme (grant no. 862756) and Sustainable Development and Technologies National Programme of the Hungarian Academy of Sciences.

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
