# Peer review of "Addressing soil data needs and data-gaps in catchment scale environmental modelling: the European perspective"

_EGUsphere, 2023_

## Referee Comment (RC2)

Dear Editor and authors,

Thank you for the invitation to review this manuscript. This study compiles a series of datasets of soil properties and pedotransfer functions available for catchment scale modeling, evaluate them and provide an R open script for soil data derivation. I believe this is a valuable piece of work, however, I have several major concerns about its suitability for publication in its current shape but I believe it could be significantly improved. The main concerns is the fact that there is no data openness in this work, the main data used is not available to assess reproducibility and another smaller dataset (independent dataset provided by agricultural company) is not even described. If the authors are willing to share the data and correct the identified problems, I believe this could be an excellent contribution.

My comments for improvement follows:

Introduction

Lines 39 -58 – The introduction could approach more deeply the importance/availability of all the soil properties mentioned in the abstract (Lines 23-25), however the examples provided seemed disconnected between each other. Alternatively, further explanation could be provided to justify giving those examples and not other.

Lines 64-67 – What would this allow the scientific community to do?  Support modeling studies? Assist researchers in the decision for methodological approaches?  Please provide a statement.

Materials and Methods

I was hoping to read here how you determine the compilation in table 1. How to ensure this dataset is complete?

Lines 92 -109 – These are the "soil properties most frequently required by environmental models" based on what? How did you determined this list?

Lines 112 – Regarding the statement "Local and national datasets provide more accurate input information", I would say depends. If is a spatially explicit modeled database I would say definitely yes, if is point data with precise coordinates of the source of the data (e.g. LUCAS) I would say no. Of course, in this condition is not possible to address quantity of information (i.e. density of points) but one would expect local data to present higher data availability, but this is not mentioned in the text. Please be more precise on this matter.

Line 114-119 – I read this text several times, I understand you want to make the case regarding the previous sentence, but is not entirely clear please rephrase it.

Line 119 – 121 – Perhaps there is an expert on soil cracking in this team, but I have to admit I never used such variable. I had to search SWAT documentation, and in the EU only a small amount of soils are classified as Vertisoils. Therefore, I question the authors to justify the importance of soil cracking in comparison with all the other properties listed.

Lines 122 – 126 – These are suggestions and not materials and methods, please move this to an appropriate section, or rephrase it if means you took those considerations when analyzing data.

Lines 127 -134 – same as before.

Lines 139 – 141 – Not clear. I understand there is partial data availability for comparison, but I don't understand what is the consequence of such approach. Please clarify the text.

Line 138 -139 – All right so this EU-HYDI is you data for validation right? It is reasonable to say this dataset hasn't been updated since 2013, and considering the report is actual, should I ask if you only considered measured values? Because I see in the report that part of this dataset contains estimated values. In addition to that, could you point the readers to the data itself? The possibility to reproduce the same analysis is necessary.

Line 152 -333– I honestly don't understand why you wrote this as a protocol format, please reformulate to describe the data analysis that you produced. This is written as a textbook that is not the purpose right? Moreover, out of the sudden I realized that there are datasets that have not been used (e.g depth to water table) and others that have been, could you synthesize that information?

Line 312 – How can we know about local fertilization schemes?

Line 319 - Locally independent measured dataset?? Provided by an agricultural company? How many samples? When samples were taken? Laboratorial methods? Statistical analysis? This is clearly an insufficient methods description.

Results & discussion

Porosity – Lines 396-399 – so what have you done regarding this? If 43% of the samples presented errors, was this data excluded? (explain for all parameters)

Soil erodibility – I thought the Renard et al 1997 was the most used version of the RUSLE model (almost 5 thousand citations according to semantic scholar), but in case I am not right would you provide an information about it?

Predicting Soil Erosion by Water: A Guide to Conservation Planning With the Revised Universal Soil Loss Equation (RUSLE) (usda.gov) page 65.

Either way, why testing only one equation?

Field capacity – Would be worth to make a reference to table 4 early in the beginning. (for all parameters)

Wilting point – I don't understand why the number of points available to assess VG (table 11) differs from the ptf (table 12)

Saturated hydraulic conductivity – found strange you didn't use this database ESSD - SoilKsatDB: global database of soil saturated hydraulic conductivity measurements for geoscience applications (copernicus.org) for comparisons also, any justification for that?

Line 625 -  So these workflow are the result of your analysis, whereas you described the most efficient workflow for better data quality. Right?

Conclusions

Line 653 – "Key findings underscored the significance of local soil data over global or large-scale datasets in environmental modelling", I'm afraid you haven't provided hard evidence on this. Besides this Figure 17, you presented zero information about the sampling, the timing of the year, the laboratorial analysis of this procedure, among many other details. Please provide the sufficient info in order to assess if such assessment is even comparable.

Data availability – I find this justification rather poor considering we are not talking about personal information, nor information that could reduce the value of the land. There is a report online, and this work was paid with taxpayer's money already more than 10 years ago. In the meanwhile, many things changed in science, and the open data is the new reality. Making this data available, in an open data journal as this one, would help the scientific community to overcome many obstacles in the hydrological modelling.

Greetings

Diana Vieira

---

## Author Comment (AC1)

Dear Diana Vieira,

Thank you for the time and effort you have put into the detailed review and suggestions on how to improve our manuscript. Please find below how we would address the raised issues. First, we cite your comment, then provide our response. The changes that would be applied to the manuscript text are highlighted in blue and are visible in the uploaded PDF version of this text.

GENERAL COMMENT:
This study compiles a series of datasets of soil properties and pedotransfer functions available for catchment scale modeling, evaluate them and provide an R open script for soil data derivation. I believe this is a valuable piece of work, however, I have several major concerns about its suitability for publication in its current shape but I believe it could be significantly improved. The main concerns is the fact that there is no data openness in this work, the main data used is not available to assess reproducibility and another smaller dataset (independent dataset provided by agricultural company) is not even described. If the authors are willing to share the data and correct the identified problems, I believe this could be an excellent contribution.

ANSWER FOR GENERAL COMMENT:
1) The reason for using EU-HYDI for this study is that it is the most representative soil hydraulic dataset for Europe that we could use for this study. The internal use and no external openness of the dataset has been requested by the data providers during the establishment of EU-HYDI, which was initiated and coordinated by the EC Joint Research Centre in 2013. Some contributors have given the JRC a licence to make their raw data publicly available on the European Soil Data Centre. Based on information from JRC it will soon be available from ZENODO. Information about data availability is provided here: https://github.com/melwey/euhydi_public . We will add this information to the "Data availability" section of the manuscript as soon as the link will be available from JRC:
"6,583 samples of 1999 soil profiles, summing up to 35 % of the EU-HYDI dataset, are available from ZENODO DOI LINK. The entire dataset cannot be made publicly available due to its legal restrictions."

2) Thank you for pointing out the missing description of the dataset on topsoil phosphorus content. We will add the following text on it under section 2.1 "Evaluation of the methods" in line 143:
"The LUCAS Topsoil dataset (Orgiazzi et al., 2018; Tóth et al., 2013) of 2009 was used for the computation of nutrient content of the surface soil layer. For the assessment of the topsoil phosphorus maps, we used locally measured data obtained from an agricultural company. This dataset includes soil phosphorus content measured at a depth of 30 cm using the acid ammonium acetate lactate extraction (AL-P) method (Egnér et al., 1960) for 34 agricultural parcels in the year 2009. As the phosphorus content was required according to the Olsen method (Olsen-P) (Olsen et al., 1954), we applied the equation of Sárdi et al. (2009) for converting AL-P into Olsen-P. Table 2 shows the descriptive statistics of this database."

SPECIFIC COMMENTS
COMMENT 1:
Lines 39 -58 – The introduction could approach more deeply the importance/availability of all the soil properties mentioned in the abstract (Lines 23-25), however the examples provided seemed disconnected between each other. Alternatively, further explanation could be provided to justify giving those examples and not other.

ANSWER 1:
1.1 We will modify lines 39-44 in the following way:

"The basic soil properties, i.e., soil organic carbon content, particle size distribution, in most cases are locally available at high resolution (< 100 m), but information on bulk density, albedo, soil erodibility factor, soil hydraulic properties, and soil nutrient content is often lacking. There are many PTFs available in the literature that can be used to calculate soil physical (Abbaspour et al. 2019) and hydrological (Bouma and van Lanen, 1987; Van Looy et al., 2017) parameters from basic soil properties, but determining the most suitable one might not be obvious."

1.2 Lines 50-58 will be moved before lines 45-58:
"Information on soil nutrient properties often essential for environmental modelling, such as plant-available soil phosphorus or soil nitrate content, is seldom accessible at a catchment or regional scale. In the absence of measured data on nutrient content, estimating highly mobile nutrients like nitrate poses a challenge due to seasonal fluctuations influenced by factors such as fertilizer application, rainfall, plant nutrient uptake, and microbial activity. Regarding plant-available phosphorus, its levels typically exhibit minimal variation throughout a year. Therefore, approximating its quantity could rely on land use type and area-specific phosphorus fertilization loads (Ballabio et al., 2019). Nevertheless, multiple methods are employed across Europe to measure plant-available soil phosphorus content, potentially requiring conversions between these methods for broader-scale applications. A comprehensive review on conversion equations is available specifically for European studies in Steinfurth et al. (2021)."

1.3 Under lines 45-58 those soil properties are included which are more challenging to retrieve. Thanks for the suggestion, we will add the following explanation to the text:
"Often those soil properties are required as model input data as well, which are rarely available. One example is the data on soil cracking. Cracking intensity and number of cracks are determined by i) soil mineralogy, specifically the amount and type of clay minerals, ii) type of strength that forms soil structure (Lal and Shukla, 2004) and iii) human activity, e.g. tillage, plant spacing. The aperture and closure of cracks can be dynamically related to soil water content (Xing et al., 2023). The data that could describe the variability of cracking is also not easily available, therefore prediction of this parameter is limited at catchment scale."

COMMENT 2:
Lines 64-67 – What would this allow the scientific community to do? Support modeling studies? Assist researchers in the decision for methodological approaches? Please provide a statement.

ANSWER 2:
We will modify lines 64-67 accordingly:
"Therefore, in this study we support soil data retrieval for environmental modelling across Europe by i) systemizing information on open access datasets and PTFs applicable for Europe, ii) demonstrating and quantifying the difference between some PTFs and prediction approaches to cover missing soil properties based on the point data of EU-HYDI, and iii) providing a comprehensive workflow and accompanying open-source R script and library for the derivation of missing soil data."

COMMENT 3:
Materials and Methods
I was hoping to read here how you determine the compilation in table 1. How to ensure this dataset is complete?

ANSWER 3:

We will add the following text in line 111 to highlight the continuous improvement of datasets and most important sites where information on new dataset or updates is expected to be available in the future:

The availability of datasets is continuously improving. The following data sites include most of the updates:

- European Soil Data Centre, which includes soil datasets from Europe and information on EU Soil Observatory (https://esdac.jrc.ec.europa.eu/),
- ISRIC Soil Data Hub, which hosts soil data from around the world (https://data.isric.org/geonetwork/srv/eng/catalog.search#/home),
- soil related layers of the GAEZ Data Portal developed by the Food and Agriculture Organization of the United Nations (FAO) and the International Institute for Applied Systems Analysis (IIASA) (https://data.apps.fao.org),
- soil related layers of the OpenLandMap, which shares open geographical and geoscientific data (https://openlandmap.org).

However, these sources do not include products from specific institutes, such as http://globalchange.bnu.edu.cn/research. The datasets included in Table 1 might be appropriate for regional and continental modelling.

COMMENT 4:
Lines 92 -109 – These are the "soil properties most frequently required by environmental models" based on what? How did you determined this list?

ANSWER 4:
We will modify lines 92-93 in the following way:
"Soil properties most frequently required as static parameters by the environmental models – e.g. (Abbaspour et al., 2019; Dam et al., 2008; Dang et al., 2022; DHI, 2023; Hansen et al., 2012; Šimůnek et al., 2012; Yu et al., 2020) are:"

COMMENT 5:
Lines 112 – Regarding the statement "Local and national datasets provide more accurate input information", I would say depends. If is a spatially explicit modeled database I would say definitely yes, if is point data with precise coordinates of the source of the data (e.g. LUCAS) I would say no. Of course, in this condition is not possible to address quantity of information (i.e. density of points) but one would expect local data to present higher data availability, but this is not mentioned in the text. Please be more precise on this matter.

ANSWER 5:
Thank you for the suggestion, we will modify lines 111-112 accordingly:
"However, for catchment scale and national studies, local and national spatially explicit modelled datasets provide more accurate input information."

COMMENT 6:
Line 114-119 – I read this text several times, I understand you want to make the case regarding the previous sentence, but is not entirely clear please rephrase it.

ANSWER 6:

We will modify lines 112-119 lines to improve clarity:

When a certain local dataset is selected to be used as basic soil information, it is more consistent to compute the missing soil properties from this local data source rather than using other data sources. This allows to maintain consistency between the different soil properties. For example, it is not recommended to combine a local soil property map at 100 m resolution with soil hydraulic properties retrieved from EU-SoilHydroGrids at 250 m resolution (Hengl et al., 2017). Where local soil maps with soil layering, organic carbon content, clay, silt and sand content are available, it is suggested that missing soil properties such as bulk density, soil hydraulic properties, and albedo are estimated from the locally available basic soil properties to ensure consistency.

COMMENT 7:
Line 119 – 121 – Perhaps there is an expert on soil cracking in this team, but I have to admit I never used such variable. I had to search SWAT documentation, and in the EU only a small amount of soils are classified as Vertisoils. Therefore, I question the authors to justify the importance of soil cracking in comparison with all the other properties listed.

ANSWER 7:
We agree that the importance of soil cracking is significantly lower. It is included because the SWAT model requires this input and we decided to follow the structure of the SWAT+ usersoil table, but we will exclude this text because consideration of soil cracks is optional in SWAT+.

COMMENT 8:
Lines 122 – 126 – These are suggestions and not materials and methods, please move this to an appropriate section, or rephrase it if means you took those considerations when analyzing data.

ANSWER 8:
Thank you for the suggestion, we will move these recommendations under "4 Conclusions" section in line 661 in a separate paragraph, before the sentence starting with "When retrieving or deriving …".

COMMENT 9:
Lines 127 -134 – same as before.

ANSWER 9:
We will delete the first sentence of that section and move the rest to line 67.

COMMENT 10:
Lines 139 – 141 – Not clear. I understand there is partial data availability for comparison, but I don't understand what is the consequence of such approach. Please clarify the text.

ANSWER 10:
We will add the following clarification in line 142:
"This approach aimed to facilitate a more accurate and fair comparison among different PTFs, but decreased the number of samples used for the analysis."

COMMENT 11:

Line 138 -139 – All right so this EU-HYDI is you data for validation right? It is reasonable to say this dataset hasn't been updated since 2013, and considering the report is actual, should I ask if you only considered measured values? Because I see in the report that part of this dataset contains estimated values. In addition to that, could you point the readers to the data itself? The possibility to reproduce the same analysis is necessary.

ANSWER 11:

Yes, we considered only measured values. The total number of samples in EU-HYDI is 18,682. For our analysis, the number of samples varied between 1,591 and 11,287 depending on which soil property was analysed. Please find information regarding the link to the data in the "ANSWER FOR GENERAL COMMENT".

COMMENT 12:

Line 152 -333– I honestly don't understand why you wrote this as a protocol format, please reformulate to describe the data analysis that you produced. This is written as a textbook that is not the purpose right? Moreover, out of the sudden I realized that there are datasets that have not been used (e.g depth to water table) and others that have been, could you synthesize that information?

ANSWER 12:

Thank you for the suggestion on reformulation.

12.1 We will add the following section above line 150:

"2.2 Analysed soil properties

We analysed soil physical, hydraulic, and chemical parameters. Under soil physical parameters, we addressed bulk density, porosity, albedo, and soil erodibility factor. For soil hydraulic parameters, we examined water retention, saturated hydraulic conductivity and hydrological soil groups. Regarding soil nutrient content, we focused on topsoil phosphorus content and described the challenges of retrieving soil nitrate content. Hereinafter information about the analysis by soil properties is provided."

12.2 We will reformulate the lines 152-303 in the following way:

1) move lines 151-174 under "3.9 Suggested workflow to derive soil input parameters" section.

2) reorganize level of subtitles: "Soil physical parameters", "Soil hydraulic parameters" and "Soil Chemical parameters" will go under "2.2 Analysed soil properties". There would be no numbered subtitles, only the name of the soil property in the case of bulk density, porosity, albedo, soil erodibility factor, and water retention and saturated hydraulic conductivity.

3) move lines 176-180 and 189-198 under "3.9 Suggested workflow to derive soil input parameters" section. We will keep the following text under "Bulk density section":

"Table 2 lists the PTFs that were tested on point data in EU-HYDI dataset. We selected the bulk density PTFs – derived on soils of the temperate region – based on previous works (Casanova et al., 2016; Hossain et al., 2015; Palladino et al., 2022; Xiangsheng et al., 2016) that tested the prediction performance of several methods."

4) delete line 200 and move lines 209-212 above Table 3.

5) rephrase lines 246-247 in the following way:

"Soil water retention and hydraulic conductivity can be computed from the parameters of the widely used van Genuchten model (VG) (van Genuchten, 1980):"

6) move lines 257-265 above Table 4.

7) delete "This approach is recommended for the computation of FC." from line 268.

8) move text on Hydrological groups (lines 285-303) without subtitle to "3.9 Suggested workflow to derive soil input parameters" section above sentence starting with "Figures 18-21 …".

COMMENT 13:
Line 312 – How can we know about local fertilization schemes?

ANSWER 13:
We will add the following information in line 312:
"Selection of LUCAS Topsoil samples (EUROSTAT, 2015; Orgiazzi et al., 2018) from the adequate year and an agroclimatic zone (Ceglar et al., 2019) similar to the target area, preferably in the same country (NUTS region). Additional criteria for the data selection could be comparable soil types and fertilization systems. If this information is not known, the NUTS2 phosphorus map of the European cropland areas (Tóth et al., 2014) might be useful in the data selection ."

COMMENT 14:
Line 319 - Locally independent measured dataset?? Provided by an agricultural company? How many samples? When samples were taken? Laboratorial methods? Statistical analysis? This is clearly an insufficient methods description.

ANSWER 14:
Thank you for pointing it out, we will add the text inserted under point 2) of ANSWER FOR GENERAL COMMENT in line 143.

COMMENT 15:
Porosity – Lines 396-399 – so what have you done regarding this? If 43% of the samples presented errors, was this data excluded? (explain for all parameters)

ANSWER 15:
Finding discrepancy between porosity and saturated water content is common in international soil hydraulic databases. This rests in the complexity of their measurement when it comes to small laboratory practicalities, as well as assumptions that are frequently made. In terms of measuring water content, samples often drain some of the water by the time the technician has a chance to raise it from the water bath and put the sample onto the scale. It can also happen that some water is still ponding on top of the sample when the measurement is taken. At the same time, there are known error sources in the determination of particle-density, but this property is often only assumed to be 2.65g/cm3 without measuring. The same applies to measuring bulk density: different standards and/or routines are known to exist worldwide (e.g. measuring on clods vs. ring samples, or measuring at a dry state vs. at an equilibrated moisture level), and the determination of this property is also very sensitive to sample quality. Samples can often over or underfill the rings. It is also customary that data reporters just equate porosity and saturated water content without measuring one of them.

When both properties are reported, discrepancies are often seen due to the above reasons – and beyond. The independent user is typically not equipped to judge which one to trust and which one not to. Therefore, it is a routine procedure to cross-check them and report on data quality, but to only be concerned about the value-pairs that show large discrepancies. The proportion of the suspicious samples in the further analysis is low, 39 out of 1591. Therefore, we did not decrease the number of samples used for the analysis.

We will precise lines 396-397 in the following way:

"Figure 5 shows the relationship between porosity and saturated water content for 391 EU-HYDI samples with measured values of both parameters. Among these samples, 56.5% have a total porosity larger or equal to the saturated water content. For the samples where the saturated water content is higher than the total porosity (N = 170), the reason may be the uncertainties in the measurement of both parameters. It is possible that free water could have pounded on top of the sample when its saturated weight was measured, and errors in the measurement of particle density used to compute porosity may have also contributed (Kutílek and Nielsen, 1994; Nimmo, 2004), resulting in a lower porosity."

COMMENT 16:
Soil erodibility – I thought the Renard et al 1997 was the most used version of the RUSLE model (almost 5 thousand citations according to semantic scholar), but in case I am not right would you provide an information about it?
Predicting Soil Erosion by Water: A Guide to Conservation Planning With the Revised Universal Soil Loss Equation (RUSLE) (usda.gov) page 65.
Either way, why testing only one equation?

ANSWER 16:
Thanks you for the suggestion! We wanted to use only those equations which can be readily applied for the soil properties most frequently available and not use the ones that require non-easily available soil properties, such as soil structure or permeability. The Renard et al. 1997 equation fits into the logic, therefore we will add K factor computed with it (K_computed_Lenard) and compare its result with the methods already included.

COMMENT 17:
Field capacity – Would be worth to make a reference to table 4 early in the beginning. (for all parameters)

ANSWER 17:
Thank you for the suggestion, we will add the followings:
in line 452: "The FC defined (see abbreviations in Table 4) based on …"
in line 495: "Calculating WP (see abbreviations in Table 4) from …"
in line 522: "If only AWC (see abbreviations in Table 4) is required …"
in line 560-561: "Figure 14 shows the relationship between measured KS and computed with Eq. (22) based on the fitted VG parameters (KS_VG) (see abbreviation in Table 4)."

COMMENT 18:
Wilting point – I don't understand why the number of points available to assess VG (table 11) differs from the ptf (table 12)

ANSWER 18:
The number of samples differs because Table 11 shows the performances on the VG test set of the EU-HYDI, Table 12 includes performances analysed on the WP test set of the EU-HYDI. Analysis of direct WP prediction (pred_WP) was added to show the difference in accuracy between pred_WP_VG and

pred_WP. To increase clarity, we will add the following modifications in the captions of Table 11 and 12:

"Table 11. Prediction performance of wilting point ($cm^3$ $cm^{-3}$) derived with the VG model, computed by pedotransfer functions on the VG test set of the EU-HYDI dataset. Observed variable is the WP value computed based on the fitted parameters of the VG model. …"

"Table 12. Prediction performance of wilting point ($cm^3$ $cm^{-3}$) computed by pedotransfer functions on the WP test set of the EU-HYDI dataset. Observed variable is the measured WP value. …"

COMMENT 19:
Saturated hydraulic conductivity – found strange you didn't use this database ESSD - SoilKsatDB: global database of soil saturated hydraulic conductivity measurements for geoscience applications (copernicus.org) for comparisons also, any justification for that?

ANSWER 19:
Our aim was to use EU-HYDI dataset, because that is the most representative soil hydraulic dataset for Europe, therefore we would not add further datasets for the KS analysis.

COMMENT 20:
Line 625 - So these workflow are the result of your analysis, whereas you described the most efficient workflow for better data quality. Right?

ANSWER 20:
Yes, thank you for the suggestion on describing the aim of this section. We will add in line 626 the following:

"Based on the above results, we describe the most efficient workflow to retrieve the soil input parameters for European environmental modelling."

Then we will continue with the "protocol type" text that we move from the material and methods – mentioned under "ANSWER 12":

"Initially, the data source of the most relevant soil basic properties, such as soil layering, rooting depth, organic carbon content, clay, silt, and sand content, must be selected. …"

COMMENT 21:
Line 653 – "Key findings underscored the significance of local soil data over global or large-scale datasets in environmental modelling", I'm afraid you haven't provided hard evidence on this. Besides this Figure 17, you presented zero information about the sampling, the timing of the year, the laboratory analysis of this procedure, among many other details. Please provide the sufficient info in order to assess if such assessment is even comparable.

ANSWER 21:
Thank you for highlighting it. We agree with you, we will delete this conflicting sentence, because recommendation on density and timing of sampling, type of laboratory methods and other related topics is out of the scope of this manuscript.

COMMENT 22:

Data availability – I find this justification rather poor considering we are not talking about personal information, nor information that could reduce the value of the land. There is a report online, and this work was paid with taxpayer's money already more than 10 years ago. In the meanwhile, many things changed in science, and the open data is the new reality. Making this data available, in an open data journal as this one, would help the scientific community to overcome many obstacles in the hydrological modelling.

ANSWER 22:
Please find our answer in "ANSWER FOR GENERAL COMMENT". Please note that EC JRC covered the cost of harmonising the data structure and meetings on creating EU-HYDI, but data collection and laboratory analysis was covered by the participating institutions. EU-HYDI is not the property of any of the authors of this manuscript. Two of the authors were collaborators during the construction of EU-HYDI, therefore the dataset could be accessed for the presented analysis according to its licensing. Accommodating the opening of EU-HYDI dataset is beyond the task and reach of these authors.
We will add the information included in "ANSWER FOR GENERAL COMMENT" and the following text to "Data availability" section:
"LUCAS TOPSOIL data can be accessed through European Soil Data Centre (ESDAC) (European Commission Joint Research Centre, 2024; Panagos et al., 2012, 2022). Local measured topsoil phosphorus data is private, only results of analysis and derived information can be published."

REFERENCES ADDED IN THE DOCUMENT:
Abbaspour, K. C., AshrafVaghefi, S., Yang, H. and Srinivasan, R.: Global soil, landuse, evapotranspiration, historical and future weather databases for SWAT Applications, Sci. Data, 6:263, doi:https://doi.org/10.1038/s41597-019-0282-4, 2019.
Dam, J. C. van, Groenendijk, P., Hendriks, R. F. A. and Kroes, J. G.: Advances of Modeling Water Flow in Variably Saturated Soils with SWAP, Vadose Zo. J., 7(2), 640–653, doi:10.2136/VZJ2007.0060, 2008.
Dang, N. A., Jackson, B. M., Tomscha, S. A., Lilburne, L., Burkhard, K., Tran, D. D., Phi, L. H. and Benavidez, R.: Guidelines and a supporting toolbox for parameterising key soil hydraulic properties in hydrological studies and broader integrated modelling, One Ecosyst., 7, doi:10.3897/ONEECO.7.E76410, 2022.
DHI: MIKE SHE User Guide and Reference Manual, Agern Alle, 816, 2023.
Egnér, H., Riehm, H. and Domingo, W. R.: Untersuchungen über die chemische Bodenanalyse als Grundlage für die Beurteilung des Nährstoffzustandes der Böden. II. Chemische Extraktionsmethoden zur Phosphor- und Kaliumbestimmung., Lantbr. Ann., 26, 199-215., 1960.
European Commission Joint Research Centre: European Soil Data Centre (ESDAC), [online] Available from: esdac.jrc.ec.europa.eu (Accessed 17 April 2024), 2024.
Hansen, S., Abrahamsen, P., Petersen, C. T. and Styczen, M.: Daisy: Model Use, Calibration, and Validation, Trans. ASABE, 55(4), 1317–1333, doi:10.13031/2013.42244, 2012.
Kutílek, M. and Nielsen, D. R.: Soil hydrology, Catena-Verlag., 1994.
Nimmo, J. R.: Porosity and Pore-Size Distribution, in Encyclopedia of Soils in the Environment, vol. 3, edited by D. Hillel, pp. 295–303, Elsevier, London., 2004.
Olsen, R., Cole, C. V., Watanabe, F. S. and Dean, L. A.: Estimation of available phosphorus in soils by extraction with sodium bicarbonate., Washington DC., 1954.
Panagos, P., Van Liedekerke, M., Jones, A. and Montanarella, L.: European Soil Data Centre: Response to European policy support and public data requirements, Land use policy, 29(2), 329–338, doi:10.1016/j.landusepol.2011.07.003, 2012.
Panagos, P., Van Liedekerke, M., Borrelli, P., Köninger, J., Ballabio, C., Orgiazzi, A., Lugato, E., Liakos, L., Hervas, J., Jones, A. and Montanarella, L.: European Soil Data Centre 2.0: Soil data and knowledge in support of the EU policies, Eur. J. Soil Sci., 73(6), 1–18, doi:10.1111/ejss.13315, 2022.
Sárdi, K., Csathó, P. and Osztoics, E.: Evaluation of soil phosphorus contents in long-term experiments

from environmental aspects., in Proceedings of the 51st Georgikon Scientific Conference, pp. 807-815., Keszthely, Hungary., 2009.

Šimůnek, J., Van Genuchten, M. T. and Šejna, M.: Hydrus: Model use, calibration, and validation, Trans. ASABE, 55(4), 1261–1274, 2012.

Tóth, G., Guicharnaud, R.-A., Tóth, B. and Hermann, T.: Phosphorus levels in croplands of the European Union with implications for P fertilizer use, Eur. J. Agron., 55, 42–52, doi:10.1016/j.eja.2013.12.008, 2014.

Yu, L., Zeng, Y. and Su, Z.: Understanding the mass, momentum, and energy transfer in the frozen soil with three levels of model complexities, Hydrol. Earth Syst. Sci., 24(10), 4813–4830, doi:10.5194/hess-24-4813-2020, 2020.

---

## Author Comment (AC2)

Dear Reviewer 1,

We appreciate the time and effort you dedicated to thoroughly reviewing our manuscript and offering suggestions for improvement. Below, we outline how we plan to address the raised issues. We reference your comments followed by our corresponding responses. The changes that would be applied to the manuscript text are highlighted in blue and are visible in the uploaded PDF version of this text.

COMMENT 1

L40. Why information on hydraulic properties is often lacking. Particularly, now that the EU Commission proposes the Soil Monitoring Law and such attributes can be important for soil health. By the way, you did not mention anything in your manuscript about soil health. How those attributes are linked to Soil health and how to be used to estimate soil health?

ANSWER 1

1.1 We will revise that sentence in the following way:

"The basic soil properties, i.e., soil organic carbon content, particle size distribution, in most cases are locally available at high resolution (< 100 m), but information on bulk density, albedo, soil erodibility factor, soil hydraulic properties, and soil nutrient content is often lacking."

The reason for the above is that some of these properties are difficult to measure and/or too dynamic (like soil nutrients) and/or not relevant enough in some common applications (e.g. soil albedo is not important in daily agricultural management).

1.2 Soil hydraulic properties are important determinants of soil health as they influence water availability, soil structure, nutrient transport, gas exchange, and surface runoff. For soil health, the threshold values of its indicators might be interdependent. This needs an in depth description which is beyond the scope of this MS. Therefore we might not go into details, but mention soil health in the abstract and under conclusions to show that analysed soil properties are important to assess soil health.

We will highlight soil health in the first sentence of the abstract:

"To effectively guide agricultural management planning strategies and policy, it is important to simulate water quantity and quality patterns and quantify the impact of land use and climate change on soil functions, soil health, hydrological, and other underlying processes."

In the last paragraph of the Conclusions we will add the following text:

"The workflows and findings presented in the study offer practical guidance for model setup and data preprocessing in various modelling endeavours across Europe, such as hydrological simulations, assessment of soil health, land evaluation, crop modelling, and analysis of soil erosion risk among others."

COMMENT 2

Table 1 is not easily readable as proposed. I would propose to have a landscape format and add also a column as Reference. Practically, transfer the reference from first column to the new one. Some corrections are also necessary:

2.1 It is European Union (EU). Of course if you state Member states is similar but more precise to refer to the EU.

2.2 In Soil hydraulic or Physical data , you can add (as there are new datasets that have not been taken into account in your analysis ; such as the Bulk density in the EU and the Global K-factor:

1 https://esdac.jrc.ec.europa.eu/content/topsoil-physical-properties-europe-based-lucas-topsoil-data   *** Clay, silt and sand content; coarse fragments; bulk density; USDA soil textural class; available water capacity. Resolution 500m. as in Ballabio et al., 2016
2 https://esdac.jrc.ec.europa.eu/content/chemical-properties-european-scale-based-lucas-topsoil-data *** pH, pH (CaCl), Cation Exchange Capacity (CEC), Calcium carbonates (CaCO3), C:N ratio, Nitrogen (N), Phosphorus (P) and Potassium (K)  as in Ballabio et al., 2019
3 https://esdac.jrc.ec.europa.eu/content/soil-bulk-density-europe ****** Soil Bulk density in Europe as in Panagos et al., 2024
4 https://esdac.jrc.ec.europa.eu/content/global-soil-erodibility ***** The Global K-factor of Gupta et al. 2024

Those are different that LUCAS. You mentioned LUCAS but those datasets are derived through LUCAS with Machine learning.

ANSWER 2

Thank you for the suggestions, we will format it to landscape and move references into a separate column.
2.1 We will correct it.
2.2 Thank you for the suggestions we will add these four datasets to Table 1.

COMMENT 3

L116-117: in this place and other places of the manuscript. You mentioned this sentence but you have to admit that also application of PTF has a huge uncertainty and it not proper for all different pedo-climatic regions of Europe. EU is so diverse that the just one PTR is not the valid approach for the whole EU. Your statements such as this one as so strong and negative towards other estimations or assessments. In general, I would suggest a more multi-model approach where assessments based on machine learning or interpolations can be compared or assessed together with assessments of PTF.

ANSWER 3

Here we intended to draw attention on not combining local basic soil data with continental or global soil hydraulic maps, but use local basic soil data and derive the missing properties from that to keep consistency in the data (locally available and the derived one).
We will modify lines 116-118:
"Where local soil maps with soil layering, organic carbon content, clay, silt, and sand content are available, it is suggested that missing soil properties, such as bulk density, soil hydraulic properties, and albedo are estimated from the locally available basic soil properties to ensure consistency."

We agree that PTFs have uncertainty, especially when those are applied on soils which have specific soil characteristics, e.g.: specific clay mineralogy, high exchangeable sodium content, high organic carbon content, which overrides the influence of basic soil properties on soil hydraulic characteristics. We will add the following text to highlight it:
"The predictions are subject to uncertainty, which depends on the similarity between the training data used for the selected prediction method and the target area in terms of soil physical and chemical characteristics (Román Dobarco et al., 2019; Tranter et al., 2009)."
We will modify line 129 :
"For the selection of the prediction approaches, three requirements had to be fulfilled: i) the prediction algorithm had to be trained on temperate soils and should not be specific to a particular soil reference group, …"

Our aim is to provide workflows which could be easily applied anywhere in Europe. Using ensemble approach or geostatistical methods is out of the scope of this study, but we will add under conclusions (before the last sentence) the following text:

"The presented workflows could be further improved by using a multi-model approach and applying geostatistical methods."

COMMENT 4

The evaluation of your results takes place using the EU-HYDI . This database as you state is not publicly available. This is really odd and not transparent. Others do European assessments (Soil Grids, ESDAC) or Global assessments but the point data (LUCAS, global point data) are available. Therefore, the approaches are transparent and everybody can test them.

ANSWER 4

The reason for using EU-HYDI for this study is that it is the most representative soil hydraulic dataset for Europe that we could use for this study. The internal use and no external openness of the dataset has been requested by the data providers during the establishment of EU-HYDI, which was initiated and coordinated by EC Joint Research Centre (JRC) in 2013. Some contributors have given the JRC licence to distribute their raw data publicly on the European Soil Data Centre. Based on information from JRC it will soon be available from ZENODO. Information about data availability is provided here: https://github.com/melwey/euhydi_public . We will add this information under "Data availability" section as soon as the link will be available from JRC:

"6,583 samples of 1999 soil profiles, summing up to 35 % of the EU-HYDI dataset is available from ZENODO DOI LINK. The entire dataset cannot be made publicly available due to its legal restrictions."

And add information about the LUCAS dataset:

"LUCAS TOPSOIL data can be accessed through European Soil Data Centre (ESDAC) (European Commission Joint Research Centre, 2024; Panagos et al., 2012, 2022). Local measured topsoil phosphorus data is private, only results of analysis and derived information can be published."

COMMENT 5

In Bulk density, the BD of Hollis is not so simple. Hollis has proposed different PTF per different land uses. Therefore, please pay attention in this use. In addition, as mentioned before, you ignored the recent public assessment of EU Bulk density with 6,000 points available (to download).

ANSWER 5

Thank you for pointing out that Hollis derived more PTFs, we will use the PTFs derived for "cultivated topsoils", "all other mineral horizons" and "all organic horizons" based on the suggestion of Hollis et al. (2012). We will rerun the BD PTF analysis on the LUCAS point BD dataset, as well.

COMMENT 6

For K-factor, the most used function is the by Wischmeier and Smith (1978) and Renard et al. (1997) (as described in Panagos et al., in Eq. (1)).

You use a different equation and then you try to compare your results with the ones which have used the Renard equation. This is a little bit odd.

ANSWER 6

We would like to compare soil erodibility factor values which are derived in different ways, i.e. with the equation and retrieved from the European map to analyse the difference. We believe that this is informative for the readers who are not familiar with the computation of soil erodibility and very interesting to see the differences. The differences in the results can highlight the importance to use the appropriate equation for the computation or treat the erodibility factor as a parameter that has to be further tuned in the model calibration.

Our aim was to use only those equations which can be readily applied for the soil properties most frequently available and not use the ones that require non-easily available soil properties, such as soil structure or permeability. The Renard et al. 1997 equation fits into this logic, therefore we will add K factor computed with it (K_computed_Lenard) and compare its result with the methods already included.

COMMENT 7

In 2.4, for P it is not only the fertilization which plays a role. The available P in soils is a combination of P inputs (Fertilizers, manure, atmospheric deposition, chemical weathering) and outputs (plant uptake, plant residues , erosion). Therefore P level is not influenced only by fertilization. Please be careful and change as appropriate!

ANSWER 7

Thank you for the suggestion. We will modify it with the following text:

"The available P level in agricultural soils is influenced by the P inputs – fertilizers, manure, atmospheric deposition, chemical weathering – and outputs – plant uptake and erosion.  The agricultural management practices (Tóth et al., 2014) are determined by factors such as the country's economy, climate, tillage practices, and crop production characteristics."

COMMENT 8

In soil chemical parameters, authors do not explain why not N and K?

ANSWER 8

Organic nitrogen could be computed from soil organic carbon, but its inorganic part is variable in space and time, therefore it is complex to predict it. We have a paragraph on N at the end of section 2.4. There we explain why we do not consider inorganic N. Potassium is not typically included or computed in environmental models, therefore we did not add information on K.

We will modify line 325 of the manuscript:

"Organic nitrogen can be estimated from soil organic carbon content (Amorim et al., 2022; Liu et al., 2016; Pu et al., 2012; Zhai et al., 2019) if measured data are not available. The concentration of inorganic nitrogen in soil is …"

COMMENT 9

In section 3.1, the performance of BD PTF is not valid as I explained my problematic for the Hollis eq (which is not used properly).

In addition, why you do not test the PTFs against the LUCAS 6000 measured data which are publicly available?

ANSWER 9

Thank you to point it out. We have received the data from EC JRC and will perform the analysis.

COMMENT 10

The problematics on 3.3 have also described above as your results tend to compare non comparable stuff (different equations used!!!).

ANSWER 10

Our aim is to compare different methods for deriving soil erodibility data. We believe it would be informative for readers to see the variation in derived soil erodibility values when using either an available soil erodibility map or an equation. This information will show that these predicted values could be used as preliminary approximations and need for K calibration.

We will add the following modification of line 425-427:

"While both can be used for environmental modelling, i) European soil erodibility map could be linked with LUCAS topsoil dataset and maps, ii) employing Eq. (17) might offer greater consistency with the other local basic and physical soil data, aligning more seamlessly with the modelling process."

COMMENT 11

In 618-619: you refer to something that it is too obvious. IF there are local data, of course they are better. The case is how to cover the data gaps in case local data are not available? That is why I have proposed a multi-model or multi-data source assessment?

ANSWER 11

We will delete the sentence with that obvious statement (starting with "In summary, …"). As mentioned under our ANSWER 3, the aim of this study is to present easy to apply methods. Combining P map derived based on European data with locally measured data fits the idea of multi-data source solution, therefore we will add it to line 622:

"Where available, it is recommended to use measured data to overwrite the geometric mean values, creating a multi-data source solution that reflects the spatial pattern of nutrient content within arable land areas."

COMMENT 12

Similar in your conclusions L653. It is too obvious!

ANSWER 12

We agree, therefore we will delete that sentence and modify lines 654-655 in the following way:

"Local data tend to retain finer soil details, hence it is recommended that users prioritise the utilisation of local (national) soil databases when it is deemed representative and reliable."

COMMENT 13

L675-685: you cannot propose this as the only way forward without making available your reference dataset (EU-HYDI)!!!

ANSWER 13
We intended to write that the methodology can be applied on other databases as well. We will modify the sentence in the following way:
"The study's methodology can be applied for soil databases not only in Europe but also in other regions or global datasets, …"

COMMENT 14
L16: which are the underlying processes?

ANSWER 14
We will add some examples for the underlying processes:
" … it is important to simulate water quantity and quality patterns and to quantify the impact of land use and climate change on soil functions, soil health, hydrological, and other underlying processes."

COMMENT 15
L28-29: why there an significant increase of available datasets?

ANSWER 15
We will modify that sentence in the following way:
"The availability of raw and derived soil datasets, specifically soil hydraulic data, has increased significantly in Europe over the last 10 years as a result of the Soil Strategy and Soil Monitoring Law proposed by the EU Commission."

COMMENT 16
L159: Not only different methods but also through different ISO protocols, depths, etc and in different laboratories which sometimes is impossible to compare.!!

ANSWER 16
We agree.
We will add ISO protocols in L159-160:
"Different countries and institutions measure sand, silt, and clay content using different ISO protocols and methods by recognizing different cutoff limits and classification standards."

COMMENT 17
L325: It is nitrogen

ANSWER 17
Here we would like to refer to inorganic nitrogen forms, which are more soluble in water, therefore are more susceptible to leaching and loss through processes like denitrification, volatilization, and leaching. We will modify lines 325 and 326:
"The concentration of inorganic nitrogen in soil is highly variable in space and time and the dynamic of its amount is significantly influenced by leaching, denitrification, volatilization, and nitrogen fertilization (Zhu et al., 2021).

REFERENCES ADDED IN THE DOCUMENT:

Amorim, H.C.S., Hurtarte, L.C.C., Souza, I.F., Zinn, Y.L., 2022. C:N ratios of bulk soils and particle-size fractions: Global trends and major drivers. Geoderma 425, 116026. https://doi.org/10.1016/j.geoderma.2022.116026

Liu, M., Ussiri, D.A.N., Lal, R., 2016. Soil Organic Carbon and Nitrogen Fractions under Different Land Uses and Tillage Practices. Commun. Soil Sci. Plant Anal. 47, 1528–1541. https://doi.org/10.1080/00103624.2016.1194993

Pu, X., Cheng, H., Shan, Y., Chen, S., Ding, Z., Hao, F., 2012. Factor controlling soil organic carbon and total nitrogen dynamics under long-term conventional cultivation in seasonally frozen soils. Acta Agric. Scand. Sect. B Soil Plant Sci. 62, 749–764. https://doi.org/10.1080/09064710.2012.700318

Román Dobarco, M., Cousin, I., Le Bas, C. and Martin, M. P.: Pedotransfer functions for predicting available water capacity in French soils, their applicability domain and associated uncertainty, Geoderma, 336(April 2018), 81–95, doi:10.1016/J.GEODERMA.2018.08.022, 2019.

Tranter, G., McBratney, a. B. and Minasny, B.: Using distance metrics to determine the appropriate domain of pedotransfer function predictions, Geoderma, 149(3–4), 421–425, doi:10.1016/j.geoderma.2009.01.006, 2009.

Zhai, X., Liu, K., Finch, D.M., Huang, Di., Tang, S., Li, S., Liu, H., Wang, K., 2019. Stoichiometric characteristics of different agroecosystems under the same climatic conditions in the agropastoral ecotone of northern China. Soil Res. 57, 875–882. https://doi.org/10.1071/SR18355

---

## Author Response (AR1)

Dear Editor,

Thank you for reviewing our responses and continuing the evaluation of our manuscript. Please find our point-by-point response to the reviews below this letter and the revised version of our manuscript uploaded.

In addition to the modifications suggested by the reviewers, we have made the following changes:
- Table 1 has been moved to ensure a full page of text precedes the table,
- table and figure on the results of the albedo analysis have been moved into the section of albedo,
- Figure 14 updated: measured value has been moved to x-axis,
- Figure 15: units of both axes corrected,
- Fig. 18: the workflow for predicting soil physical properties has been updated to include computation of silt content (0.002-0.063 mm),
- Fig. 19: caption has been modified to give more precise information on the required particle size distributions,
- a workflow for predicting KS has been added as Fig. 20,
- the numbering of tables, figures and equations has been updated accordingly.

With regards,
Authors

**ANSWER FOR DIANA VIEIRA**

Dear Diana Vieira,

Thank you for the time and effort you have put into the detailed review and suggestions on how to improve our manuscript. Please find below how we have adressed the raised issues. First, we cite your comment, then provide our response. The changes that were applied to the manuscript text are highlighted in blue. Line numbering indicated as e.g., "RL17" refers to the line number 17 in the revised manuscript with track changes.

GENERAL COMMENT:
This study compiles a series of datasets of soil properties and pedotransfer functions available for catchment scale modeling, evaluate them and provide an R open script for soil data derivation. I believe this is a valuable piece of work, however, I have several major concerns about its suitability for publication in its current shape but I believe it could be significantly improved. The main concerns is the fact that there is no data openness in this work, the main data used is not available to assess reproducibility and another smaller dataset (independent dataset provided by agricultural company) is not even described. If the authors are willing to share the data and correct the identified problems, I believe this could be an excellent contribution.

ANSWER FOR GENERAL COMMENT:
1) The reason for using EU-HYDI for this study is that it is the most representative soil hydraulic dataset for Europe that we could use for this study. The internal use and no external openness of the dataset has been requested by the data providers during the establishment of EU-HYDI, which was initiated and coordinated by the EC Joint Research Centre in 2013. Some contributors have given the JRC a licence to make their raw data publicly available on the European Soil Data Centre. Based on information from JRC it will soon be available. Information about data availability is provided here: https://github.com/melwey/euhydi_public . We have added this information to the "Data availability" section of the manuscript:
"6,583 samples of 1999 soil profiles, summing up to 35 % of the EU-HYDI dataset, are available upon request from the European Soil Data Centre (ESDAC) at the European Commission Joint Research Centre. The entire dataset cannot be made publicly available due to its legal restrictions."

2) Thank you for pointing out the missing description of the dataset on topsoil phosphorus content. We have added the following text and table on it under section 2.1 "Evaluation of the methods" in RL188-196:
"The LUCAS Topsoil dataset of 2009 was used for the computation of nutrient content of the surface soil layer. For the assessment of the topsoil phosphorus maps, we used locally measured data obtained from an agricultural company. This dataset includes soil phosphorus content measured at a depth of 30 cm using the acid ammonium acetate lactate extraction (AL-P) method (Egnér et al., 1960) for 34 agricultural parcels in the year 2009. As the phosphorus content was required according to the Olsen method (Olsen-P) (Olsen et al., 1954), we applied the equation of Sárdi et al. (2009) for converting AL-P into Olsen-P. Table 2 shows the descriptive statistics of this database."

Table 2. Descriptive statistics of the locally measured phosphorus content, converted to Olsen-P, from 34 agricultural parcels.

| Min | Max | Range | Mean | Median | Standard deviation |
|---|---|---|---|---|---|
| 8.39 | 65.02 | 56.63 | 27.54 | 25.73 | 13.47 |

SPECIFIC COMMENTS
COMMENT 1:
Lines 39 -58 – The introduction could approach more deeply the importance/availability of all the soil properties mentioned in the abstract (Lines 23-25), however the examples provided seemed disconnected between each other. Alternatively, further explanation could be provided to justify giving those examples and not other.

ANSWER 1:
1.1 We have modified lines 39-44 in the following way in RL41-45:
"The basic soil properties, i.e., soil organic carbon content, particle size distribution, in most cases are locally available at high resolution (< 100 m), but information on bulk density, albedo, soil erodibility factor, soil hydraulic properties, and soil nutrient content is often lacking. There are many PTFs available in the literature that can be used to calculate soil physical (Abbaspour et al. 2019) and hydrological (Bouma and van Lanen, 1987; Van Looy et al., 2017) parameters from basic soil properties, but determining the most suitable one might not be obvious."

1.2 Lines 50-58 have been moved to RL49-57:
"Information on soil nutrient properties often essential for environmental modelling, such as plant-available soil phosphorus or soil nitrate content, is seldom accessible at a catchment or regional scale. In the absence of measured data on nutrient content, estimating highly mobile nutrients like nitrate poses a challenge due to seasonal fluctuations influenced by factors such as fertilizer application, rainfall, plant nutrient uptake, and microbial activity. Regarding plant-available phosphorus, its levels typically exhibit minimal variation throughout a year. Therefore, approximating its quantity could rely on land use type and area-specific phosphorus fertilization loads (Ballabio et al., 2019). Nevertheless, multiple methods are employed across Europe to measure plant-available soil phosphorus content, potentially requiring conversions between these methods for broader-scale applications. A comprehensive review on conversion equations is available specifically for European studies in Steinfurth et al. (2021)."

1.3 Under lines 45-58 those soil properties are included which are more challenging to retrieve. Thanks for the suggestion, we have added the following explanation to the text in RL58:
"Often those soil properties are required as model input data as well, which are rarely available. One example is the data on soil cracking. Cracking intensity and number of cracks are determined by i) soil mineralogy, specifically the amount and type of clay minerals, ii) type of strength that forms soil structure (Lal and Shukla, 2004) and iii) human activity, e.g. tillage, plant spacing. The aperture and closure of cracks can be dynamically related to soil water content (Xing et al., 2023). The data that could describe the variability of cracking is also not easily available, therefore prediction of this parameter is limited at catchment scale."

COMMENT 2:

Lines 64-67 – What would this allow the scientific community to do? Support modeling studies? Assist researchers in the decision for methodological approaches? Please provide a statement.

ANSWER 2:

We have modified these lines accordingly (RL102-106):

"Therefore, in this study we support soil data retrieval for environmental modelling across Europe by i) systemizing information on open access datasets and PTFs applicable for Europe, ii) demonstrating and quantifying the difference between some PTFs and prediction approaches to cover missing soil properties based on the point data of EU-HYDI, and iii) providing a comprehensive workflow and accompanying open-source R script and library for the derivation of missing soil data."

COMMENT 3:

Materials and Methods

I was hoping to read here how you determine the compilation in table 1. How to ensure this dataset is complete?

ANSWER 3:

We have added the following text in RL139-150 to highlight the continuous improvement of datasets and most important sites where information on new dataset or updates is expected to be available in the future:

The availability of datasets is continuously improving. The following data sites include most of the updates:

- European Soil Data Centre, which includes soil datasets from Europe and information on EU Soil Observatory (https://esdac.jrc.ec.europa.eu/),
- ISRIC Soil Data Hub, which hosts soil data from around the world (https://data.isric.org/geonetwork/srv/eng/catalog.search#/home),
- soil related layers of the GAEZ Data Portal developed by the Food and Agriculture Organization of the United Nations (FAO) and the International Institute for Applied Systems Analysis (IIASA) (https://data.apps.fao.org),
- soil related layers of the OpenLandMap, which shares open geographical and geoscientific data (https://openlandmap.org).

Nevertheless, these sources do not include products from specific institutes, such as http://globalchange.bnu.edu.cn/research. The datasets included in Table 1 might be appropriate for regional and continental modelling.

COMMENT 4:

Lines 92 -109 – These are the "soil properties most frequently required by environmental models" based on what? How did you determined this list?

ANSWER 4:

We have modified it in RL119-121 in the following way:

"Soil properties most frequently required as static parameters by the environmental models – e.g. (Abbaspour et al., 2019; Dam et al., 2008; Dang et al., 2022; DHI, 2023; Hansen et al., 2012; Šimůnek et al., 2012; Yu et al., 2020) are:"

COMMENT 5:

Lines 112 – Regarding the statement "Local and national datasets provide more accurate input information", I would say depends. If is a spatially explicit modeled database I would say definitely yes, if is point data with precise coordinates of the source of the data (e.g. LUCAS) I would say no. Of course, in this condition is not possible to address quantity of information (i.e. density of points) but one would expect local data to present higher data availability, but this is not mentioned in the text. Please be more precise on this matter.

ANSWER 5:

Thank you for the suggestion, we have modified it in RL 150-152 accordingly:

"However, for catchment scale and national studies, local and national spatially explicit modelled datasets provide more accurate input information."

COMMENT 6:

Line 114-119 – I read this text several times, I understand you want to make the case regarding the previous sentence, but is not entirely clear please rephrase it.

ANSWER 6:

We have modified it in RL152-159 to improve clarity:

When a certain local dataset is selected to be used as basic soil information, it is more consistent to compute the missing soil properties from this local data source rather than using other data sources. This allows to maintain consistency between the different soil properties. For example, it is not recommended to combine a local soil property map at 100 m resolution with soil hydraulic properties retrieved from EU-SoilHydroGrids at 250 m resolution (Hengl et al., 2017). Where local soil maps with soil layering, organic carbon content, clay, silt and sand content are available, it is suggested that missing soil properties such as bulk density, soil hydraulic properties, and albedo are estimated from the locally available basic soil properties to ensure consistency.

COMMENT 7:

Line 119 – 121 – Perhaps there is an expert on soil cracking in this team, but I have to admit I never used such variable. I had to search SWAT documentation, and in the EU only a small amount of soils are classified as Vertisoils. Therefore, I question the authors to justify the importance of soil cracking in comparison with all the other properties listed.

ANSWER 7:

We agree that the importance of soil cracking is significantly lower. It is included because the SWAT model requires this input and we decided to follow the structure of the SWAT+ usersoil table, but we have removed this text and "information on soil cracking" (line 96) because consideration of soil cracks is optional in SWAT+ (RL 124, 161-164).

COMMENT 8:

Lines 122 – 126 – These are suggestions and not materials and methods, please move this to an appropriate section, or rephrase it if means you took those considerations when analyzing data.

ANSWER 8:

Thank you for the suggestion, we have moved these recommendations under "4 Conclusions" section in RL840-844 in a separate paragraph, before the sentence starting with "When retrieving or deriving …".

COMMENT 9:

Lines 127 -134 – same as before.

ANSWER 9:

We have deleted the first sentence of that section (RL 170-171) and moved the rest to RL106-112.

COMMENT 10:
Lines 139 – 141 – Not clear. I understand there is partial data availability for comparison, but I don't understand what is the consequence of such approach. Please clarify the text.

ANSWER 10:
We have added the following clarification in RL185-186:
"This approach aimed to facilitate a more accurate and fair comparison among different PTFs, but decreased the number of samples used for the analysis."

COMMENT 11:
Line 138 -139 – All right so this EU-HYDI is you data for validation right? It is reasonable to say this dataset hasn't been updated since 2013, and considering the report is actual, should I ask if you only considered measured values? Because I see in the report that part of this dataset contains estimated values. In addition to that, could you point the readers to the data itself? The possibility to reproduce the same analysis is necessary.

ANSWER 11:
Yes, we considered only measured values. The total number of samples in EU-HYDI is 18,682. For our analysis, the number of samples varied between 1,591 and 11,287 depending on which soil property was analysed. Please find information regarding the link to the data in the "ANSWER FOR GENERAL COMMENT".

COMMENT 12:
Line 152 -333– I honestly don't understand why you wrote this as a protocol format, please reformulate to describe the data analysis that you produced. This is written as a textbook that is not the purpose right? Moreover, out of the sudden I realized that there are datasets that have not been used (e.g depth to water table) and others that have been, could you synthesize that information?

ANSWER 12:
Thank you for the suggestion on reformulation.
12.1 We have added the following section in RL203-208:
"2.2 Analysed soil properties
We analysed soil physical, hydraulic, and chemical parameters. Under soil physical parameters, we addressed bulk density, porosity, albedo, and soil erodibility factor. For soil hydraulic parameters, we examined water retention, saturated hydraulic conductivity and hydrological soil groups. Regarding soil nutrient content, we focused on topsoil phosphorus content and described the challenges of retrieving soil nitrate content. Hereinafter information about the analysis by soil properties is provided."
12.2 We have reformulated this part in the following way:
1) moved lines 151-174 under "3.9 Suggested workflow to derive soil input parameters" section (RL 738-760).
2) reorganized level of subtitles: "2.2.1 Soil physical parameters", "2.2.2 Soil hydraulic parameters" and "2.2.3 Soil Chemical parameters" went under "2.2 Analysed soil properties". There are no numbered subtitles, only the name of the soil property in the case of bulk density, porosity, albedo, soil erodibility factor, and water retention and saturated hydraulic conductivity.
3) moved lines 176-180 and 189-198 under "3.9 Suggested workflow to derive soil input parameters" section (RL 761-776). We have kept the following text under "Bulk density section":
"Table 3 lists the PTFs that were tested on point data in EU-HYDI and 2018 LUCAS Topsoil dataset. We selected the bulk density PTFs – derived on soils of the temperate region – based on previous works (Casanova et al., 2016; Hossain et al., 2015; Palladino et al., 2022; Xiangsheng et al., 2016) that tested the prediction performance of several methods."
4) deleted line 200 (RL 263) and moved lines 209-212 above Table 4 (RL267-270).
5) rephrased lines 246-247 in the following way (RL 320-321):
"Soil water retention and hydraulic conductivity can be computed from the parameters of the widely used van Genuchten model (VG) (van Genuchten, 1980):"

6) moved lines 257-265 above Table 5 (RL356-364).

7) deleted "This approach is recommended for the computation of FC." from original line 268 (RL 342).

8) moved text on Hydrological soil groups (original lines 285-303) without subtitle to "3.9 Suggested workflow to derive soil input parameters" (RL777-792).

COMMENT 13:

Line 312 – How can we know about local fertilization schemes?

ANSWER 13:

We have added the following information in RL396-400:

"Selection of LUCAS Topsoil samples (EUROSTAT, 2015; Orgiazzi et al., 2018) from the adequate year and an agroclimatic zone (Ceglar et al., 2019) similar to the target area, preferably in the same country (NUTS region). Additional criteria for the data selection could be comparable soil types and fertilization systems. If this information is not known, the NUTS2 phosphorus map of the European cropland areas might be useful in the data selection (Tóth et al., 2014)."

COMMENT 14:

Line 319 - Locally independent measured dataset?? Provided by an agricultural company? How many samples? When samples were taken? Laboratorial methods? Statistical analysis? This is clearly an insufficient methods description.

ANSWER 14:

Thank you for pointing it out, we have added the text and Table 2 inserted under point 2) of ANSWER FOR GENERAL COMMENT in line 143.

COMMENT 15:

Porosity – Lines 396-399 – so what have you done regarding this? If 43% of the samples presented errors, was this data excluded? (explain for all parameters)

ANSWER 15:

Finding discrepancy between porosity and saturated water content is common in international soil hydraulic databases. This rests in the complexity of their measurement when it comes to small laboratory practicalities, as well as assumptions that are frequently made. In terms of measuring water content, samples often drain some of the water by the time the technician has a chance to raise it from the water bath and put the sample onto the scale. It can also happen that some water is still ponding on top of the sample when the measurement is taken. At the same time, there are known error sources in the determination of particle-density, but this property is often only assumed to be 2.65g/cm3 without measuring. The same applies to measuring bulk density: different standards and/or routines are known to exist worldwide (e.g. measuring on clods vs. ring samples, or measuring at a dry state vs. at an equilibrated moisture level), and the determination of this property is also very sensitive to sample quality. Samples can often over or underfill the rings. It is also customary that data reporters just equate porosity and saturated water content without measuring one of them.

When both properties are reported, discrepancies are often seen due to the above reasons – and beyond. The independent user is typically not equipped to judge which one to trust and which one not to. Therefore, it is a routine procedure to cross-check them and report on data quality, but to only be concerned about the value-pairs that show large discrepancies. The proportion of the suspicious samples in the further analysis is low, 39 out of 1591. Therefore, we did not decrease the number of samples used for the analysis.

We have precised it in RL492-499 in the following way:

"Figure 5 shows the relationship between porosity and saturated water content for 391 EU-HYDI samples with measured values of both parameters. Among these samples, 56.5% have a total porosity larger or equal to the

saturated water content. For the samples where the saturated water content is higher than the total porosity (N = 170), the reason may be the uncertainties in the measurement of both parameters. It is possible that free water could have pounded on top of the sample when its saturated weight was measured, and errors in the measurement of particle density used to compute porosity may have also contributed (Kutílek and Nielsen, 1994; Nimmo, 2004), resulting in a lower porosity."

COMMENT 16:
Soil erodibility – I thought the Renard et al 1997 was the most used version of the RUSLE model (almost 5 thousand citations according to semantic scholar), but in case I am not right would you provide an information about it? Predicting Soil Erosion by Water: A Guide to Conservation Planning With the Revised Universal Soil Loss Equation (RUSLE) (usda.gov) page 65.
Either way, why testing only one equation?

ANSWER 16:
Thank you for the suggestion! We wanted to use only those equations which can be readily applied for the soil properties most frequently available and not use the ones that require non-easily available soil properties, such as soil structure or permeability. The equation published in Renard et al. (1997) (Chapter 3) requires information only about particle size distribution, therefore we have added K factor computed with it (K_Renard) and compared its result with the methods already included.
We have modified this part in RL302-317 in the following way:
"The most widely used equation that can be readily applied to the most frequently available soil properties was published by Sharpley and Williams (1990) (Eq. 17) and Renard et al. (1997) (Eq. 18). The advantage of these methods is that they require only the sand, silt, clay, and organic carbon content of the soil.

$$K_{USLE} = \left( 0.2 \ + \ 0.3 \cdot \exp\left( 0.0256 \cdot \text{sand} \cdot \left( 1 \ - \frac{\text{silt}}{100} \right) \right) \right) \cdot \left( \left( \frac{\text{silt}}{\text{clay} \ + \ \text{silt}} \right)^{0.3} \right) \cdot$$

$$\left( 1 - \left( \frac{0.25 \cdot OC}{(OC \ + \exp(3.72 \ - \ 2.95 \cdot OC))} \right) \right) \cdot \left( 1 - \left( \frac{0.7 \cdot \left( 1 - \frac{\text{sand}}{100} \right)}{\left( \left( 1 - \frac{\text{sand}}{100} \right) + \exp\left( -5.51 + 22.9 \cdot \left( 1 - \frac{\text{sand}}{100} \right) \right) \right)} \right) \right) \tag{17}$$

$$K_{RUSLE} = 7.594 \left( 0.0034 + 0.0405 \cdot exp\left( -0.5 \cdot \left( \frac{log(D_g) + 1659}{0.7101} \right)^2 \right) \right) \quad \text{with } D_g = \exp(0.01 \cdot \sum f_i \cdot \ln m_i) \tag{18}$$

where $K_{USLE}$ is the Universal Soil Loss Equation (USLE), $K_{RUSLE}$ is the Revised Universal Soil Loss Equation (RUSLE) soil erodibility factor $\left( \frac{t \cdot arce \cdot h}{hundreds \ of \ acre \cdot foot - tonf \cdot inch} \right)$, silt is silt content (mass%, 0.002-0.05 mm), sand is sand content (mass %, 0.05-2 mm), OC is organic carbon content (mass %), $D_g$ is the geometric mean particle diameter (mm), $f_i$ is the particle size fraction (mass%), $m_i$ is the arithmetic mean of the particle size limits of the $f_i$ particle size fraction (mm). If the unit is required in $\left( \frac{t \cdot ha \cdot h}{ha \cdot MJ \cdot mm} \right)$, the value of the soil erodibility factor computed with Eq. (17) or Eq. (18) has to be multiplied with 0.1317 (Foster et al., 1981).

We computed the soil erodibility factor for the EU-HYDI dataset. Similarly to the above mentioned albedo, there is no measured soil erodibility value in the EU-HYDI dataset, thus we compared the values computed for the topsoils of EU-HYDI with the values extracted from the European map of Panagos et al. (2014)."

We have added the results of K_Renard analysis in RL530-536 in the following way:
"The Renard equation resulted in a higher median value but lower possible maximum value because the computed soil erodibility factor is capped at 0.044 $\left( \frac{t \cdot ha \cdot h}{ha \cdot MJ \cdot mm} \right)$ due to the constraints of the model. The relationship between the soil erodibility factors derived by different methods is strongest between the values computed using the Sharpley

and Williams (1990) method and the Renard et al. (1997) method. This is logical because both methods consider the particle size distribution of the soil as input information.

Both approaches, whether directly applying the equations (Eq. 17 or 18) or extracting values, generate predicted soil erodibility values."

We have added the results of K_Renard analysis to RL543-559 in the following way:

**Table 9.** Descriptive statistics of soil erodibility factor values computed with the Sharpley and Williams (1990) and Renard et al. (1997) equations on the topsoil samples of the EU-HYDI dataset (N = 11,287) provided in two different units.

| Method | USLE K factor in different units | Min | Max | Range | Mean | Median | Standard deviation |
|---|---|---|---|---|---|---|---|
| Sharpley and Williams (1990) | $\left(\dfrac{t \cdot arce \cdot h}{hundreds\ of\ acre \cdot foot - tonf \cdot inch}\right)$ | 0.00 | 0.48 | 0.48 | 0.27 | 0.27 | 0.09 |
| | $\left(\dfrac{t \cdot ha \cdot h}{ha \cdot MJ \cdot mm}\right)$ | 0.000 | 0.063 | 0.063 | 0.036 | 0.035 | 0.012 |
| Renard et al. (1997) | $\left(\dfrac{t \cdot arce \cdot h}{hundreds\ of\ acre \cdot foot - tonf \cdot inch}\right)$ | 0.05 | 0.33 | 0.29 | 0.24 | 0.27 | 0.09 |
| | $\left(\dfrac{t \cdot ha \cdot h}{ha \cdot MJ \cdot mm}\right)$ | 0.006 | 0.044 | 0.038 | 0.032 | 0.035 | 0.012 |

[Figure]

**Figure 7.** Histogram of the soil erodibility factor $\left(\dfrac{t \cdot ha \cdot h}{ha \cdot MJ \cdot mm}\right)$ computed with the Sharpley and Williams (1990) (K_Sharpley_Williams, N = 3276) (a) and Renard et al. (1997) (K_Renard, N = 3276) (b) equations on the topsoil samples of the EU-HYDI dataset, and extracted from the soil erodibility map of Europe for the EU-HYDI topsoil layers without (K_ESDAC, N = 3100) (c) and considering stoniness (K_st_ESDAC, N = 3190) (d). Vertical dashed lines indicate the median values.

[Figure]

**Figure 8.** Scatterplot of computed soil erodibility factors versus extracted from the European soil erodibility factor map without (a, c) and with considering stoniness (b, d) based on the topsoil samples of the EU-HYDI dataset $\left(\frac{t\cdot ha\cdot h}{ha\cdot MJ\cdot mm}\right)$. Plot (e) shows the relationship between the values computed by the Sharpley and Williams (1990) and Renard et al. (1997) methods.

COMMENT 17:
Field capacity – Would be worth to make a reference to table 4 early in the beginning. (for all parameters)

ANSWER 17:
Thank you for the suggestion. Numbering of that table has been changed to Table 5. We have added the followings:
in RL561: "The FC defined (see abbreviations in Table 5) based on …"
in RL604: "Calculating WP (see abbreviations in Table 5) from …"

in RL631: "If only AWC (see abbreviations in Table 5) is required …"
in RL669-670: "Figure 14 shows the relationship between measured KS and computed with Eq. (22) based on the fitted VG parameters (KS_VG) (see abbreviation in Table 5)."

COMMENT 18:
Wilting point – I don't understand why the number of points available to assess VG (table 11) differs from the ptf (table 12)

ANSWER 18:
The number of samples differs because Table 11 shows the performances on the VG test set of the EU-HYDI, Table 12 includes performances analysed on the WP test set of the EU-HYDI. Analysis of direct WP prediction (pred_WP) was added to show the difference in accuracy between pred_WP_VG and pred_WP. To increase clarity, we have added the following modifications in the captions of Table 11 (RL 616-617) and 12 (RL 622):
"Table 11. Prediction performance of wilting point ($cm^3$ $cm^{-3}$) derived with the VG model, computed by pedotransfer functions on the VG test set of the EU-HYDI dataset. Observed variable is the WP value computed based on the fitted parameters of the VG model. …"
"Table 12. Prediction performance of wilting point ($cm^3$ $cm^{-3}$) computed by pedotransfer functions on the WP test set of the EU-HYDI dataset. Observed variable is the measured WP value. …"

COMMENT 19:
Saturated hydraulic conductivity – found strange you didn't use this database ESSD - SoilKsatDB: global database of soil saturated hydraulic conductivity measurements for geoscience applications (copernicus.org) for comparisons also, any justification for that?

ANSWER 19:
Our aim was to use EU-HYDI dataset, because that is the most representative soil hydraulic dataset for Europe, therefore we would not add further datasets for the KS analysis.

COMMENT 20:
Line 625 - So these workflow are the result of your analysis, whereas you described the most efficient workflow for better data quality. Right?

ANSWER 20:
Yes, thank you for the suggestion on describing the aim of this section. We have added in RL736-737 the following:
"Based on the above results, we describe the most efficient workflow to retrieve the soil input parameters for European environmental modelling."
Then we have continued with the "protocol type" text that we move from the material and methods – mentioned under "ANSWER 12":
"Initially, the data source of the most relevant soil basic properties, such as soil layering, rooting depth, organic carbon content, clay, silt, and sand content, must be selected. …"

COMMENT 21:
Line 653 – "Key findings underscored the significance of local soil data over global or large-scale datasets in environmental modelling", I'm afraid you haven't provided hard evidence on this. Besides this Figure 17, you presented zero information about the sampling, the timing of the year, the laboratory analysis of this procedure, among many other details. Please provide the sufficient info in order to assess if such assessment is even comparable.

ANSWER 21:

Thank you for highlighting it. We agree with you, we have deleted this conflicting sentence (RL 831), because recommendation on density and timing of sampling, type of laboratory methods and other related topics is out of the scope of this manuscript.

COMMENT 22:

Data availability – I find this justification rather poor considering we are not talking about personal information, nor information that could reduce the value of the land. There is a report online, and this work was paid with taxpayer's money already more than 10 years ago. In the meanwhile, many things changed in science, and the open data is the new reality. Making this data available, in an open data journal as this one, would help the scientific community to overcome many obstacles in the hydrological modelling.

ANSWER 22:

Please find our answer in "ANSWER FOR GENERAL COMMENT". Please note that EC JRC covered the cost of harmonising the data structure and meetings on creating EU-HYDI, but data collection and laboratory analysis was covered by the participating institutions. EU-HYDI is not the property of any of the authors of this manuscript. Two of the authors were collaborators during the construction of EU-HYDI, therefore the dataset could be accessed for the presented analysis according to its licensing. Accommodating the opening of EU-HYDI dataset is beyond the task and reach of these authors.

We have added the information included in "ANSWER FOR GENERAL COMMENT" and the following text to "Data availability" section (RL 882-884):

"LUCAS TOPSOIL data can be accessed through ESDAC (European Commission Joint Research Centre, 2024; Panagos et al., 2012, 2022). Local measured topsoil phosphorus data is private, only results of analysis and derived information can be published."

REFERENCES ADDED IN THE ANSWERS FOR DIANA VIEIRA:

Abbaspour, K. C., AshrafVaghefi, S., Yang, H. and Srinivasan, R.: Global soil, landuse, evapotranspiration, historical and future weather databases for SWAT Applications, Sci. Data, 6:263, doi:https://doi.org/10.1038/s41597-019-0282-4, 2019.

Dam, J. C. van, Groenendijk, P., Hendriks, R. F. A. and Kroes, J. G.: Advances of Modeling Water Flow in Variably Saturated Soils with SWAP, Vadose Zo. J., 7(2), 640–653, doi:10.2136/VZJ2007.0060, 2008.

Dang, N. A., Jackson, B. M., Tomscha, S. A., Lilburne, L., Burkhard, K., Tran, D. D., Phi, L. H. and Benavidez, R.: Guidelines and a supporting toolbox for parameterising key soil hydraulic properties in hydrological studies and broader integrated modelling, One Ecosyst., 7, doi:10.3897/ONEECO.7.E76410, 2022.

DHI: MIKE SHE User Guide and Reference Manual, Agern Alle, 816, 2023.

Egnér, H., Riehm, H. and Domingo, W. R.: Untersuchungen über die chemische Bodenanalyse als Grundlage für die Beurteilung des Nährstoffzustandes der Böden. II. Chemische Extraktionsmethoden zur Phosphor- und Kaliumbestimmung., Lantbr. Ann., 26, 199-215., 1960.

European Commission Joint Research Centre: European Soil Data Centre (ESDAC), [online] Available from: esdac.jrc.ec.europa.eu (Accessed 17 April 2024), 2024.

Hansen, S., Abrahamsen, P., Petersen, C. T. and Styczen, M.: Daisy: Model Use, Calibration, and Validation, Trans. ASABE, 55(4), 1317–1333, doi:10.13031/2013.42244, 2012.

Kutílek, M. and Nielsen, D. R.: Soil hydrology, Catena-Verlag., 1994.

Nimmo, J. R.: Porosity and Pore-Size Distribution, in Encyclopedia of Soils in the Environment, vol. 3, edited by D. Hillel, pp. 295–303, Elsevier, London., 2004.

Olsen, R., Cole, C. V., Watanabe, F. S. and Dean, L. A.: Estimation of available phosphorus in soils by extraction with sodium bicarbonate., Washington DC., 1954.

Panagos, P., Van Liedekerke, M., Jones, A. and Montanarella, L.: European Soil Data Centre: Response to European policy support and public data requirements, Land use policy, 29(2), 329–338,

doi:10.1016/j.landusepol.2011.07.003, 2012.

Panagos, P., Van Liedekerke, M., Borrelli, P., Köninger, J., Ballabio, C., Orgiazzi, A., Lugato, E., Liakos, L., Hervas, J., Jones, A. and Montanarella, L.: European Soil Data Centre 2.0: Soil data and knowledge in support of the EU policies, Eur. J. Soil Sci., 73(6), 1–18, doi:10.1111/ejss.13315, 2022.

Renard, K. G., Foster, G. R., Weesies, G. A., McCool, D. K. and Yoder, D. C. 1997. Predicting soil erosion by water: a guide to conservation planning with the Revised Universal Soil Loss Equation (RUSLE), U.S. Department of Agriculture. [online] Available from: http://www.ars.usda.gov/SP2UserFiles/Place/64080530/RUSLE/AH_703.pdf

Sárdi, K., Csathó, P. and Osztoics, E.: Evaluation of soil phosphorus contents in long-term experiments from environmental aspects., in Proceedings of the 51st Georgikon Scientific Conference, pp. 807-815., Keszthely, Hungary., 2009.

Šimůnek, J., Van Genuchten, M. T. and Šejna, M.: Hydrus: Model use, calibration, and validation, Trans. ASABE, 55(4), 1261–1274, 2012.

Tóth, G., Guicharnaud, R.-A., Tóth, B. and Hermann, T.: Phosphorus levels in croplands of the European Union with implications for P fertilizer use, Eur. J. Agron., 55, 42–52, doi:10.1016/j.eja.2013.12.008, 2014.

Yu, L., Zeng, Y. and Su, Z.: Understanding the mass, momentum, and energy transfer in the frozen soil with three levels of model complexities, Hydrol. Earth Syst. Sci., 24(10), 4813–4830, doi:10.5194/hess-24-4813-2020, 2020.

**ANSWER FOR ANONYMOUS REVIEWER 1**

Dear Reviewer 1,
We appreciate the time and effort you dedicated to thoroughly reviewing our manuscript and offering suggestions for improvement. Below, we outline how we addressed the raised issues. We reference your comments followed by our corresponding responses. The changes applied to the manuscript text are highlighted in blue and are visible in the uploaded PDF version of this text. Line numbering indicated as e.g., "RL17" refers to the line number 17 in the revised manuscript with track changes.

COMMENT 1
L40. Why information on hydraulic properties is often lacking. Particularly, now that the EU Commission proposes the Soil Monitoring Law and such attributes can be important for soil health. By the way, you did not mention anything in your manuscript about soil health. How those attributes are linked to Soil health and how to be used to estimate soil health?

ANSWER 1
1.1 We have revised that sentence in the following way in RL41-43:
"The basic soil properties, i.e., soil organic carbon content, particle size distribution, in most cases are locally available at high resolution (< 100 m), but information on bulk density, albedo, soil erodibility factor, soil hydraulic properties, and soil nutrient content is often lacking."
The reason for the above is that some of these properties are difficult to measure and/or too dynamic (like soil nutrients) and/or not relevant enough in some common applications (e.g. soil albedo is not important in daily agricultural management).

1.2 Soil hydraulic properties are important determinants of soil health as they influence water availability, soil structure, nutrient transport, gas exchange, and surface runoff. For soil health, the threshold values of its indicators might be interdependent. This needs an in depth description which is beyond the scope of this MS. Therefore we might not go into details, but mention soil health in the abstract and under conclusions to show that analysed soil properties are important to assess soil health.
We have highlighted soil health in the first sentence of the abstract (RL16-17):

"To effectively guide agricultural management planning strategies and policy, it is important to simulate water quantity and quality patterns and quantify the impact of land use and climate change on soil functions, soil health, hydrological, and other underlying processes."

In the last paragraph of the Conclusions we will add the following text:

"The workflows and findings presented in the study offer practical guidance for model setup and data preprocessing in various modelling endeavours across Europe, such as hydrological simulations, assessment of soil health, land evaluation, crop modelling, and analysis of soil erosion risk among others."

COMMENT 2

Table 1 is not easily readable as proposed. I would propose to have a landscape format and add also a column as Reference. Practically, transfer the reference from first column to the new one. Some corrections are also necessary:

2.1 It is European Union (EU). Of course if you state Member states is similar but more precise to refer to the EU.

2.2 In Soil hydraulic or Physical data , you can add (as there are new datasets that have not been taken into account in your analysis ; such as the Bulk density in the EU and the Global K-factor:

1. https://esdac.jrc.ec.europa.eu/content/topsoil-physical-properties-europe-based-lucas-topsoil-data *** Clay, silt and sand content; coarse fragments; bulk density; USDA soil textural class; available water capacity. Resolution 500m. as in Ballabio et al., 2016
2. https://esdac.jrc.ec.europa.eu/content/chemical-properties-european-scale-based-lucas-topsoil-data *** pH, pH (CaCl), Cation Exchange Capacity (CEC), Calcium carbonates (CaCO3), C:N ratio, Nitrogen (N), Phosphorus (P) and Potassium (K)  as in Ballabio et al., 2019
3. https://esdac.jrc.ec.europa.eu/content/soil-bulk-density-europe ****** Soil Bulk density in Europe as in Panagos et al., 2024
4. https://esdac.jrc.ec.europa.eu/content/global-soil-erodibility ***** The Global K-factor of Gupta et al. 2024

Those are different that LUCAS. You mentioned LUCAS but those datasets are derived through LUCAS with Machine learning.

COMMENT 2

ANSWER 2

Thank you for the suggestions. We have formatted it to landscape position within a portrait page format, in line with the journal's formatting requirements, and moved the references into a separate column (RL 76-99).

2.1 We have corrected it.

2.2 Thank you for the suggestions we have added these four datasets to Table 1. We further extend the table with information on DSOLMap (page 13 in the revised manuscript with track changes).

COMMENT 3

L116-117: in this place and other places of the manuscript. You mentioned this sentence but you have to admit that also application of PTF has a huge uncertainty and it not proper for all different pedo-climatic regions of Europe. EU is so diverse that the just one PTR is not the valid approach for the whole EU. Your statements such as this one as so strong and negative towards other estimations or assessments. In general, I would suggest a more multi-model approach where assessments based on machine learning or interpolations can be compared or assessed together with assessments of PTF.

ANSWER 3

Here we intended to draw attention on not combining local basic soil data with continental or global soil hydraulic maps, but use local basic soil data and derive the missing properties from that to keep consistency in the data (locally available and the derived one).

We have modified it in RL157-159:

"Where local soil maps with soil layering, organic carbon content, clay, silt, and sand content are available, it is suggested that missing soil properties, such as bulk density, soil hydraulic properties, and albedo are estimated from the locally available basic soil properties to ensure consistency."

We agree that PTFs have uncertainty, especially when those are applied on soils which have specific soil characteristics, e.g.: specific clay mineralogy, high exchangeable sodium content, high organic carbon content, which overrides the influence of basic soil properties on soil hydraulic characteristics. We have added the following text to highlight it in RL159-161:

"The predictions are subject to uncertainty, which depends on the similarity between the training data used for the selected prediction method and the target area in terms of soil physical and chemical characteristics (Román Dobarco et al., 2019; Tranter et al., 2009)."

We have modified and moved it to the last paragraph of "1 Introduction" (RL106-107):

"For the selection of the prediction approaches, three requirements had to be fulfilled: i) the prediction algorithm had to be trained on temperate soils and should not be specific to a particular soil reference group, …"

Our aim is to provide workflows which could be easily applied anywhere in Europe. Using ensemble approach or geostatistical methods is out of the scope of this study, but we have added under conclusions in RL867-868 the following text:

"The presented workflows could be further improved by using a multi-model approach and applying geostatistical methods."

**COMMENT 4**
The evaluation of your results takes place using the EU-HYDI . This database as you state is not publicly available. This is really odd and not transparent. Others do European assessments (Soil Grids, ESDAC) or Global assessments but the point data (LUCAS, global point data) are available. Therefore, the approaches are transparent and everybody can test them.

**ANSWER 4**
The reason for using EU-HYDI for this study is that it is the most representative soil hydraulic dataset for Europe that we could use for this study. The internal use and no external openness of the dataset has been requested by the data providers during the establishment of EU-HYDI, which was initiated and coordinated by EC Joint Research Centre (JRC) in 2013. Some contributors have given the JRC a licence to make their raw data publicly available on the European Soil Data Centre. Based on information from JRC it will soon be available. Information about data availability is provided here: https://github.com/melwey/euhydi_public . We have added this information to the "Data availability" section of the manuscript:

"6,583 samples of 1999 soil profiles, summing up to 35 % of the EU-HYDI dataset, are available upon request from the European Soil Data Centre (ESDAC) at the European Commission Joint Research Centre. The entire dataset cannot be made publicly available due to its legal restrictions."

And added information about the LUCAS dataset:

"LUCAS TOPSOIL data can be accessed through ESDAC (European Commission Joint Research Centre, 2024; Panagos et al., 2012, 2022). Local measured topsoil phosphorus data is private, only results of analysis and derived information can be published."

**COMMENT 5**
In Bulk density, the BD of Hollis is not so simple. Hollis has proposed different PTF per different land uses. Therefore, please pay attention in this use. In addition, as mentioned before, you ignored the recent public assessment of EU Bulk density with 6,000 points available (to download).

**ANSWER 5**
Thank you for pointing out that Hollis derived more PTFs, we have modified the equation in our computation: incorporated the PTFs derived for "cultivated topsoils", "all other mineral horizons" and "all organic horizons" based on the suggestion of Hollis et al. (2012). We have rerun the BD PTF analysis on the LUCAS point BD dataset, as well.

The following text has been added accordingly:

1. Under "2.1 Evaluation of methods in RL186-187:

[revised manuscript text omitted]

5. In supplemetary material Figure S1 has been updated accordingly.

  COMMENT 6
For K-factor, the most used function is the by Wischmeier and Smith (1978) and Renard et al. (1997) (as described in Panagos et al., in Eq. (1)).
You use a different equation and then you try to compare your results with the ones which have used the Renard equation. This is a little bit odd.

ANSWER 6

We would like to compare soil erodibility factor values which are derived in different ways, i.e. with the equation and retrieved from the European map to analyse the difference. We believe that this is informative for the readers who are not familiar with the computation of soil erodibility and very interesting to see the differences. The differences in the results can highlight the importance to use the appropriate equation for the computation or treat the erodibility factor as a parameter that has to be further tuned in the model calibration.

Our aim was to use only those equations which can be readily applied for the soil properties most frequently available and not use the ones that require non-easily available soil properties, such as soil structure or permeability. The equation published in Renard et al. (1997) (Chapter 3) requires information only about particle size distribution, therefore we have added K factor computed with it (K_Renard) and compared its result with the methods already included.

We have added modifications in RL302-317 in the following way:

"The most widely used equation that can be readily applied to the most frequently available soil properties was published by Sharpley and Williams (1990) (Eq. 17) and Renard et al. (1997) (Eq. 18). The advantage of these methods is that they require only the sand, silt, clay, and organic carbon content of the soil.

$$K_{USLE} = \left(0.2 + 0.3 \cdot \exp\left(0.0256 \cdot \text{sand} \cdot \left(1 - \frac{\text{silt}}{100}\right)\right)\right) \cdot \left(\left(\frac{\text{silt}}{\text{clay} + \text{silt}}\right)^{0.3}\right) \cdot$$

$$\left(1 - \left(\frac{0.25 \cdot OC}{(OC + \exp(3.72 - 2.95 \cdot OC))}\right)\right) \cdot \left(1 - \left(\frac{0.7 \cdot \left(1 - \frac{\text{sand}}{100}\right)}{\left(\left(1 - \frac{\text{sand}}{100}\right) + \exp\left(-5.51 + 22.9 \cdot \left(1 - \frac{\text{sand}}{100}\right)\right)\right)}\right)\right) \quad (17)$$

$$K_{RUSLE} = 7.594\left(0.0034 + 0.0405 \cdot \exp\left(-0.5 \cdot \left(\frac{\log(D_g) + 1659}{0.7101}\right)^2\right)\right) \quad \text{with } D_g = \exp(0.01 \cdot \sum f_i \cdot \ln m_i) \quad (18)$$

where $K_{USLE}$ is the Universal Soil Loss Equation (USLE), $K_{RUSLE}$ is the Revised Universal Soil Loss Equation (RUSLE) soil erodibility factor $\left(\frac{t \cdot arce \cdot h}{hundreds \ of \ acre \cdot foot - tonf \cdot inch}\right)$, silt is silt content (mass%, 0.002-0.05 mm), sand is sand content (mass %, 0.05-2 mm), OC is organic carbon content (mass %), $D_g$ is the geometric mean particle diameter (mm), $f_i$ is the particle size fraction (mass%), $m_i$ is the arithmetic mean of the particle size limits of the $f_i$ particle size fraction (mm) . If the unit is required in $\left(\frac{t \cdot ha \cdot h}{ha \cdot MJ \cdot mm}\right)$, the value of the soil erodibility factor computed with Eq. (17) or Eq. (18) has to be multiplied with 0.1317 (Foster et al., 1981).

We computed the soil erodibility factor for the EU-HYDI dataset. Similarly to the above mentioned albedo, there is no measured soil erodibility value in the EU-HYDI dataset, thus we compared the values computed for the topsoils of EU-HYDI with the values extracted from the European map of Panagos et al. (2014)."

We have added the results of K_Renard analysis in RL530-536 in the following way:

"The Renard equation resulted in a higher median value but lower possible maximum value because the computed soil erodibility factor is capped at 0.044 $\left(\frac{t \cdot ha \cdot h}{ha \cdot MJ \cdot mm}\right)$ due to the constraints of the model. The relationship between the soil erodibility factors derived by different methods is strongest between the values computed using the Sharpley and Williams (1990) method and the Renard et al. (1997) method. This is logical because both methods consider the particle size distribution of the soil as input information.

Both approaches, whether directly applying the equations (Eq. 17 or 18) or extracting values, generate predicted soil erodibility values."

We have added the results of K_Renard analysis in RL543-559 in the following way:

**Table 9.** Descriptive statistics of soil erodibility factor values computed with the Sharpley and Williams (1990) and Renard et al. (1997) equations on the topsoil samples of the EU-HYDI dataset (N = 11,287) provided in two different units.

| Method | USLE K factor in different units | Min | Max | Range | Mean | Median | Standard deviation |
|---|---|---|---|---|---|---|---|
| Sharpley and Williams (1990) | $\left(\dfrac{t \cdot arce \cdot h}{hundreds\ of\ acre \cdot foot - tonf \cdot inch}\right)$ | 0.00 | 0.48 | 0.48 | 0.27 | 0.27 | 0.09 |
| | $\left(\dfrac{t \cdot ha \cdot h}{ha \cdot MJ \cdot mm}\right)$ | 0.000 | 0.063 | 0.063 | 0.036 | 0.035 | 0.012 |
| Renard et al. (1997) | $\left(\dfrac{t \cdot arce \cdot h}{hundreds\ of\ acre \cdot foot - tonf \cdot inch}\right)$ | 0.05 | 0.33 | 0.29 | 0.24 | 0.27 | 0.09 |
| | $\left(\dfrac{t \cdot ha \cdot h}{ha \cdot MJ \cdot mm}\right)$ | 0.006 | 0.044 | 0.038 | 0.032 | 0.035 | 0.012 |

[Figure]

**Figure 7.** Histogram of the soil erodibility factor $\left(\dfrac{t \cdot ha \cdot h}{ha \cdot MJ \cdot mm}\right)$ computed with the Sharpley and Williams (1990) (K_Sharpley_Williams, N = 3276) (a) and Renard et al. (1997) (K_Renard, N = 3276) (b) equations on the topsoil samples of the EU-HYDI dataset, and extracted from the soil erodibility map of Europe for the EU-HYDI topsoil layers without (K_ESDAC, N = 3100) (c) and considering stoniness (K_st_ESDAC, N = 3190) (d). Vertical dashed lines indicate the median values.

[Figure]

**Figure 8.** Scatterplot of computed soil erodibility factors versus extracted from the European soil erodibility factor map without (a, c) and with considering stoniness (b, d) based on the topsoil samples of the EU-HYDI dataset $\left(\frac{t \cdot ha \cdot h}{ha \cdot MJ \cdot mm}\right)$. Plot (e) shows the relationship between the values computed by the Sharpley and Williams (1990) and Renard et al. (1997) methods.

COMMENT 7

In 2.4, for P it is not only the fertilization which plays a role. The available P in soils is a combination of P inputs (Fertilizers, manure, atmospheric deposition, chemical weathering) and outputs (plant uptake, plant residues, erosion). Therefore P level is not influenced only by fertilization. Please be careful and change as appropriate!

ANSWER 7

Thank you for the suggestion. We have modified it in RL390-393 with the following text:

"The available P level in agricultural soils is influenced by the P inputs – fertilizers, manure, atmospheric deposition, chemical weathering – and outputs – plant uptake and erosion. The agricultural management practices (Tóth et al., 2014) are determined by factors such as the country's economy, climate, tillage practices, and crop production characteristics."

**COMMENT 8**
In soil chemical parameters, authors do not explain why not N and K?

**ANSWER 8**
Organic nitrogen could be computed from soil organic carbon, but its inorganic part is variable in space and time, therefore it is complex to predict it. We have a paragraph on N at the end of section 2.4. There we explain why we do not consider inorganic N. Potassium is not typically included or computed in environmental models, therefore we did not add information on K.

We have modified it in RL412-413 of the manuscript:

"Organic nitrogen can be estimated from soil organic carbon content (Amorim et al., 2022; Liu et al., 2016; Pu et al., 2012; Zhai et al., 2019) if measured data are not available. The concentration of inorganic nitrogen in soil is …"

**COMMENT 9**
In section 3.1, the performance of BD PTF is not valid as I explained my problematic for the Hollis eq (which is not used properly).

In addition, why you do not test the PTFs against the LUCAS 6000 measured data which are publicly available?

**ANSWER 9**
Thank you to point it out. We have received the data from EC JRC and have performed the analysis. Please find modification related to it under our ANSWER 5.

**COMMENT 10**
The problematics on 3.3 have also described above as your results tend to compare non comparable stuff (different equations used!!!).

**ANSWER 10**
Our aim is to compare different methods for deriving soil erodibility data. We believe it would be informative for readers to see the variation in derived soil erodibility values when using either an available soil erodibility map or an equation. This information will show that these predicted values could be used as preliminary approximations and need for K calibration.

We have added the following modification in RL536-538:

"While both can be used for environmental modelling, i) European soil erodibility map could be linked with LUCAS topsoil dataset and maps, ii) employing Eq. (14) or (15) might offer greater consistency with the other local basic and physical soil data, aligning more seamlessly with the modelling process."

**COMMENT 11**
In 618-619: you refer to something that it is too obvious. IF there are local data, of course they are better. The case is how to cover the data gaps in case local data are not available? That is why I have proposed a multi-model or multi-data source assessment?

ANSWER 11

We have deleted the sentence with that obvious statement (starting with "In summary, …") (RL 726-728). As mentioned under our ANSWER 3, the aim of this study is to present easy to apply methods. Combining P map derived based on European data with locally measured data fits the idea of multi-data source solution, therefore we have added it in RL731-732:

"Where available, it is recommended to use measured data to overwrite the geometric mean values, creating a multi-data source solution that reflects the spatial pattern of nutrient content within arable land areas."

COMMENT 12

Similar in your conclusions L653. It is too obvious!

ANSWER 12

We agree, therefore we have deleted that sentence in RL831 and modified RL833 in the following way:
"Local data tend to retain finer soil details, hence it is recommended that users prioritise the utilisation of local (national) soil databases when it is deemed representative and reliable."

COMMENT 13

L675-685: you cannot propose this as the only way forward without making available your reference dataset (EU-HYDI)!!!

ANSWER 13

We intended to write that the methodology can be applied on other databases as well. We have modified the sentence in RL864 in the following way:
"The study's methodology can be applied for soil databases not only in Europe but also in other regions or global datasets, …"

COMMENT 14

L16: which are the underlying processes?

ANSWER 14

We have added some examples for the underlying processes in RL15-17:
" … it is important to simulate water quantity and quality patterns and to quantify the impact of land use and climate change on soil functions, soil health, hydrological, and other underlying processes."

COMMENT 15

L28-29: why there an significant increase of available datasets?

ANSWER 15

We have modified that sentence in RL29-30 in the following way:
"The availability of raw and derived soil datasets, specifically soil hydraulic data, has increased significantly in Europe over the last 10 years as a result of the Soil Strategy and Soil Monitoring Law proposed by the EU Commission."

COMMENT 16
L159: Not only different methods but also through different ISO protocols, depths, etc and in different laboratories which sometimes is impossible to compare.!!

ANSWER 16
We agree.
We have added ISO protocols in RL745:
"Different countries and institutions measure sand, silt, and clay content using different ISO protocols and methods by recognizing different cutoff limits and classification standards."

COMMENT 17
L325: It is nitrogen

ANSWER 17
Here we would like to refer to inorganic nitrogen forms, which are more soluble in water, therefore are more susceptible to loss through processes like denitrification, volatilization, and leaching. We have modified this sentence in RL413-415:
"The concentration of inorganic nitrogen in soil is highly variable in space and time and the dynamic of its amount is significantly influenced by leaching, denitrification, volatilization, and nitrogen fertilization (Zhu et al., 2021).

REFERENCES ADDED IN THE ANSWERS FOR ANONYMOUS REVIEWER 1:
Amorim, H.C.S., Hurtarte, L.C.C., Souza, I.F., Zinn, Y.L., 2022. C:N ratios of bulk soils and particle-size fractions: Global trends and major drivers. Geoderma 425, 116026. https://doi.org/10.1016/j.geoderma.2022.116026

Ballabio, C., Panagos, P. and Monatanarella, L. 2016. Mapping topsoil physical properties at European scale using the LUCAS database, Geoderma, 261, 110–123, doi:10.1016/j.geoderma.2015.07.006

Gupta, S., Borrelli, P., Panagos, P. and Alewell, C. 2024. An advanced global soil erodibility (K) assessment including the effects of saturated hydraulic conductivity, Sci. Total Environ., 908(July 2023), 168249, doi:10.1016/j.scitotenv.2023.168249

Liu, M., Ussiri, D.A.N., Lal, R., 2016. Soil Organic Carbon and Nitrogen Fractions under Different Land Uses and Tillage Practices. Commun. Soil Sci. Plant Anal. 47, 1528–1541. https://doi.org/10.1080/00103624.2016.1194993

Panagos, P., Van Liedekerke, M., Jones, A. and Montanarella, L. 2012. European Soil Data Centre: Response to European policy support and public data requirements, Land use policy, 29(2), 329–338, doi:10.1016/j.landusepol.2011.07.003

Panagos, P., De Rosa, D., Liakos, L., Labouyrie, M., Borrelli, P. and Ballabio, C. 2024. Soil bulk density assessment in Europe, Agric. Ecosyst. Environ., 364(October 2023), 108907, doi:10.1016/j.agee.2024.108907

Pu, X., Cheng, H., Shan, Y., Chen, S., Ding, Z., Hao, F., 2012. Factor controlling soil organic carbon and total nitrogen dynamics under long-term conventional cultivation in seasonally frozen soils. Acta Agric. Scand. Sect. B Soil Plant Sci. 62, 749–764. https://doi.org/10.1080/09064710.2012.700318

Renard, K. G., Foster, G. R., Weesies, G. A., McCool, D. K. and Yoder, D. C. 1997. Predicting soil erosion by water: a guide to conservation planning with the Revised Universal Soil Loss Equation (RUSLE), U.S. Department of Agriculture. [online] Available from: http://www.ars.usda.gov/SP2UserFiles/Place/64080530/RUSLE/AH_703.pdf

Román Dobarco, M., Cousin, I., Le Bas, C. and Martin, M. P.: Pedotransfer functions for predicting available water capacity in French soils, their applicability domain and associated uncertainty, Geoderma, 336(April 2018), 81–95, doi:10.1016/J.GEODERMA.2018.08.022, 2019.

Tranter, G., McBratney, a. B. and Minasny, B.: Using distance metrics to determine the appropriate domain of pedotransfer function predictions, Geoderma, 149(3–4), 421–425, doi:10.1016/j.geoderma.2009.01.006, 2009.

Zhai, X., Liu, K., Finch, D.M., Huang, Di, Tang, S., Li, S., Liu, H., Wang, K., 2019. Stoichiometric characteristics of different agroecosystems under the same climatic conditions in the agropastoral ecotone of northern China. Soil Res. 57, 875–882. https://doi.org/10.1071/SR18355

---

## Author Response (AR2)

Dear Editor,

Thank you for reviewing the revised manuscript and evaluating the reviewers' feedback on our responses and the modifications applied to the manuscript. Please find our point-by-point response to the reviews below this letter and the revised version of our manuscript uploaded.

With regards,
Authors

**ANSWER FOR ANONYMOUS REFEREE #1**

Dear Referee #1,

Thank you for evaluating the revision of the manuscript and providing further recommendations. Please find our answers below, following your suggestions. The changes applied to the manuscript text are highlighted in blue. Line numbering indicated as, e.g., "RL30" refers to the line number 30 in the revised manuscript with track changes.

COMMENT 1:
In the first paragraph of the introduction, the authors should be more precise. The increased availability of data is not related to the proposal of the Soil Monitoring Law . However, the increased importance of soils in different policy agendas takes place within the EU Green Deal which had an important impact. Just read the document of Montanarella et al (2021) on the importance of soils in the EU Green Deal. That impact had an effect in increased availability of soil data. Of course, LUCAS and other open accesses large scale datasets have been pioneer in this trend.

ANSWER 1:
Thank you for the information, we have corrected the sentence accordingly in RL30-32:
"The availability of raw and derived soil datasets, specifically soil hydraulic data, has increased significantly in Europe over the last 10 years as a results of the European Green Deal through initiatives and strategies aimed at promoting sustainable land use, soil health, and environmental protection (Montanarella and Panagos, 2021)."

COMMENT 2:
L51: you can also add the harvest cycle among the factors.

ANSWER 2:
Thank you, it has been added in RL52.

COMMENT 3:
In table 6, why mean (or median values) are not mentioned? Can you please explain how you calculate the Weighted rank?

ANSWER 3:

Table 6 follows the logic of tables showing the prediction performance (Tables 7, 10-14.) and therefore it includes error metrics. The following description on weighted rank has been added under Table 6 in LR317-319:

**Rank based on the Kruskal-Wallis test, 1 denotes the best performing method.*** Sample-number-weighted average results of the Kruskal-Wallis test.

COMMENT 4:

Table 9. Is it possible to bring the USA units (the sharply and Williams) in the metric system? This will allow to compare the results.

ANSWER 4:

Yes, we have highlighted in the caption of Table 9 that K values are provided in both U.S. Customary Unit $\left(\frac{t \cdot arce \cdot h}{hundreds\ of\ acre \cdot foot - tonf \cdot inch}\right)$ and SI Unit $\left(\frac{t \cdot ha \cdot h}{ha \cdot MJ \cdot mm}\right)$ in RL417-418, and edited the table to make it more evident that the results of both methods are given in the table in both units:

**Table 9.** Descriptive statistics of soil erodibility factor values computed with the Sharpley and Williams (1990) and Renard et al. (1997) equations on the topsoil samples of the EU-HYDI dataset (N = 11,287) provided in U.S. Customary Unit $\left(\frac{t \cdot arce \cdot h}{hundreds\ of\ acre \cdot foot - tonf \cdot inch}\right)$ and SI Unit $\left(\frac{t \cdot ha \cdot h}{ha \cdot MJ \cdot mm}\right)$ .

| Method | USLE K factor in different units | | | | | | |
|---|---|---|---|---|---|---|---|
| | Unit | Min | Max | Range | Mean | Median | Standard deviation |
| Sharpley and Williams (1990) | $\left(\frac{t \cdot arce \cdot h}{hundreds\ of\ acre \cdot foot - tonf \cdot inch}\right)$ | 0.00 | 0.48 | 0.48 | 0.27 | 0.27 | 0.09 |
| | $\left(\frac{t \cdot ha \cdot h}{ha \cdot MJ \cdot mm}\right)$ | 0.000 | 0.063 | 0.063 | 0.036 | 0.035 | 0.012 |
| Renard et al. (1997) | $\left(\frac{t \cdot arce \cdot h}{hundreds\ of\ acre \cdot foot - tonf \cdot inch}\right)$ | 0.05 | 0.33 | 0.29 | 0.24 | 0.27 | 0.09 |
| | $\left(\frac{t \cdot ha \cdot h}{ha \cdot MJ \cdot mm}\right)$ | 0.006 | 0.044 | 0.038 | 0.032 | 0.035 | 0.012 |

With regards,
Authors

**ANSWER FOR DIANA VIEIRA**

Dear Diana Vieira,

Thank you for reviewing our responses and the modifications applied to the manuscript. Your suggestions helped us significantly improve our manuscript.

With regards,
Authors